# RETuning: Upgrading Inference-Time Scaling for Stock Movement Prediction with Large Language Models

## Abstract

Recently, large language models (LLMs) have demonstrated outstanding reasoning capabilities and inference-time scaling on mathematical and coding tasks. However, their application to financial tasks—especially the most fundamental task of stock movement prediction—remains underexplored. We study a three-class classification problem (up, hold, down) and, by analyzing existing reasoning responses, observe that: (1) LLMs are easily swayed by contextual viewpoints, tending to follow analysts' opinions rather than exhibit a systematic, independent analytical logic in their chain-of-thoughts (CoTs). (2) LLMs often list summaries from different sources without weighing adversarial evidence, yet such counterevidence is crucial for reliable prediction. It shows that the model does not make good use of its reasoning ability to complete the task. To address this, we propose **R**eflective **E**vidence **Tuning** (**RETuning**), a cold-start method prior to reinforcement learning, to enhance prediction ability. While generating CoT, **RETuning** encourages to dynamically construct an analytical framework from diverse information sources, organize and score evidence for price up or down based on that framework—rather than on contextual viewpoints—and finally reflect to derive the prediction. This approach maximally aligns the model with its learned analytical framework, ensuring independent logical reasoning and reducing undue influence from context. We also build a large-scale dataset spanning all of 2024 for 5,123 A-share stocks, with long contexts (32K tokens) and over 200K samples. In addition to price and news, it incorporates analysts' opinions, quantitative reports, fundamental data, macroeconomic indicators, and similar stocks. Experiments on this new dataset show that, as a cold-start method, **RETuning** successfully unlocks the model's reasoning ability in the financial domain. During reinforcement learning, response length steadily increases under the designed curriculum setting. Furthermore, inference-time scaling still works even after 6 months or on out-of-distribution stocks, since the models gain valuable insights about stock movement prediction.

## 1 Introduction

**Stock Movement Prediction (SMP)** is one of the most fundamental and consequential tasks in finance. It not only directly affects the interests of individual investors but also plays a central role in algorithmic trading (Mahfooz et al., 2022; Ta et al., 2018), financial risk control (Adyatma & Alamsyah, 2022; Vui et al., 2013), and intelligent research platforms (Shi et al., 2020). In recent years, Large Language Models (LLMs) (Brown et al., 2020; Achiam et al., 2023; Bai et al., 2023; Touvron et al., 2023) have demonstrated remarkable reasoning capabilities in domains such as code generation and mathematical problem-solving (DeepSeek-AI et al., 2025; OpenAI et al., 2024; Uesato et al., 2022). This has sparked growing interest in exploring whether such models can also excel in financial tasks. However, it remains an open question whether LLMs' strength in *reasoning* and *inference-time scaling* can be effectively harnessed for stock price prediction.

In the traditional planning phase of a financial agent, the model has access to a wide range of information sources—news articles, analyst opinions, research reports, quantitative factor analyzes, and more. Despite this richness, making reliable and interpretable predictions remains a major challenge. On one hand, LLMs often exhibit strong prior biases due to the optimistic slant of their

training data, which is skewed toward long positions and excludes contrarian views for political or regulatory reasons. On the other hand, these models tend to lack the ability to construct independent reasoning frameworks, reconcile conflicting information, and perform reflective analysis—capabilities that are essential for robust financial decision-making.

To address these challenges, we propose a novel modeling paradigm that treats stock movement prediction as a *generative reasoning task*. By processing all textual information sources end-to-end, this approach aims to simulate the thought process of a human trader, ultimately generating structured and interpretable predictions. Two key innovations underpin this paradigm.

First, we introduce **Reflective Evidence Tuning (RETuning)**, which instills LLMs to dynamically construct reasoning frameworks based on diverse information sources, collect and evaluate evidence for potential price directions (up, down, or hold), and reflect on the evidence before making a final prediction. This structured approach is a cold-start training mechanism prior to reinforcement learning (RL). It enables models to avoid merely summarizing or echoing external viewpoints and instead follow an internally consistent logic, improving both interpretability and accuracy.

Second, we explore the role of **inference-time scalability**, a technique that has shown promise in mathematical and programming tasks (Muennighoff et al., 2025; Li et al., 2025). Specifically, we investigate whether *majority voting* can significantly improve predictive accuracy in financial domain. Although widely successful elsewhere, its efficacy in stock movement prediction has not yet been systematically examined.

To support this research, we construct a large-scale, high-quality dataset that reflects the complexity and information density of real-world financial environments. Covering the full year of 2024 across over 4,000 A-share stocks, this dataset integrates six heterogeneous information sources: news, fundamentals, analyst opinions, quantitative factor reports, macroeconomic context, and stocks of similar trends. With over 200,000 samples and an average input length of up to 32K tokens, it overcomes the limitations of prior datasets that were outdated and lacked information diversity (Xu & Cohen, 2018; Luo et al., 2023; Zhou et al., 2021). The details of the dataset construction are discussed in Appendix C.

Empirical results show that **RETuning** effectively enhances reasoning structure and improves predictive performance over strong baselines. It also generalizes beyond stock movement prediction, yielding significant improvements in other financial tasks, and demonstrates strong performance under inference-time scaling and out-of-distribution settings.

Our contribution can be summarized as follows: (1) We build a large-scale, long-context financial dataset with diverse evidence sources beyond price and news, which fill the gap that existing datasets are outdated and lack information diversity. (2) We introduce **RETuning**, synthesizing cold-start responses that guide LLMs to construct and reflect on an analytical framework for stock movement prediction. It allows significant inference-time scalability of LLMs in the prediction task. (3) We empirically show that **RETuning** unlocks prediction ability and generalizes beyond stock movement prediction. We believe this research lays the groundwork for deploying trustworthy, reasoning-driven LLMs in real-world financial applications.

## 2 RELATED WORK

**Stock Movement Prediction with LLMs.** Recent work has focused on exploring various types of information sources for stock movement prediction with LLMs. Several studies emphasize the importance of stock-related news in revealing fundamental market insights (Vargas et al., 2018; Li et al., 2021). Meanwhile, other research highlights the significance of understanding the relationships between companies and industries (Feng et al., 2019; Hsu et al., 2021). Recent studies have also provided empirical evidence of the impact of public sentiment on market trends, with researchers working to extract sentiment and keyphrases from news and social media data (Nguyen et al., 2015; Hao et al., 2021). However, these works often focus on a single type of information source, such as news or sentiment, and do not fully leverage the potential of LLMs to integrate multiple heterogeneous sources. In contrast, our work aims to construct a comprehensive dataset that incorporates diverse information sources, including news, fundamentals, analyst opinions, quantitative reports, macroeconomic indicators, and similar stocks, to enhance the predictive capabilities of LLMs in stock movement prediction.

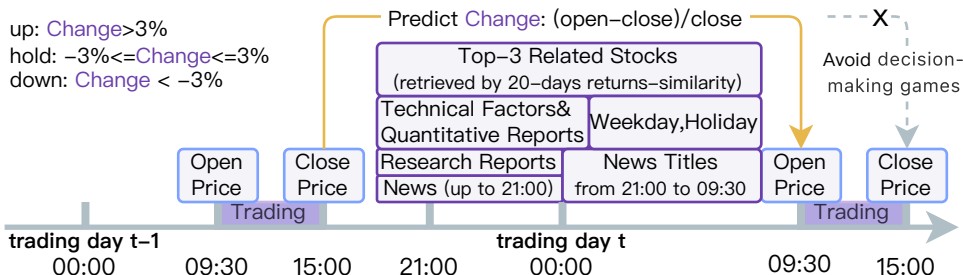

Figure 1: Overview of data sample for one stock at trading day t.

**Inference-Time Scaling for LLMs.** Inference-time scaling methods (Snell et al., 2024) enhance LLM performance by leveraging additional computation during generation, broadly categorized into three strategies. **Repeated sampling** improves diversity and accuracy via parallel candidate generation, utilizing verification strategies like majority voting (Li et al., 2024a; Lin et al., 2024; Wang et al., 2023; Toh et al., 2024) or best-of-N (BoN) selection with verifiers (Stiennon et al., 2020; Cobbe et al., 2021; Nakano et al., 2022; Li et al., 2023; Liu et al., 2025a), while efficiency optimizations prune low-scoring paths early (Zhang et al., 2024b; Qiu et al., 2024; Sun et al., 2024; Manvi et al., 2024; Ye & Ng, 2024). **Self-correction** iteratively refines outputs using feedback from tools, external models, or self-critique (Shinn et al., 2023; Gou et al., 2024; Li et al., 2024b; Song et al., 2025), though its efficacy depends on feedback reliability (Olausson et al., 2024; Huang et al., 2024; Wang et al., 2024a; Yang et al., 2024). **Tree searching** combines parallel and sequential scaling via algorithms like MCTS or A* (Yao et al., 2023; Xie et al., 2023; Long, 2023; Chari et al., 2025) guided by value functions (Xu, 2023; Hao et al., 2023; Chen et al., 2024; Zhang et al., 2024a). Training techniques distill these scaling benefits into more efficient models (Gao et al., 2023; Hou et al., 2024; Gulcehre et al., 2023; Zhang et al., 2024c). However, these methods have not been systematically applied to financial tasks, particularly stock movement prediction. Our work aims to fill this gap by exploring how inference-time scaling can be effectively utilized in this domain.

## 3 PRELIMINARIES

### 3.1 STRICT CONTROLLED DATASET FOR STOCK MOVEMENT PREDICTION

We aim to evaluate the ability of LLMs to predict stock movements based on diverse information sources. To this end, we construct a strict controlled dataset named **Fin-2024**, which covers the entire year of 2024, including 5,123 A-share stocks and 209,063 samples. Each sample is designed with a long context window of 32K tokens. Figure 1 illustrates the data sample for one stock at trading day $t$. LLMs will be trained from January to November, and then evaluated on December. In addition, we also collect data **Fin-2025[June]** on June 2025 for long-horizon prediction evaluation. The dataset construction process is detailed in Appendix C.

**Information Sources** The dataset consists of diverse information sources that have been proven valuable in machine learning-based quantitative trading research. These include: (1) **News articles** providing real-time market updates and company-specific & sector-specific information, (2) **Fundamental reports** reflecting company financial health and performance, (3) **Analyst opinions** offering professional market insights, (4) **Quantitative reports** containing technical analysis and market indicators, (5) **Macroeconomic indicators** showing broader economic trends, and (6) **Similar stocks information** for comparative analysis. The information are primarily textual, friendly for LLMs.

**Prediction Target** We define the prediction target as the price movement between the current trading day's opening price and the previous trading-day's closing price. This setting is less common in pre-training data compared to closing price-based movements, which helps prevent the model from exploiting memorization. The setting also avoids the model from decision-making games in trading periods, which is hard to capture in the given context. Based on price change, we classify the stock movement into three classes: **up** for change > 3%, **down** for < -3%, and **hold** for else. The

three-class classification scheme requires more significant signals for price movements than binary classification. The **hold** class serves as a decoy. If the model never learns to distinguish between **up** and **down**, it would indicate a shortage on the model's ability to make price movement predictions.

**Evaluation Protocol**    During evaluation, we require the model to simultaneously predict both the price change percentage and direction to assess its instruction-following capability and verify the consistency between the two predictions, as the direction should align with the predicted change percentage.

## 3.2    OBSERVATION ON EXISTING MODELS

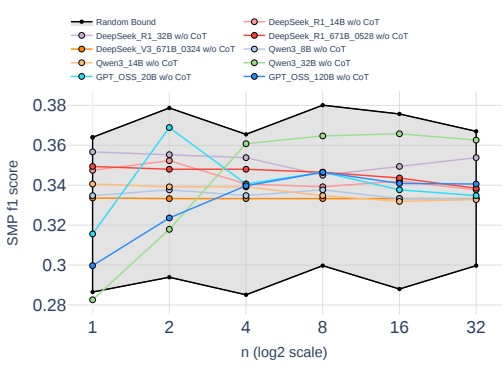 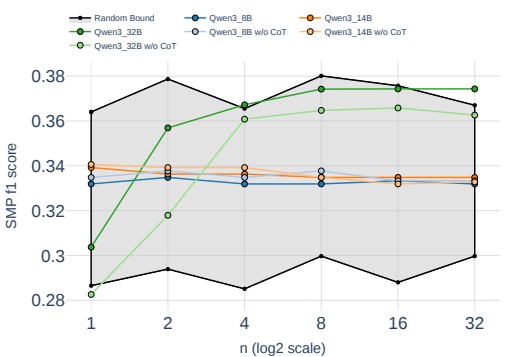

Figure 2: Existing LLMs without RETuning CoT perform no better than random guessing in stock movement prediction, and most (except Qwen3 32B) fail to scale at inference time.

Figure 3: Abaltion on CoT prompting. The proposed CoT in Section 4.1 can help improve the prediction performance on Qwen3 32B, but not smaller models. Others lose to random guessing.

To understand the limitations of existing models in stock movement prediction, we analyze their performance on **Fin-2024[December]** in Section 3.1. We use 32 different random seeds to uniformly sample the prediction (hold, up, and down) to construct the random bound (grey area in the figure). Any results that are covered by the random bound will be regarded failing to make trust-worthy prediction. The results reveal two key issues:

Firstly, by investigating existing LLMs' performance in Figure 2, we observe that current LLMs are almost randomly guessing the prediction result. And the most models cannot scale their ability of prediction at inference time. Secondly, we prompt the model with fine-grained CoT (Section 4.1) to inject knowledge of financial analysis into the reasoning process. We further analyze the responses of the models, as shown in Figure 3. We find that CoT can help improve the model's prediction performance on Qwen3 32B, but not other models.

To understand why these models fail, we further sample multiple responses for case study (Appendix E.1) and find that: the outputs of these LLMs tend to be vague, detached from the prediction target, and biased toward the `hold` class on label-balanced datasets. Thus, we propose to utilize supervised fine-tuning to induce coherent, task-specific reasoning to change the output distribution of these models, as discussed in the following section.

## 4    REFLECTIVE EVIDENCE TUNING (RETUNING)

This section introduces **Reflective Evidence Tuning (RETuning)**, a two-stage framework designed to unlock the latent reasoning ability of LLMs in stock movement prediction tasks. As illustrated in Figure 4, RETuning comprises: (1) an SFT stage to *cold-start generative reasoning modeling*, and (2) *rule-based reinforcement learning* for performance refinement and alignment.

## 4.1    STOCK MOVEMENT PREDICTION VIA GENERATIVE REASONING MODELING

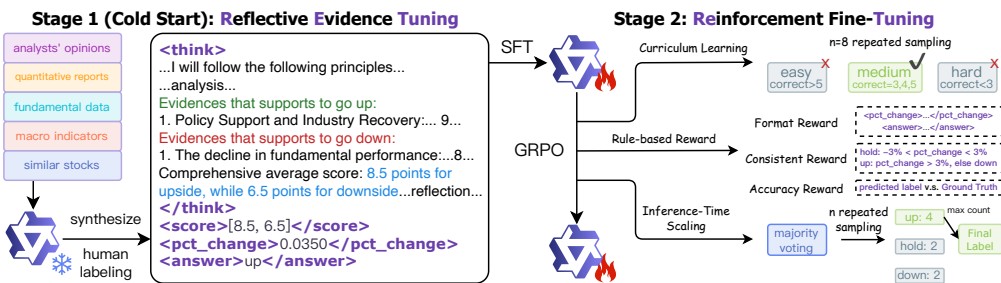

Figure 4: Two-stage stock movement prediction model training framework: **Stage 1** (Cold Start) uses multi-source data with human labeling and synthesis pipeline for Reflective Evidence Tuning; **Stage 2** applies Reinforcement Fine-Tuning with curriculum learning, reward shaping, and inference-time scaling for final label determination.

We frame the SMP task as a *generative reasoning problem*, in which the LLM leverage its reasoning ability to make predictions. It must construct an analytical framework, extract and score evidence from heterogeneous sources, and reflect before reaching a conclusion. As is shown in Figure 5a, this contrasts with zero-shot settings, where models superficially summarize and avoid making grounded predictions.

Then we employ supervised fine-tuning (SFT) to instill this reasoning structure into the model. The SFT dataset is constructed through a semi-automated pipeline (Appendix C.2) that uses DeepSeek-R1 (671B) as the backbone model to rejected sampling to synthesize 118 golden cold-start items.

By grounding each prediction in a dynamically built framework, **RETuning** promotes robust and context-aware reasoning. This reduces susceptibility to dominant context bias and improves the model's ability to rationally weigh adversarial evidence—crucial for reliable financial forecasting. By fine-tuning on this structured reasoning process, we find that the model DeepSeek_R1_14b_SFT can scale its prediction performance via repeated sampling far beyond random guessing, as shown in Figure 5b. It indicates that the model already possesses a certain level of predictive capability. We can further leverage this weak predictive power to assess the difficulty of predicting samples, thereby enhancing the overall predictive performance of the model more efficiently.

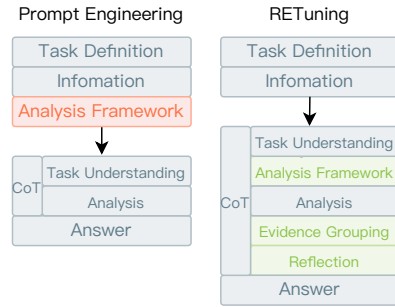

(a) **RETuning** guides the model to generate a principle, collect evidence, and reflect before making a prediction.

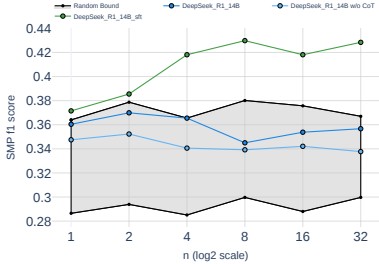

(b) DeepSeek_R1_14B_SFT scales prediction performance via repeated sampling.

## 4.2 RULE-BASED REINFORCEMENT LEARNING

To further align model outputs with desired reasoning behavior, we introduce a rule-based reinforcement learning (RL) stage. Rather than relying on simple correctness-based rewards—which are noisy and statistically uninformative in financial prediction—we design more principled signals through **reward shaping** and **curriculum learning**.

**Reward Shaping** We design a **multi-faceted reward function** to capture both the structural and semantic correctness of model outputs. First, a **Format Score** ensures that the response adheres to the expected structured format, maintaining clarity and consistency. The **Accuracy Score** focuses on whether the model correctly predicts the directional movement—up, down, or hold. Lastly, the **Consistency Score** encourages logical alignment between the predicted percentage change and the stated directional label. Final score is given by $R = \alpha \cdot \text{Format} + \beta \cdot \text{Accuracy} + \gamma \cdot \text{Consistency}$, where $\alpha, \beta, \gamma$ are hyperparameters. This design mitigates the issue of misleading signals from noisy.

**Curriculum Learning**   Not all samples are equally difficult: **hold** predictions are often trivial, while confident **up/down** predictions require strong signal integration. To make training more efficient and targeted, we propose a curriculum learning strategy:

We use the cold-started model to generate $N$ (=8 in practice) predictions for each training sample. The difficulty of a sample is measured by counting how many of these predictions are incorrect. Based on this difficulty score, we categorize examples into three groups: **easy** (correct $\in [\frac{2}{3}N, N)$ ), **medium** (correct $\in [\frac{1}{3}N, \frac{2}{3}N)$ ), and **hard** (correct $\in [0, \frac{1}{3}N)$ ).

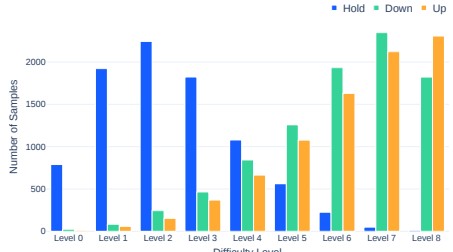

Figure 6: Difficulty distribution given by the cold-started model (DeepSeek_R1_14B_SFT).

In Figure 6, we observe that a clear correlation between difficulty levels and labels. Low-difficulty samples are mostly dominated by **hold** predictions, which tend to be either spurious or too simple to be informative. High-difficulty samples, on the other hand, often involve **up** or **down** predictions but with weak or noisy signals. In contrast, medium-difficulty samples tend to reflect realistic market complexities and require non-trivial reasoning. To ensure the model focuses on meaningful learning signals, we discard both low and high-difficulty examples and train only on medium-difficulty ones, progressing in order of increasing difficulty.

**Inference-Time Scaling**   We apply majority voting on predicted labels over $n$ repeated generations with temperature 0.6. The final decision is given by: $\hat{y} = \arg\max_{y \in \{\text{up},\text{down},\text{hold}\}} \sum_{i=1}^{n} \mathbf{1}[y_i = y]$

## 5 EXPERIMENT AND RESULTS

We conduct several experiments to ascertain the effectiveness of **RETuning**, with the aim to gain insights into the following: (1) Basically, can we improve the stock movement prediction performance of LLMs? How does **RETuning** compare with existing methods? What insights does the model learn from the data? (2) Can we scale the prediction performance of LLMs at inference time? (3) What are the key factors that contribute to the success of **RETuning**? (4) Does the enhanced prediction ability contribute to other financial tasks?

### 5.1 EXPERIMENT SETUP

**Datasets.**   We use the data from January to November in **Fin-2024** for training, and the December data **Fin-2024[December]** for testing. We also use the **Fin-2025[June]** dataset to evaluate whether the model persists its scaling ability on prediction performance after 6 months. Besides, we evaluate the generalization ability on **BizFinBench** (Lu et al., 2025), a comprehensive financial benchmark covering 10 tasks, including Anomalous Event Attribution (AEA), Financial Time Reasoning (FTR), Financial Tool Usage (FTU), Financial Numerical Computation (FNC), Financial Knowledge QA (FQA), Financial Data Description (FDD), Emotion Recognition (ER), Stock Price Prediction (SP), Financial Named Entity Recognition (FNER). The details of the datasets are shown in Appendix C.

**Evaluation and Metrics.**   We care about the generalization ability and consider three types of out-of-distribution (OOD) settings: **OOD_Stock**, **OOD_Date**, and **OOD_Stock&Date**. We choose 50 stocks in random as the OOD stocks and the last month of 2024 as the OOD dates. The 50 stocks are also OOD stocks in **Fin-2025[June]**. We adopt the standard metrics F1-score because it balances precision and recall, making it suitable for our multi-class classification task.

**Implementation and Baselines.**   The models are trained on up to 4*8 H100 GPUs. Rollout $n$ is set to 8. Default results are obtained by greedy decoding. For inference-time scaling, we use $k \in \{1, 2, 4, 8, 16, 32\}$ and temperature=0.6. More implementation details are shown in Appendix B. Based on DeepSeek_R1_14B_Instruct (originally DeepSeek-R1-Distill-Qwen-14B (DeepSeek-AI et al., 2025)), we apply **RETuning** and get DeepSeek_R1_14B_SFT and DeepSeek_R1_14B_SFT_GRPO, which are after SFT stage and after SFT + GRPO stages, respectively. We also implement DeepSeek_R1_32B_SFT and DeepSeek_R1_32B_SFT_GRPO based on

DeepSeek_R1_32B_Instruct. We compare to several strong baselines, including: LLMFactor (Wang et al., 2024b), Fino1 (Qian et al., 2025), Fin-R1 (Liu et al., 2025b), CMIN (Luo et al., 2023) and StockNet (Xu & Cohen, 2018). We also report results of several state-of-the-art open-weight LLMs: DeepSeek (DeepSeek-AI et al., 2025) (R1-7B, R1-14B, R1-32B, R1-671B, V3-671B), Qwen3 (Yang et al., 2025) (8B, 14B, 32B), GPT-OSS (OpenAI et al., 2025) (20B, 120B).

## 5.2 RESULTS AND ANALYSIS

Table 1: Results of different methods on **Fin-2024[December]** benchmarks. w/ CoT means using the CoT prompting in Section 3.2. The relative improvements (%) over the baselines are shown in parentheses. The best results are in **bold**.

| Model | F1 Score |
|---|---|
| *Results of Public Models* | |
| Random Guessing | 0.3333 |
| LLMFactor (Wang et al., 2024b) | 0.3345 |
| Fino1 (Qian et al., 2025) | 0.0622 |
| Fin-R1 (Liu et al., 2025b) | 0.2543 |
| CMIN (Luo et al., 2023) | 0.3275 |
| StockNet (Xu & Cohen, 2018) | 0.3081 |
| *Results with CoT Ablation* | |
| Qwen3_8B (Yang et al., 2025) | 0.3348 |
| w/ CoT | 0.3319 (-0.87%) |
| Qwen3_14B (Yang et al., 2025) | 0.3406 |
| w/ CoT | 0.3392 (-0.41%) |
| Qwen3_32B (Yang et al., 2025) | 0.2826 |
| w/ CoT | 0.3037 (+7.47%) |
| GPT_OSS_20B (OpenAI et al., 2025) | 0.3156 |
| w/ CoT | 0.3249 (+2.95%) |
| GPT_OSS_120B (OpenAI et al., 2025) | 0.2997 |
| w/ CoT | 0.3436 (+14.65%) |
| DeepSeek_R1_671B_0528 | 0.3333 |
| w/ CoT | 0.3494 (+4.83%) |
| DeepSeek_V3_671B_0324 | 0.3336 |
| w/ CoT | 0.3456 (+3.60%) |
| *Results of Our Models* | |
| DeepSeek_R1_14B_Instruct | 0.3475 (baseline) |
| w/ CoT | 0.3604 (+3.71%) |
| DeepSeek_R1_14B_GRPO (**Ours**) | 0.3377 (-2.25%) |
| DeepSeek_R1_14B_SFT (**Ours**) | 0.3715 (+6.91%) |
| DeepSeek_R1_14B_SFT_GRPO (**Ours**) | **0.4196** (+20.75%) |
| DeepSeek_R1_32B_Instruct | 0.3567 (baseline) |
| w/ CoT | 0.3589 (+0.62%) |
| DeepSeek_R1_32B_GRPO (**Ours**) | 0.3683 (+3.22%) |
| DeepSeek_R1_32B_SFT (**Ours**) | 0.3639 (+2.02%) |
| DeepSeek_R1_32B_SFT_GRPO (**Ours**) | 0.4071 (+14.13%) |

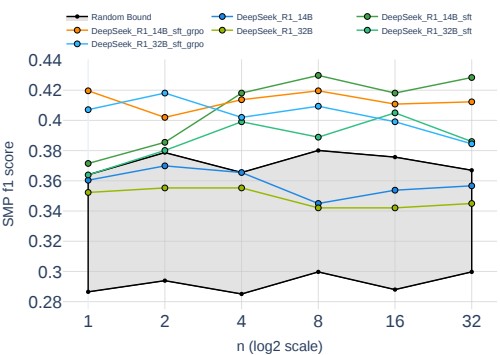

Figure 7: Inference-time scalability results on **Fin-2024[December]**. SFT model already has prediction ability, and GRPO further refines it.

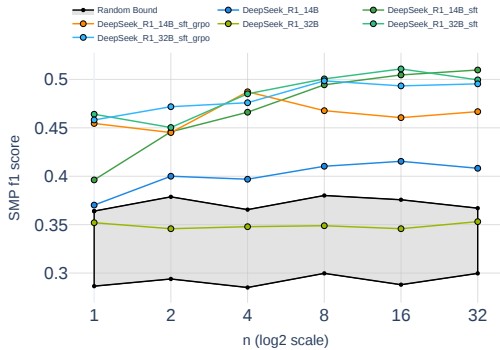

Figure 8: Inference-time scalability results on **Fin-2025[June]**. Finetuned models continue to scale via repeated sampling even after 6 months.

**RETuning vs baselines.** The results of different methods on **Fin-2024[December]** benchmarks are shown in Table 1. We observe that RETuning (SFT + GRPO) significantly outperforms all baselines, including state-of-the-art open-weight LLMs (DeepSeek, Qwen3, GPT-OSS) and public models specifically designed for stock movement prediction (LLMFactor, Fino1, Fin-R1, CMIN, StockNet). For instance, DeepSeek_R1_14B_SFT_GRPO achieves an F1 score of 0.4196, which is a 20.75% relative improvement over its instruct baseline (0.3475) and surpasses the best public model (GPT-OSS-120B w/ CoT at 0.3436) by 22.15%. Similarly, DeepSeek_R1_32B_SFT_GRPO attains an F1 score of 0.4071, marking a 14.13% relative improvement over its instruct baseline (0.3567) and outperforming the best public model by 18.55%.

**Can stock movement prediction benefit from inference-time scaling?** Yes, but the gains from inference-time scaling are limited. Figure 7 and Figure 8 present the inference-time scalability results on **Fin-2024[December]** and **Fin-2025[June]**, respectively. We observe monotonic or near-monotonic improvements up to $n \approx 8$–16, after which returns plateau and can even regress for some settings. RL (GRPO) makes test-time scaling less necessary by improving one-sample quality, yet does not increase the peak accuracy.

**Can predictive ability generalize to unseen stocks, future dates, or both?** Yes. We evaluate out-of-distribution (OOD) robustness along two axes: unseen stocks and forward-in-time generalization. The dataset **Fin-2024[December]** consists of **OOD_Stock**, **OOD_Date**, and **OOD_Stock&Date** cases, where RETuning maintains or increases F1 score as the number of inference-time samples $n$ grows (Figure 7). On **Fin-2025[June]** (future dates only), RETuning preserves its advantage and continues to benefit from moderate repeated sampling (Figure 8), indicating strong temporal and cross-ticker generalization.

To further determine how the model scales on different OOD cases, we group the results by OOD split and present in Figure 91011. The scaling is significant in **OOD_Stock**, then is **OOD_Date**. **OOD_Stock&Date** is the hardest cases to scale up, but it still outperforms baselines. We leave detailed analysis in Appendix D.2.

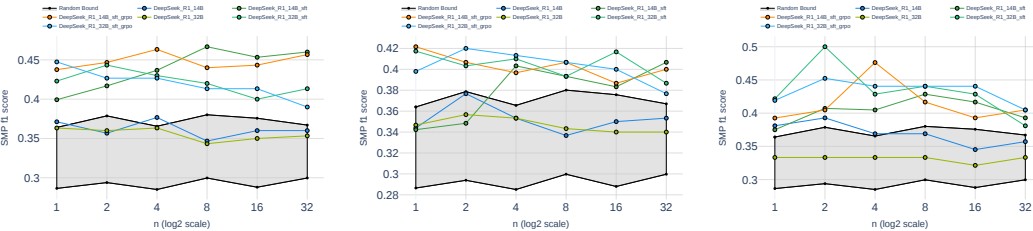

Figure 9: **OOD_Stock** results on **Fin-2024[December]**

Figure 10: **OOD_Date** results on **Fin-2024[December]**

Figure 11: **OOD_Stock&Date** results on **Fin-2024[December]**

We also explore how the model scales on different ground truth labels. The results are grouped by ground truth label and presented in Figure 121314. The scaling is significant in **hold** cases, and the model performance exceeds the baseline on **up** and **down** cases. We claim that **up** and **down** cases are more challenging, and **RETuning** enhances the model's performance in these scenarios by enabling it to better leverage its reasoning capabilities, thus to identify factors influencing stock movements more effectively, thereby allowing the model to make more accurate predictions.

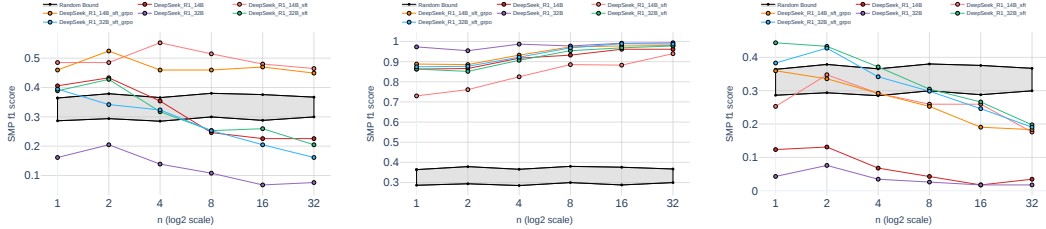

Figure 12: Ground truth **up** results on **Fin-2024[December]**

Figure 13: Ground truth **hold** results on **Fin-2024[December]**

Figure 14: Ground truth **down** results on **Fin-2024[December]**

**Ablation on CoT prompting.** We further examine the effect of Chain-of-Thought (CoT) prompting (Table 1). It causes slightly negative effects on smaller models (Qwen3-8B, Qwen3-14B), but significantly benefits larger models (Qwen3-32B, GPT-OSS-120B, DeepSeek-R1-671B, DeepSeek-V3-671B). This suggests that larger models have a greater capacity to leverage CoT prompting effectively. For the 14B model, CoT improves the baseline by +3.71%, while the 32B model only gains a marginal +0.62%. This indicates that CoT alone yields limited benefits. In contrast, our SFT and SFT+GRPO variants consistently outperform CoT, suggesting that structured fine-tuning and reinforcement optimization are more effective than relying on prompting strategies alone.

**Ablation on SFT stage.** We compare the effect of applying GRPO directly versus combining it with an SFT stage (*_SFT_GRPO). For the 14B model, GRPO alone underperforms the baseline (0.3377 vs. 0.3475), while SFT followed by GRPO achieves a substantial gain (0.4196, +20.75%). A similar trend is observed in the 32B model: GRPO alone yields modest improvement (0.3683, +3.22%), whereas SFT+GRPO achieves the best performance (0.4071, +14.13%). These results highlight that the SFT stage provides essential initialization, enabling GRPO to realize its full benefit.

Table 2: Performance Comparison of LLMs on BizFinBench (Lu et al., 2025). The colors represent the top three performers for each task: golden indicates the top-performing model, silver represents the second-best result, and bronze denotes the third-best performance.

| Model[†] | AEA | FNC | FTR | FTU | FQA | FDD | ER | SP | FNER | Average |
|---|---|---|---|---|---|---|---|---|---|---|
| Closed-Source LLMs | | | | | | | | | | |
| ChatGPT-o3 | 86.23 | 61.30 | 75.36 | 89.15 | 91.25 | 98.55 | 44.48 | 53.27 | 65.13 | 73.86 |
| ChatGPT-o4-mini | 85.62 | 60.10 | 71.23 | 74.40 | 90.27 | 95.73 | 47.67 | 52.32 | 64.24 | 71.29 |
| GPT-4o | 79.42 | 56.51 | 76.20 | 82.37 | 87.79 | 98.84 | 45.33 | 54.33 | 65.37 | 71.80 |
| Gemini-2.0-Flash | 86.94 | 62.67 | 73.97 | 82.55 | 90.29 | 98.62 | 22.17 | 56.14 | 54.43 | 69.75 |
| Claude-3.5-Sonnet | 84.68 | 63.18 | 42.81 | 88.05 | 87.35 | 96.85 | 16.67 | 47.60 | 63.09 | 65.59 |
| Open-Weight LLMs | | | | | | | | | | |
| Qwen3-14B | 84.20 | 58.20 | 65.80 | 82.19 | 84.12 | 92.91 | 33.00 | 52.31 | 50.70 | 67.05 |
| Qwen3-32B | 83.80 | 59.60 | 64.60 | 85.12 | 85.43 | 95.37 | 39.00 | 52.26 | 49.19 | 68.26 |
| DeepSeek_R1_14B_Instruct[1] | 71.33 | 44.35 | 50.45 | 81.96 | 85.52 | 92.81 | 39.50 | 50.20 | 52.76 | 59.49 |
| DeepSeek_R1_32B_Instruct[1] | 73.68 | 51.20 | 50.86 | 83.27 | 87.54 | 97.81 | 41.50 | 53.92 | 56.80 | 66.29 |
| Our LLMs | | | | | | | | | | |
| DeepSeek_R1_14B_SFT[2] | 80.63 | 51.67 | 52.61 | 83.53 | 89.05 | 96.72 | 36.68 | 50.43 | 50.85 | 65.36 |
| 14B $\Delta_{\text{Instruct}}$(SFT) | +9.25 | +7.28 | +2.19 | +1.53 | +3.42 | +3.85 | -2.93 | +0.24 | -1.92 | +5.83 |
| DeepSeek_R1_14B_SFT_GRPO[2] | 81.46 | 52.41 | 53.47 | 83.57 | 89.02 | 95.58 | 36.83 | 54.06 | 51.24 | 66.92 |
| 14B $\Delta_{\text{Instruct}}$(SFT_GRPO) | +10.03 | +8.09 | +2.91 | +1.64 | +3.45 | +2.63 | -2.74 | +3.82 | -1.53 | +7.46 |
| 14B $\Delta_{\text{SFT}}$(SFT_GRPO) | +0.82 | +0.85 | +0.81 | +0.06 | 0.00 | -1.23 | +0.25 | +3.63 | +0.42 | +1.53 |
| DeepSeek_R1_32B_SFT | 80.45 | 66.42 | 63.28 | 86.88 | 88.43 | 93.76 | 46.05 | 55.27 | 68.41 | 70.08 |
| 32B $\Delta_{\text{Instruct}}$(SFT) | +6.75 | +15.23 | +12.37 | +3.64 | +0.83 | -4.14 | +4.52 | +1.25 | +11.63 | +3.75 |
| DeepSeek_R1_32B_SFT_GRPO | 80.67 | 66.83 | 64.45 | 86.79 | 88.52 | 91.26 | 45.68 | 54.83 | 67.75 | 70.44 |
| 32B $\Delta_{\text{Instruct}}$(SFT_GRPO) | +6.95 | +15.62 | +13.57 | +3.55 | +0.93 | -6.64 | +4.13 | +0.85 | +10.93 | +4.12 |
| 32B $\Delta_{\text{SFT}}$(SFT_GRPO) | +0.23 | +0.42 | +1.23 | -0.06 | +0.13 | -2.53 | -0.42 | -0.43 | -0.72 | +0.33 |

[†] Closed-source LLMs results are sourced from Lu et al. (2025). Open-weight LLMs results are reproduced using temperature 0.6.
[1] DeepSeek_R1_14B_Instruct (32B) here is the short of DeepSeek-R1-Distill-Qwen-14B (32B) in the original paper (DeepSeek-AI et al., 2025).
[2] DeepSeek_R1_14B_SFT and DeepSeek_R1_14B_SFT_GRPO are DeepSeek_R1_14B_Instruct model after RETuning SFT stage and after RETuning SFT + GRPO stages, respectively.

**Does the enhanced predictive ability contribute to other financial tasks?** Yes. On the financial benchmark BizFinBench (Lu et al., 2025) (Table 2), **RETuning** generalizes beyond SMP: for 14B, the average score improves from 59.49 (Instruct) to 65.36 (SFT, +5.83) and 66.92 (SFT+GRPO, +7.46); for 32B, it improves from 66.29 to 70.08 (+3.75) and 70.44 (+4.12). The 32B models reach top-3 results on several tasks (e.g., FNC, FTU, ER, SP, FNER), while minor regressions appear on highly saturated dimensions (e.g., FDD after RL: −1.23 for 14B, −2.53 for 32B). Overall, **RETuning** yields broad, transferable gains with small trade-offs on a few tasks.

**What insights does model learn to predict stock movement?** Through analyzing the model's responses detailed in Appendix E.2, we find that the model learns to: 1. Identifying key evidences that influence daily fluctuations in stock prices from multiple information sources. 2. Trending to correctly evaluate the short-term impact of gathered evidences, which benefits from the trade-off ability induced by adversarial scoring. 3. Gradually improving the consistency between the predicted fluctuation and the movement label during reinforcement learning, which we named "vibe prediction".

# 6 CONCLUSION

In this work, we explored the underexamined problem of applying large language models (LLMs) to stock movement prediction. Our analysis revealed that vanilla LLMs tend to rely on contextual viewpoints rather than developing independent analytical reasoning, which limits their predictive reliability. To address this challenge, we introduced **RETuning**, a reflective evidence-based tuning method that encourages models to construct analytical frameworks, weigh adversarial evidence, and refine predictions through reflection. Experiments on our newly constructed large-scale financial dataset demonstrate that RETuning substantially improves predictive performance over strong baselines, enabling more systematic reasoning in the financial domain. Moreover, RETuning generalizes beyond stock movement prediction, yielding gains across diverse financial tasks, and exhibits robustness under inference-time scaling and out-of-distribution settings. Overall, this study highlights the importance of evidence-oriented reasoning in financial LLMs and establishes RETuning as a promising direction for enhancing their reliability and applicability in real-world financial decision-making.

## ETHICS STATEMENT

To address potential ethical considerations related to our research on large language models (LLMs) for stock movement prediction, we provide the following statement:

First, regarding data ethics: Our large-scale dataset (spanning 2024 for 5,123 A-share stocks) is constructed exclusively from publicly available sources, including market price data, publicly disclosed news, analysts' public opinions, company fundamental reports, official macroeconomic indicators, and publicly accessible information on peer stocks. We strictly comply with China's Data Security Law, Securities Law, and relevant financial regulatory requirements, ensuring no collection or use of private, sensitive personal data, or non-public material information that could violate market fairness.

Second, on potential harm and application boundaries: This research is conducted for academic purposes only to advance LLM reasoning capabilities in financial tasks. We explicitly emphasize that our model (RETuning) and findings do not constitute financial advice, nor do they endorse or promote real-world investment decisions. Stock market prediction inherently carries high uncertainty, and any practical application of such models for trading could lead to financial risks; we disclaim responsibility for any losses arising from non-academic use of our work.

Third, on bias and fairness: While we designed RETuning to enhance independent logical reasoning (reducing undue reliance on contextual viewpoints) and constructed a diverse dataset to cover a broad range of A-share stocks, we acknowledge potential residual biases (e.g., sector-specific skews in training data or sensitivity to market cycles). Future work will further validate and mitigate such biases to improve the model's fairness across different market scenarios.

Finally, regarding research integrity: Our study involves no human subjects, so institutional review board (IRB) approval is not applicable. We commit to transparency in dataset construction details and methodology implementation (as detailed in the full paper) to enable reproducibility. We adhere to rigorous academic standards to avoid misrepresentation of results or misuse of technical insights.

## REPRODUCIBILITY STATEMENT

We make every effort to ensure that the experiments in this paper are reproducible. Specifically, anonymized source code (training and evaluation scripts), model checkpoints, and processed dataset splits will be released as supplementary material and at the repositories indicated in the Appendix.[1] The Appendix contains detailed descriptions of data collection and preprocessing (Appendix C), the prompt templates and example inputs/outputs (Figures 17–27), and the exact training and evaluation settings including compute and hardware details (Appendix B, Tables 3 and 4). Evaluation splits used for OOD and long-horizon tests (e.g., **Fin-2024[December]**, **Fin-2025[June]**) and the scripts to compute all reported metrics will also be provided. Where full raw data cannot be released due to third-party licensing or privacy constraints, we describe the access procedure and provide processed, reproducible derivatives in the supplementary materials.

## BROADER IMPACT

This work investigates the use of large language models (LLMs) for stock movement prediction, a domain with potentially high economic and societal implications. Our proposed method, RETuning, demonstrates how reflective evidence organization can enhance independent reasoning in financial tasks. On the positive side, this research contributes to the broader effort of making LLMs more reliable in high-stakes applications by encouraging systematic analysis rather than context-driven imitation. Such improvements may benefit both academic research in reasoning and practical applications in financial decision-support systems.

At the same time, we emphasize that financial forecasting is inherently uncertain and subject to market volatility, regulatory constraints, and ethical considerations. Automated prediction systems, if misused, could amplify risks, encourage speculative behavior, or contribute to unfair advantages for

---

[1] https://anonymous.4open.science/r/RETuning, https://anonymous.4open.science/r/RETuning, https://anonymous.4open.science/r/RETuning

certain market participants. Our dataset and methods are designed for research purposes only, and we strongly discourage their direct use for live trading or investment without rigorous safeguards, human oversight, and compliance with financial regulations.

More broadly, this work highlights both the opportunities and limitations of applying LLMs in sensitive domains. We hope that our findings spur further research into building transparent, evidence-based reasoning frameworks that improve model reliability while also ensuring responsible deployment in practice.

## LIMITATIONS

**Domain specificity.** Our study focuses on the Chinese A-share market in 2024, which provides a rich testbed but may limit the generalizability of findings to other markets, such as U.S. equities or emerging markets with different structures, liquidity, and regulatory conditions.

**Data coverage.** Although our dataset integrates multiple information sources (prices, news, analyst opinions, fundamentals, and macroeconomic indicators), it remains incomplete. Certain high-frequency signals (e.g., intraday order flow, alternative data, or global macro shocks) are not incorporated, potentially constraining predictive accuracy.

**Model assumptions.** RETuning assumes that LLMs can benefit from explicitly structuring and reflecting on evidence. While this holds in our experiments, the approach may be less effective in domains like healthcare or cryptocurrency, where reliable evidence is scarce, noisy, or difficult to formalize.

**Evaluation scope.** Our evaluation mainly relies on F1 scores for three-class stock movement prediction and selected benchmarks (e.g., BizFinBench (Lu et al., 2025)). Broader metrics, such as profitability in trading simulations or risk-adjusted returns, are not considered. It takes time to validate real-world trading performance, which is beyond the scope of this paper.

**Computation and scalability.** Both training (SFT + GRPO) and inference-time scaling are computationally expensive. This may limit accessibility for smaller institutions or researchers without large-scale compute resources, raising questions about cost-efficiency in real-world deployment.

## FUTURE WORK

**Extending to other markets.** Future research could examine the effectiveness of RETuning across diverse markets such as U.S. equities, European exchanges, and emerging markets. This would validate whether the approach generalizes under different regulatory regimes, liquidity conditions, and investor behaviors.

**Incorporating richer data sources.** Enhancing the dataset with high-frequency trading signals, alternative data (e.g., satellite imagery, ESG reports), and global macroeconomic factors could provide a more comprehensive information environment and further strengthen predictive power.

**Advancing reasoning frameworks.** While RETuning encourages evidence-based reasoning, future work may integrate causal inference, probabilistic reasoning, or game-theoretic perspectives to capture deeper structures behind stock movements and reduce susceptibility to spurious correlations.

**Evaluation beyond prediction accuracy.** A natural next step is to link model predictions with financial outcomes, such as profitability, portfolio optimization, and risk-adjusted returns. This would bridge the gap between benchmark metrics and real-world decision-making.

**Efficiency and accessibility.** Research on lightweight RETuning variants, parameter-efficient finetuning, and inference-time acceleration could reduce computational costs, making the method more accessible to practitioners and researchers with limited resources.

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

# Appendix

## A   THE USAGE OF LARGE LANGUAGE MODELS (LLMs)

This section details the specific role of Large Language Models (LLMs) in this paper.

We employ LLMs *GPT-5* (developed by OpenAI) and *Doubao* (developed by ByteDance) to enhance the clarity, coherence, and overall quality of our manuscript. The LLM assistance primarily focus on language polishing (refining structure, terminology consistency, grammar) and formatting adjustments, ensuring that the paper meets high standards of academic writing. No other LLMs were used for research ideation or image generation.

All reviewed/approved by authors. All authors bear *full responsibility* for the final paper. All content (including LLM-generated/polished text) was verified: factual claims cross-checked against datasets/literature, and the manuscript screened to avoid unintended plagiarism.

## B   IMPLEMENTATION DETAILS

### B.1   OPEN SOURCE, OPEN WEIGHTS, AND OPEN DATA

The source code is available at GitHub[2]. The model weights are available at HuggingFace[3]. The training and evaluation datasets are available at HuggingFace[4].

### B.2   MODEL TRAINING

The models are trained on up to 4*8 H100 GPUs. Rollout $n$ is set to 8. The training epochs are set to 3 for SFT and 1 for RL. The details of the data synthesis workflow for building SFT dataset are shown in Appendix C.2. With the help of the SFT model to determine the prediction difficulty, we further apply GRPO Shao et al. (2024) with curriculum learning and reward shaping. The objective is

$$
\mathcal{J}_{GRPO}(\theta) = \mathbb{E}[q \sim P(Q), \{o_i\}_{i=1}^{G} \sim \pi_{\theta_{old}}(O|q)] \frac{1}{G} \sum_{i=1}^{G} \frac{1}{|o_i|} \sum_{t=1}^{|o_i|}
$$
$$
\left\{ \min\left[ \frac{\pi_\theta(o_{i,t}|q, o_{i,<t})}{\pi_{\theta_{old}}(o_{i,t}|q, o_{i,<t})} \hat{A}_{i,t}, \text{clip}\left( \frac{\pi_\theta(o_{i,t}|q, o_{i,<t})}{\pi_{\theta_{old}}(o_{i,t}|q, o_{i,<t})}, 1 - \epsilon, 1 + \epsilon \right) \hat{A}_{i,t} \right] - \beta \mathbb{D}_{KL}\left[\pi_\theta || \pi_{ref}\right] \right\},
$$

(1)

where advantage $\hat{A}_{i,t} = \frac{r_i - \text{mean}(\mathbf{r})}{\text{std}(\mathbf{r})}$, $G$ is the group size, $\beta$ is the coefficient of KL penalty, and $q = (x, \{y_i\}_{i=1}^{n})$ with prompt $x$ and $n$ generations $y$.

### B.3   HYPERPARAMETERS

We report the detailed hyperparameters for Supervised Fine-Tuning (SFT) in Table 3 and for GRPO in Table 4. We use up to 4*8 H100 80GB GPUs for experiments.

### B.4   TRAINING FRAMEWORKS

We use Xtuner Contributors (2023) to SFT with DeepSpeed Rasley et al. (2020) to accelerate training and ZeRO-3 to reduce memory usage, and verl Sheng et al. (2025) to implement GRPO.

## C   DATASET DETAILS

### C.1   FIN-2024 AND FIN-2025

We consider studying the stock movement prediction task based on data from the Chinese A-share market. Naturally, the collected data are in Chinese, and consequently, the associated prompts and

---

[2]https://anonymous.4open.science/r/RETuning
[3]https://anonymous.4open.science/r/RETuning
[4]https://anonymous.4open.science/r/RETuning

Table 3: The training hyperparameters for Supervised Fine-Tuning (SFT). 32B, 14B, and 7B denote models based on DeepSeek-R1-Distill-Qwen with 32B, 14B, and 7B parameters respectively.

| Hyperparameter Category | 32B | 14B | 7B |
|---|---|---|---|
| **1. Data Configuration** | | | |
| Train Micro-Batch Size per GPU | 1 | 1 | 1 |
| Gradient Accumulation Steps | 128 | 128 | 128 |
| Total Effective Batch Size | 1024 | 1024 | 1024 |
| | $(1 \times 128 \times 8)$ | $(1 \times 128 \times 8)$ | $(1 \times 128 \times 8)$ |
| Pack Sequences to Max Length | False | False | False |
| Data Shuffling Before Packing | True | True | True |
| **2. Model & LoRA Configuration** | | | |
| LLM Torch Dtype | torch.float16 | torch.float16 | torch.float16 |
| LoRA Rank ($r$) | 32 | 32 | 32 |
| LoRA Alpha ($\alpha$) | 64 | 64 | 64 |
| LoRA Dropout | 0.1 | 0.1 | 0.1 |
| LoRA Bias Type | none | none | none |
| LoRA Task Type | CAUSAL_LM | CAUSAL_LM | CAUSAL_LM |
| Variable-Length Attention | False | False | False |
| **3. Optimizer & LR Scheduler** | | | |
| Optimizer Type | torch.optim.AdamW | | |
| Learning Rate (LR) | $2 \times 10^{-4}$ | $2 \times 10^{-4}$ | $2 \times 10^{-4}$ |
| AdamW Betas ($\beta_1$, $\beta_2$) | (0.9, 0.999) | (0.9, 0.999) | (0.9, 0.999) |
| Weight Decay | 0 | 0 | 0 |
| Gradient Clipping Max Norm | 1 | 1 | 1 |
| LR Scheduler Type | Linear Warmup + Cosine Annealing | | |
| Warmup Ratio (Warmup Epochs / Total Epochs) | 0.03 (0.09/3) | 0.03 (0.09/3) | 0.03 (0.09/3) |
| Warmup Start Factor | $1 \times 10^{-5}$ | $1 \times 10^{-5}$ | $1 \times 10^{-5}$ |
| Cosine Annealing Final LR ($\eta_{min}$) | 0.0 | 0.0 | 0.0 |
| **4. Training Strategy & Distributed Config** | | | |
| Training Strategy Type | DeepSpeedStrategy | | |
| DeepSpeed Zero Optimization Stage | 3 | 3 | 3 |
| BF16 Precision Enabled | True | True | True |
| FP16 Precision Enabled | False | False | False |
| Sequence Parallel Size | 8 | 8 | 8 |
| Sampler Shuffling | True | True | True |
| **6. Environment & Misc Config** | | | |
| Launcher Type | pytorch | | |
| Distributed Backend | nccl | nccl | nccl |
| Multiprocessing Start Method | fork | fork | fork |
| Deterministic Training | False | False | False |

Table 4: The training hyperparameters for GRPO. 32B SFT, 14B SFT, and 7B SFT denote models based on DeepSeek R1 with 32B, 14B, and 7B parameters after SFT stage respectively.

| Base Model | 32B SFT | 14B SFT | 7B SFT |
|---|---|---|---|
| **1. Data Configuration** | | | |
| Training Batch Size | 256 | 256 | 256 |
| Validation Batch Size | 256 | 256 | 256 |
| Max Prompt Length | 32768 | 32768 | 32768 |
| Max Response Length | 4096 | 4096 | 4096 |
| Data Shuffling | True | True | True |
| **2. Algorithm Configuration** | | | |
| Advantage Estimator | GRPO | GRPO | GRPO |
| Gamma ($\gamma$, Discount Factor) | 1.0 | 1.0 | 1.0 |
| Lambda ($\lambda$, Advantage Smoothing) | 1.0 | 1.0 | 1.0 |
| KL Coefficient | 0.001 | 0.001 | 0.001 |
| Target KL Divergence | 0.1 | 0.1 | 0.1 |
| Normalize Advantage by Std | True | True | True |
| **3. Actor & Ref Configuration** | | | |
| Learning Rate | $3 \times 10^{-7}$ | $3 \times 10^{-7}$ | $3 \times 10^{-7}$ |
| Weight Decay | 0.01 | 0.01 | 0.01 |
| Clip Ratio | 0.2 | 0.2 | 0.2 |
| Entropy Coefficient | 0.0 | 0.0 | 0.0 |
| PPO Epochs | 1 | 1 | 1 |
| Log Prob Micro Batch Size | 8 | 4 | 4 |
| Tensor Parallel Size | 4 | 4 | 4 |
| **4. Rollout Configuration** | | | |
| Rollout Count (n) | 8 | 8 | 8 |
| Rollout Mode | sync | sync | sync |
| Engine Name | vllm | vllm | vllm |
| Data Type (dtype) | bfloat16 | bfloat16 | bfloat16 |
| Temperature | 0.6 | 0.6 | 0.6 |
| Max Num Batched Tokens | 36864 | 36864 | 36864 |
| Tensor Parallel Size | 4 | 4 | 4 |
| Enable Chunked Prefill | True | True | True |
| **5. Trainer Configuration** | | | |
| Number of Nodes | 4 | 2 | 1 |
| GPUs per Node | 8 | 8 | 8 |
| Total Epochs | 1 | 1 | 1 |
| Checkpoint Save Frequency (steps) | 10 | 10 | 10 |
| Validation Frequency (steps) | 10 | 10 | 10 |
| Validate Before Training | True | True | True |

synthetic data are also in Chinese. This consistency within a single language allows LLMs to achieve better understanding and more coherent reasoning over the data. Thus, the language model can fully leverage its pre-trained knowledge in Chinese to analyze the stock market. Therefore, in this paper, we choose Qwen Bai et al. (2023) and DeepSeek DeepSeek-AI et al. (2025) as the backbone models, both of which are strong Chinese LLMs. We apologize for any inconvenience the language gap may cause to readers who are not native Chinese speakers, and we hope that this gap will not hinder the understanding of our work.

The reason why we decide to construct a new dataset to study the stock movement prediction task is twofold. First, existing datasets (StockNet Xu & Cohen (2018), CMIN-US Luo et al. (2023), CMIN-CN Luo et al. (2023), EDT Zhou et al. (2021)) are outdated and do not reflect the current market conditions. Financial markets are dynamic and constantly evolving, with new trends, regulations, and events shaping the landscape. Using outdated datasets may lead to models that are not well-suited for current market scenarios. Second, existing datasets lack diversity in data sources. Relying solely on price and news data may not capture the full complexity of stock movements. Incorporating additional data sources such as analyst reports, macroeconomic indicators, and quantitative reports can provide a more comprehensive view of the market and improve prediction accuracy. Rich enough data sources are crucial for reliable forecasting in the high–signal-noise ratio of financial markets.

Then, we build a new dataset **Fin-2024** covering January to December 2024, with a test split set **Fin-2024[December]** and an additional long-horizon evaluation set **Fin-2025[June]**. We collect data from multiple sources, including stock prices, financial news, analyst reports, macroeconomic indicators, and quantitative reports. We process the raw data into a structured format suitable for LLMs, including entity recognition, sentiment analysis, event extraction, and traditional time-series analysis (for generating quantitative reports). The dataset contains 209,063 data points across 5,123 A-share stocks from various sectors. Each data point includes a timestamp, stock identifier, historical prices (open, close, high, low, volume), relevant news articles, analyst reports, macroeconomic indicators, quantitative reports, and the corresponding stock movement label (up/down/hold), which is based on the change_pct between the open price of the next trading day and the close price of the current trading day.

In order to ensure data quality, we apply several filtering steps. We remove data points with missing or incomplete information, filter out stocks with low trading volume or insufficient historical data, and balance the label distribution in the dataset to avoid bias towards any particular class. The data processing pipeline is shown in Figure 15, and the prompt length distribution is shown in Figure 16.

The final dataset is split into training (90%) and out-of-distribution (OOD) (10%) sets. The training set is used for model fine-tuning and reinforcement learning, and the OOD set for final evaluation. The long-horizon evaluation set **Fin-2025[June]** contains data from June 2025 to assess model performance in a future market scenario. Please refer to Figure 15 for detailed numbers and splits.

We present an example of the prompt template in Figure 17, which consists of multiple parts: stock news (Figure 18 and Figure 19), stock price information of the current stock and top-3 similar stocks (Figure 20), macroeconomic indicators report (Figure 21), stock fundamentals report (Figure 22 and Figure 23), stock basic information (Figure 24), stock quantitative reports (Figure 25), model response (Figure 26), and model response grading (Figure 27).

## C.2 Fin-2024-SFT

The dataset **Fin-2024-SFT** contains 188 cold-start items and 10K general reasoning samples. Firstly, we selected 10K items from `https://huggingface.co/datasets/GeneralReasoning/GeneralThought-323K`, which aims to avoid catastrophic forgetting Kirkpatrick et al. (2017) and even strengthen reasoning ability when fine-tuning the model. These data points are related to math, code, common sense, chatting, role play, writing, etc. However, they are not allowed to be related to finance to avoid bias, so that we can determine the model's performance in a more controlled setting.

Then, the samples for cold-starting are constructed using a workflow with DeepSeek-R1 (671B) as the backbone model and the polish prompt (Figure 28) to generate high-quality, diverse training samples that follow the proposed thinking schema presented in Section 4.1. These synthesized samples are further filtered by a reward function, which checks format, validates prediction (`<score>`,

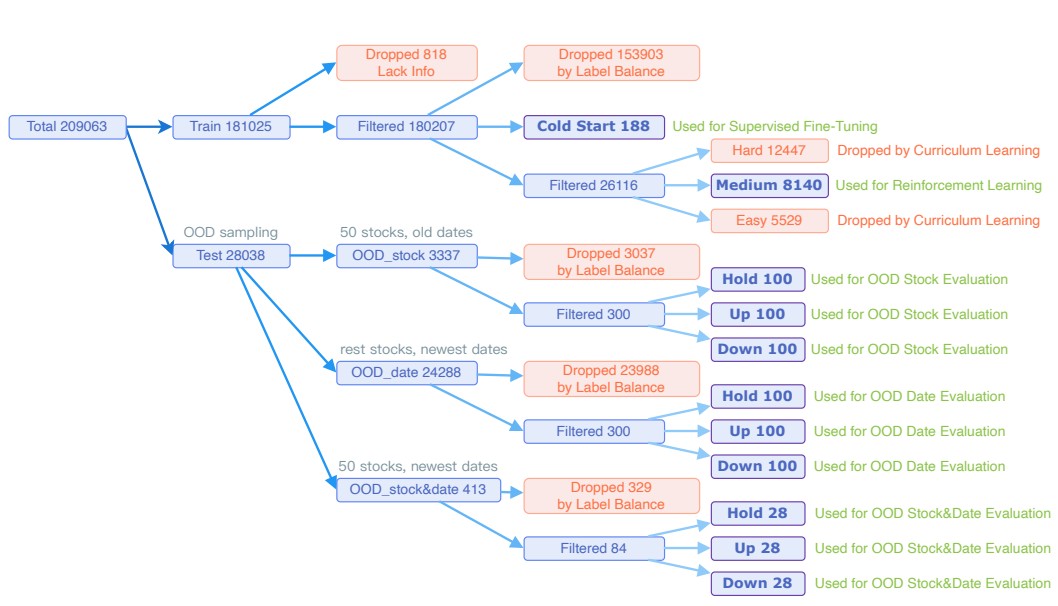

Figure 15: Data processing pipeline showing total 209063 points split into train/test, filtered via multiple steps (lack info, label balance, curriculum learning), and categorized for SFT, RL, OOD evaluations.

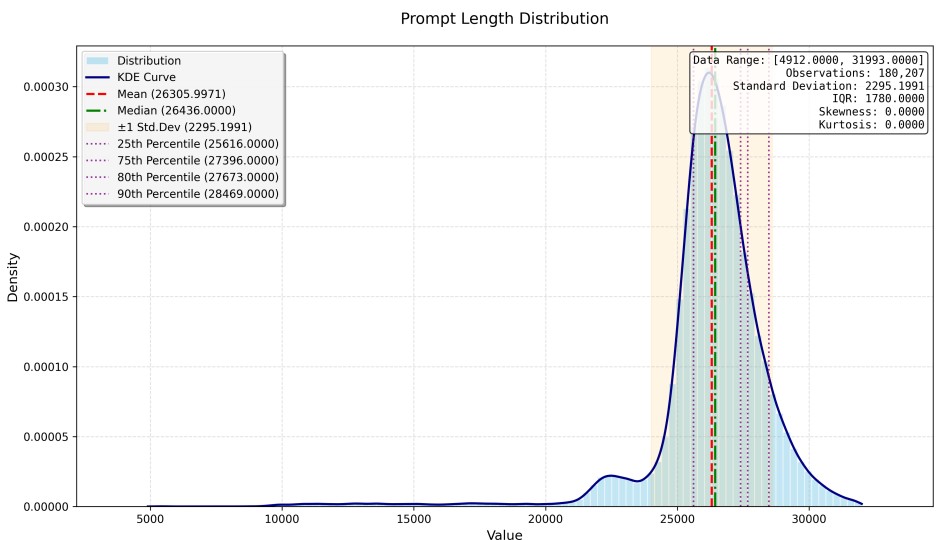

Figure 16: Prompt length distribution across the dataset, illustrating the varying lengths of input prompts used for training and evaluation.

现在要求你预测股价涨跌：给出支持上涨和下跌的证据评分，并预测下一个交易日的开盘价相对于当前交易日的收盘价的涨跌幅，最后给出涨跌分类结果。请先在脑海中思考推理过程，然后向用户提供最终答案。
请回忆一下你在金融领域的知识和经验，结合当前的市场环境，给出一个合理的预测。

你需要考虑：做好这个预测任务所需的关键条件；整理自己的状态；明确自己的分析范式；
为了避免被材料中的情绪误导，请遵循以下思维范式：
1. 理解所要分析的股票的个性，如蓝筹股、成长股、ST股等，不同类型股票的分析方法不同
2. 理解所要预测的时间特征，关注节假日、周末、月末等，不同时间特征的分析方法不同
3. 查看所提供的市场状态包含哪方面的信息，不同的信息覆盖的维度不同，因此分析方法也不同
4. 初步动态构建分析方法逻辑
5. 按分析方法逻辑分析各维度信息
6. 开始整理这些信息，按支持涨的、跌的进行分类，每一类都要对每一条证据进行评分，10分制
7. 进行假设检验，市场模拟，未来推演，反事实假设等，对这些证据进行反思，直到你确信你已经考虑了所有可能的情况。
8. 综合平均这些评分，给出最终支持分数。如 <score>[a, b]</score> 表示支持涨的a分，支持跌的b分；a和b的范围在[0, 10]之间
9. 给出最终的涨跌幅预测和方向预测

预测目标：
1. 证据评分：请给出支持上涨和下跌的证据评分，范围在[0, 10]之间。请在 <score>[a, b]</score> 中填写你的评分，即支持涨的a分，支持跌的b分；a和b的范围在[0, 10]之间。
2. 涨跌幅预测：你需要预测下一个交易日的开盘价相对于当前交易日的收盘价的涨跌幅，范围在[−1, 1]之间，精确到小数点后四位。请在 <pct_change>0.xxxx</pct_change> 中填写你的答案。
3. 方向预测：必须且只能从以下三个选项中选择一个
 – 所预测的涨跌幅大于 3%（显著上涨）：<answer>up</answer>
 – 所预测的涨跌幅小于 −3%（显著下跌）：<answer>down</answer>
 – 所预测的涨跌幅在 −3% 和 3% 之间（震荡）：<answer>hold</answer>

推理过程、证据评分、涨跌幅预测、答案分别包含在<think></think>、<score></score>、<pct_change></pct_change>和<answer></answer>标签中，即<think>此处为推理过程</think><score>此处为证据评分</score><pct_change>此处为涨跌幅预测</pct_change><answer>此处为答案</answer>。

当前环境：

Current Trading Date: {{ example.cur_trading_date }}({{ get_weekday(example.cur_trading_date) }})
Next Trading Date: {{ example.next_trading_date }}({{ get_weekday(example.next_trading_date) }})
Stock: {{ example.code }}

```
// region 基础信息:
{{ example.base_info }}
// endregion

// region 截止到 {{ example.cur_trading_date }} 晚上 9:00 的新闻信息:
{{ example.news }}
// endregion

// region 技术面数据:
{{ example.price }}
// endregion

// region 宏观环境:
{{ example.macro }}
// endregion

// region 基本面数据:
{{ example.fundamental }}
// endregion

{% if example.olhcv_price %}
// region 价格:
{{ example.olhcv_price }}
// endregion
{% endif %}

{% if example.stock_news %}
// region 截止到 {{ example.next_trading_date }} 上午 9:30 开盘的新闻信息:
{% for item in stock_news %}
编号：{{ loop.index0 }}
{{ item }}

{% endfor %}
// endregion
{% endif %}
```

Figure 17: The prompt template for stock movement prediction.

Current Trading Date: 2025-05-30(星期五)
Next Trading Date: 2025-06-03(星期二)
Stock: 000555.SZ

//region 截止到 2025-06-03 上午 9:30 开盘的新闻信息:
编号: 0
Title: 【热点直击】多重利好加持 数字货币含金量还在提升, Content: <p>  5月30日, 数字货币概念股开盘即启动集体大涨模式, 海联金汇、翠微股份竞价涨停, 并最终出现大幅涨停, 雄帝科技、四方精创盘中最大涨幅双双超过10%, 恒宝股份、神州信息、天阳科技、汇金股份、中科金财、金证股份、京北方等概念股涨涨。</p><p>  香港推行稳定币出现重大进展</p><p>  数字货币概念强力启动得到多个利好消息刺激。5月21日, 香港特别行政区立法会通过《稳定币条例草案》, 以在香港设立法币稳定币发行人的发牌制度, 完善对虚拟资产活动在香港的监管框架。以该条例为基础, 香港将正式设立法币稳定币发行人的发牌制度。此前, 京东币链科技于2024年7月入选香港金管局公布的首批"稳定币沙盒"参与者, 京东进入币圈的大动作引起市场高度关注。</p><p>  花旗银行报告认为, 稳定币是去中心化金融的入口, 跟踪稳定币发行增长有助于确定整体数字资产环境健康状况和增长情况。稳定币可以被视为没有原生币固有波动性的价值储存手段, 可用于支付和跨境交易。作为交换媒介, 稳定币支付比例正在增长, 随着监管态度的明确, 支付市场空间有望进一步打开。</p><p>  国海证券认为, 稳定币交易相对欧美传统银行交易有着更高的效率, 尤其是跨境支付方面, 稳定币的转账费用较低, 费用取决于网络情况(以USDT为例, 通常只有几美元), 而一些支付系统的手续费按金额比例收费, 且费率较高。</p><p>  比特币突破11万美元</p><p>  从全球数字货币市场表现看, 近期受资金回暖、机构持续加仓等多重因素影响, 以比特币为代表的加密货币整体呈现上涨趋势。5月22日, 比特币价格一度突破11万美元, 超过今年1月创下的高点。</p><p>  2024年12月5日, 比特币价格有史以来首次突破10万美元, 后在今年1月美国总统特朗普的就职日创下历史新高, 价格突破10.9万美元。但是, 随着特朗普的关税政策影响, 加密货币整体暴跌, 比特币价格在4月初曾短暂跌破8万美元。</p><p>  比特币的强劲表现对包括稳定币在内的数字货币, 产生了正向刺激作用。</p><p>  香港稳定币概念股: </p><p>  众安在线: 去年7月, 众安银行成为香港首家为稳定币发行方提供储备银行服务的数字银行, 并与金管局首批沙盒参与者之一的圆币创新科技合作, 其将成为首家利用众安银行储备银行服务的稳定币发行人。</p><p>  连连数字: 连连数字通过多层全资子公司间接持有连连国际100%股权。连连国际与圆币科技合作稳定币在跨境支付的场景应用项目。</p><p>  渣打集团: 渣打银行(香港)联合安拟集团和香港电讯成立合资公司, 专注于港元稳定币发行, 目前处于沙盒测试阶段, 目标锁定大湾区汇兑成本。</p><p>  京东集团: 京东币链科技已进入沙盒测试第二阶段, 开发与数字人民币衔接的稳定币(JD-HKD), 重点测试跨境支付和供应链金融应用。</p><p>  A股稳定币概念股: </p><p>  海联金汇、翠微股份、雄帝科技、四方精创、恒宝股份、神州信息、天阳科技、汇金股份、中科金财、金证股份、京北方。</p><p>  青岛财经日报/首页新闻记者 荣晓敏</p>

编号: 1
Title: 神州信息王永利: 推动大模型金融应用需解决数据壁垒问题, Content: <p>  新浪科技讯 5月31日午间消息, 近日, 由神州信息主办的"数云原力2025·数智金融论坛"于西安召开。神州信息联席董事长王永利在致辞中表示, AI大模型的广泛应用正成为金融业实现新发展的重大推动力。回顾金融科技的发展脉络, 一次次关键技术的革新, 促使金融行业一步步的持续发展、健康发展。今天, 人工智能作为引领未来的战略性技术, 已成为重塑金融行业核心竞争力的关键要素, 促使行业迈入金融与科技领域深度融合, 积极探索发展新机的关键时点。</p><p>  王永利称, 推动大模型金融应用, 需要着重解决"数据壁垒、投入产出和安全保障"三方面问题。金融科技的高质量发展, 需要全行业生态协同和跨领域的资源整合。实现技术的突破, 以及与场景的深度融合, 需要汇聚政产学研多方智慧, 开放交流共解AI转型难题, 以生态合力加速技术普惠。</p><p>  论坛现场, 针对金融安全标准、金融大模型联合研发、金融大模型产业生态、国产云金融新核心等内容, 神州信息与产业生态伙伴发布系列成果。神州信息正式发布"乾坤"企业级数智底座白皮书。作为行业首个基于"数云融合"理念打造的企业级数智底座, "乾坤"平台具有"云原生、数字原生、AI原生"三大技术特点, 可针对"企业级开发平台、企业级工艺及架构治理、企业级云原生平台和企业级AI平台"四大典型建设场景, 提供一站式解决方案。</p><p>  此外, 神州数码宣布自研的神州问学由AI原生赋能平台升级为企业级Agent中台, 为AI规模化落地提供全栈解决方案; 神州信息与青岛银行成立金融科技大模型联合实验室; 由神州信息牵头, 金融国际标准——"金融机构信息安全控制措施指南标准"研究 正式启动。</p><p>  同时, 神州信息与华为、腾讯云、阿里云、中科海光和昆仑芯等行业科技伙伴举行多项成果发布: 神州信息与华为正式对外发布"金融知识问答"联合智能体; 神州信息与腾讯云签署大数据产品合作框架协议; 神州信息发起行业首个国产云金融核心系统联盟; 神州信息发起成立行业首个AIGC大模型金融生态体系。</p>

编号: 2
Title: 数云原力2025·数智金融论坛在西安召开, Content: <p>  上证报中国证券网讯(记者 张问之)5月29日, 由神州信息主办的"数云原力2025·数智金融论坛"于西安召开。论坛以"信启原力·智启未来"为主题, 围绕金融AI应用、金融场景创新、金融生态协同等数智化发展关键问题展开交流。</p> <p>  本次论坛邀请全国近200家商业银行, 300多位科技和相关业务负责人参会。来自工商银行、民生银行、恒丰银行、北京银行、陕西农信等机构负责人分享数智化转型建设经验。多家银行相关负责人围绕"金融大模型应用"和"银行转型新机遇"等话题展开讨论。</p> <p>  陕西省决策咨询委员会委员、省委金融办一级巡视员张春明</p><p>  陕西省决策咨询委员会委员、省委金融办一级巡视员张春明在致辞中表示, 陕西依托科教优势, 紧扣中央金融工作会议关于数字金融的指导思想, 在数智金融领域持续探索创新, 形成了一系列特色实践与显著成果。通过深化科技赋能应用、优化金融服务模式, 在缓解中小企业融资难、助力乡村振兴、绿色发展等重点领域积极作为, 为全省经济高质量发展注入了强劲动力。陕西省委金融办、金融工委将一如既往地高度重视数智金融发展, 持续完善政策支持体系, 加强数字金融基础设施建设, 为各类金融机构和企业在陕发展提供全方位、高质量的服务。陕西将以更加开放的格局, 深化与国内外金融机构、科研院所及企业的协同合作, 汇聚全球资源与创新智慧, 携手打造具有国际竞争力的数智金融产业集群, 抢占数字经济时代的产业制高点。</p> <p>  神州信息联席董事长王永利</p><p>  神州信息联席董事长王永利在致辞中表示, AI大模型的广泛应用正成为金融业实现新发展的重大推动力。回顾金融科技的发展脉络, 一次次关键技术的革新, 促使金融行业一步步的持续发展、健康发展。今天, 人工智能作为引领未来的战略性技术, 已成为重塑金融行业核心竞争力的关键要素, 促使行业迈入金融与科技领域深度融合, 积极探索发展新机的关键时点。推动大模型金融应用, 需要重解决"数据壁垒、投入产出和安全保障"三方面问题。</p> <p>  金融科技的高质量发展, 需要全行业生态协同和跨领域的资源整合。实现技术的突破, 以及与场景的深度融合, 需要汇聚政产学研多方智慧, 开放交流共解AI转型难题, 以生态合力加速技术普惠。"神州信息愿以本次论坛为契机, 以AI技术为笔, 以数据要素为墨, 共同书写金融数字化转型的新篇章, 共筑行业发展的新生态。"王永利表示。</p> <p>  论坛同期, 神州信息与中国金融标准研究院、中国信通院、华为、腾讯云、阿里云、中科海光、昆仑芯等产业侧相关机构达成合作, 并发布"金融安全标准、区域银行新核心建设、全栈国产云金融核心、AIGC金融生态体系、大模型金融一体机、智能大数据合作"等多项行业重磅合作成果。</p>

Figure 18: Example. Part 1.1: Stock News.

编号：3

Title: 神州信息发布"乾坤"企业级数智底座白皮书, Content: <p>  上证报中国证券网讯（记者 张问之）近日，在神州信息主办的"数云原力2025·数智金融论坛"现场，针对金融安全标准、金融大模型联合研发、金融大模型产业生态、国产云金融新核心等内容，神州信息与产业生态伙伴发布系列成果。</p> <p>  其中，神州信息正式发布"乾坤"企业级数智底座白皮书。作为行业首个基于"数云融合"理念打造的企业级数智底座，"乾坤"平台具有"云原生、数字原生、AI原生"三大技术特点，可针对"企业级开发平台、企业级工艺及架构治理、企业级云原生平台和企业级AI平台"四大典型建设场景，提供一站式解决方案。</p> <p>  发布现场，神州信息副总裁徐启昌表示，神州信息"乾坤"数智化底座可以帮助金融机构实现"基础设施资源利用率提升一倍、研发效率提升30%至50%、数据利用率提升60%以上、AI模型迭代速度提升10倍以上"。"乾坤"数智化底座，已经成为金融企业的新质生产力平台。</p> <p>  神州数码副总裁、CTO李刚正式宣布神州数码自研的神州问学由AI原生赋能平台升级为企业级Agent中台，为AI规模化落地提供全栈解决方案。</p> <p>  同时，神州信息与青岛银行成立金融科技大模型联合实验室。论坛现场，青岛银行信息技术部总经理韩朝丽与神州信息大行BU总经理李拥军代表双方共同签署合作协议。联合实验室的成立，将成为产业协同合作的标杆。通过技术与业务深度融合，双方将共同加速银行业大模型技术的场景化创新与落地实践。同时，实验室正式启动首个重点研究课题——对公多智能体业务助手研发，将为银行对公业务带来全新的智能化服务体验。</p> <p>  由神州信息牵头，金融国际标准——"金融机构信息安全控制措施指南标准"研究也正式启动。神州信息与邮储银行、兴业银行、中国银联、国泰海通、北京国家金融标准化研究院、中金金融认证中心、新华三集团等单位共同启动该标准研制。该标准研究将围绕金融服务机构在内部运营和外部企业交易中的特殊需求，针对金融服务领域的流程控制系统，提供基于信息安全管理的实施指导规范，旨在将ISO/IEC 27002:2022的标准内容扩展应用到金融服务领域的流程控制系统以及自动化技术的使用范围。</p>

编号：4

Title: 神州信息与华为发布"金融知识问答"联合智能体, Content: <p>  上证报中国证券网讯（记者 张问之）近日，在神州信息主办的"数云原力2025·数智金融论坛"现场，神州信息与华为、腾讯云、阿里云、中科海光和昆仑芯等行业科技伙伴举行多项成果发布。</p> <p>  神州信息与华为正式对外发布"金融知识问答"联合智能体。以神州信息FinancialMaster知识问答智能体为核心应用，基于华为昇腾一体机，金融知识问答联合智能体可为金融机构提供高效的资料查询与信息提炼能力，有效改善用户体验。</p> <p>  神州信息与腾讯云签署大数据产品合作框架协议。面对金融云转型建设趋势和金融大模型应用热潮，金融行业对国产数据库应用需求极大提升。双方将加大在国产分布式数据库及相关数据产品的市场推广力度。通过联合推广，联合技术合作等方式，实现双方产业生态的深度发展，满足金融机构各类大数据建设需求。</p> <p>  同时，神州信息发起行业首个国产云金融核心系统联盟。面对金融行业新一代核心系统建设，为了满足金融机构对更高效、稳定、安全的核心应用系统建设需求，神州信息与华为云、腾讯云和阿里云，联合成立国产云金融核心联盟，通过新一代云原生金融核心+全栈国产云，为金融机构数智化夯实数字底座。</p> <p>  神州信息亦发起成立行业首个AIGC大模型金融生态体系，旨在面对金融数智化转型热潮，为金融机构提供低成本、高效率、高安全和高可用的金融大模型建设服务。神州信息与华为、腾讯云、阿里云、中科海光、昆仑芯合作，围绕核心应用、国产云等多层面，构建从基础硬件到上层应用软件，全层面的AIGC生态体系。</p>

编号：5

Title: 神州信息发起成立行业首个AIGC大模型金融生态体系, Content: <p>  本报讯（记者桂小笋）6月2日，《证券日报》记者从神州数码信息服务集团股份有限公司(以下简称"神州信息")处获悉，在日前由神州信息主办的"数云原力2025.数智金融论坛"中，神州信息正式对外发布"乾坤"企业级数智底座白皮书。作为行业首个基于"数云融合"理念打造的企业级数智底座，"乾坤"平台具有"云原生、数字原生、AI原生"三大技术特点，可针对"企业级开发平台、企业级工艺及架构治理、企业级云原生平台和企业级AI平台"四大典型建设场景，提供一站式解决方案。</p> <p>  神州信息副总裁徐启昌表示，神州信息"乾坤"数智化底座可以帮助金融机构实现"基础设施资源利用率提升一倍、研发效率提升30%至50%、数据利用率提升60%以上、AI模型迭代速度提升10倍以上"。"乾坤"数智化底座，已经成为金融企业的新质生产力平台。</p><p>  神州信息联席董事长王永利表示："AI大模型的广泛应用正成为金融业实现新发展的重大推动力。推动大模型金融应用，需要着重解决'数据壁垒、投入产出和安全保障'三方面问题。"</p><p>  王永利分析，金融科技的高质量发展，需要全行业生态协同和跨领域的资源整合。实现技术的突破，以及与场景的深度融合，需要汇聚政产学研多方智慧，开放交流共解AI转型难题，以生态合力加速技术普惠。</p><p>  与此同时，神州信息还发起行业首个国产云金融核心系统联盟。面对金融行业新一代核心系统建设，为了满足金融机构对更高效、稳定、安全的核心应用系统建设需求，神州信息与华为云、腾讯云和阿里云，联合成立行业首个国产云金融核心联盟，通过新一代云原生金融核心+全栈国产云，为金融机构数智化夯实数字底座。同时，神州信息发起成立行业首个AIGC大模型金融生态体系。旨在面对金融数智化转型热潮，为金融机构提供低成本、高效率、高安全和高可用的金融大模型建设服务。神州信息与华为、腾讯云、阿里云、中科海光、昆仑芯合作，围绕核心应用、国产云、国产芯等多层面，构建从基础硬件到上层应用软件，全层面的AIGC生态体系。</p>

// endregion

//region 截止到 2025–05–30 晚上 9:00 的新闻信息:
2025–05–30日新闻: 2025–05–30每日信息总结:
实时新闻: <神州信息在5月29日获得融资流入6044.86万元，占当日流入资金比例为12.65%。当前融资余额为5.15亿元，占流通市值的4.39%，超过历史60%分位水平。融券方面，5月29日融券偿还0股，融券卖出1.95万股，融券余额为174.75万元，低于历史50%分位水平。
分析师观点: <融资买入的增加表明市场对神州信息的信心有所提升，投资者情绪偏向买方，可能反映出对公司未来发展的乐观预期。尽管融券卖出量较小，但融券余额的下降显示出市场对该股的谨慎态度。>/n2025–05–31日新闻: 2025–05–31每日信息总结:
实时新闻:
1. 神州信息在"数云原力2025·数智金融论坛"上发布了"乾坤"企业级数智底座白皮书，强调其在金融机构基础设施和研发效率方面的提升。
2. 神州信息与青岛银行成立金融科技大模型联合实验室，旨在推动银行业大模型技术的应用。
3. 神州信息牵头启动金融国际标准"金融机构信息安全控制措施指南标准"的研究，涉及多家金融机构的合作。
4. 数字货币概念股集体上涨，神州信息作为其中一员，受益于香港稳定币政策的推进。
5. 神州信息与华为等科技伙伴发布"金融知识问答"联合智能体，提升金融机构的信息查询能力。

分析师观点:
1. AI大模型的应用被视为金融行业发展的重要推动力，需解决数据壁垒等问题以实现高质量发展。
2. 数字货币市场的积极动态和政策支持可能进一步提升相关概念股的市场表现，神州信息在此领域的布局将增强其竞争力。
// endregion

Figure 19: Example. Part 1.2: Stock News.

```
//region 价格:
### 000555.SZ price infos
|index|date|open|high|low|close|volume|turnover_rate|pct_change|lowerband|middleband|upperband|
|----|----|----|----|----|----|----|----|----|----|----|----|
|77|2025-04-30|10.6500|10.8200|10.5800|10.7000|13307055.0000|1.3688|0.0113|9.9833|10.7405|11.4977|
|78|2025-05-06|10.8800|11.2800|10.8600|11.2700|25991423.0000|2.8792|0.0533|10.0506|10.7195|11.3884|
|79|2025-05-07|11.4500|11.5200|11.0400|11.1300|25053400.0000|2.5770|-0.0124|10.0850|10.7055|11.3260|
|80|2025-05-08|11.1000|11.3800|11.0900|11.3500|21417050.0000|2.2030|0.0198|10.1126|10.7595|11.4064|
|81|2025-05-09|11.3500|11.5200|11.1600|11.2200|23180221.0000|2.3843|-0.0115|10.2202|10.8165|11.4128|
|82|2025-05-12|11.3200|11.4100|11.2300|11.3700|19581446.0000|2.0142|0.0134|10.2767|10.8700|11.4633|
|83|2025-05-13|11.4900|11.5400|11.1800|11.2300|18089400.0000|1.8607|-0.0123|10.3005|10.9010|11.5015|
|84|2025-05-14|11.2000|11.4200|11.1100|11.3100|17626700.0000|1.8131|0.0071|10.3064|10.9285|11.5506|
|85|2025-05-15|11.2500|11.3100|10.9600|10.9900|16474300.0000|1.6946|-0.0283|10.3132|10.9350|11.5568|
|86|2025-05-16|10.9200|11.0700|10.8900|10.9500|10325300.0000|1.0621|-0.0036|10.3301|10.9455|11.5609|
|87|2025-05-19|11.0100|11.2200|10.8500|11.1100|16296246.0000|1.6762|0.0146|10.3923|10.9760|11.5597|
|88|2025-05-20|11.1100|11.4200|11.0000|11.2400|18187500.0000|1.8708|0.0117|10.4795|11.0165|11.5535|
|89|2025-05-21|11.2400|11.2400|11.0400|11.1600|12196250.0000|1.2545|-0.0071|10.5595|11.0485|11.5375|
|90|2025-05-22|11.0900|11.1800|10.9200|10.9200|13373250.0000|1.3756|-0.0215|10.5639|11.0505|11.5371|
|91|2025-05-23|11.0200|11.5500|10.9600|11.0400|43082619.0000|4.4315|0.0110|10.5607|11.0455|11.5303|
|92|2025-05-26|10.9300|11.1800|10.9300|11.1000|23766389.0000|2.4446|0.0054|10.5598|11.0430|11.5262|
|93|2025-05-27|11.1400|11.1500|10.9100|11.0700|16139100.0000|1.6601|-0.0027|10.5880|11.0570|11.5260|
|94|2025-05-28|11.0200|11.1200|10.9200|10.9600|14931917.0000|1.5359|-0.0099|10.5994|11.0620|11.5246|
|95|2025-05-29|10.9600|12.0600|10.9200|12.0600|69115467.0000|7.1093|0.1004|10.5587|11.1380|11.7173|
|96|2025-05-30|12.7900|12.8600|12.0100|12.0900|103139544.0000|10.6090|0.0025|10.5564|11.2135|11.8706|

### 300079.SZ price infos (rank 1 similar to 000555.SZ)
|index|date|open|high|low|close|volume|turnover_rate|pct_change|lowerband|middleband|upperband|
|----|----|----|----|----|----|----|----|----|----|----|----|
|77|2025-04-30|5.2400|5.3400|5.2200|5.2600|38795267.0000|3.0274|0.0057|4.4328|5.0760|5.7192|
|78|2025-05-06|5.3200|5.4900|5.3100|5.4900|57199858.0000|4.4636|0.0437|4.4271|5.0805|5.7339|
|79|2025-05-07|5.5500|5.6000|5.3700|5.4300|58119984.0000|4.5354|-0.0109|4.4247|5.0830|5.7413|
|80|2025-05-08|5.4200|5.4900|5.3900|5.4700|39975485.0000|3.1195|0.0074|4.5629|5.1405|5.7181|
|81|2025-05-09|5.4500|5.4600|5.3100|5.3400|33286401.0000|2.5975|-0.0238|4.6617|5.1795|5.6973|
|82|2025-05-12|5.3800|5.4100|5.3400|5.4000|26762700.0000|2.0884|0.0112|4.7279|5.2125|5.6971|
|83|2025-05-13|5.4800|5.5100|5.3400|5.3600|28349215.0000|2.2122|-0.0074|4.7730|5.2365|5.7000|
|84|2025-05-14|5.3300|5.4300|5.3000|5.3900|30913111.0000|2.4123|0.0056|4.8200|5.2610|5.7020|
|85|2025-05-15|5.3600|5.3800|5.2300|5.2500|24995500.0000|1.9505|-0.0260|4.8526|5.2745|5.6964|
|86|2025-05-16|5.2400|5.2900|5.2000|5.2500|17641898.0000|1.3767|0.0000|4.8860|5.2875|5.6890|
|87|2025-05-19|5.2400|5.3000|5.1700|5.2800|21042500.0000|1.6420|0.0057|4.9491|5.3070|5.6649|
|88|2025-05-20|5.2700|5.3800|5.2200|5.3800|25037225.0000|1.9538|0.0189|5.0248|5.3310|5.6372|
|89|2025-05-21|5.4400|5.5300|5.3400|5.3900|56088305.0000|4.3768|0.0019|5.0974|5.3525|5.6076|
|90|2025-05-22|5.3600|5.4200|5.2600|5.2600|35309931.0000|2.7554|-0.0241|5.1233|5.3595|5.5957|
|91|2025-05-23|5.2400|5.4000|5.1600|5.1600|35569924.0000|2.7757|-0.0190|5.0995|5.3510|5.6025|
|92|2025-05-26|5.1400|5.2500|5.1400|5.2400|23263219.0000|1.8153|0.0155|5.1486|5.3255|5.5024|
|93|2025-05-27|5.2700|5.2700|5.1700|5.2400|19213460.0000|1.4993|0.0000|5.1399|5.3180|5.4961|
|94|2025-05-28|5.2500|5.2800|5.1900|5.2300|18887298.0000|1.4739|-0.0019|5.1319|5.3140|5.4961|
|95|2025-05-29|5.2300|5.4900|5.2100|5.4900|60926160.0000|4.7544|0.0497|5.1339|5.3270|5.5201|
|96|2025-05-30|5.4500|5.5300|5.3400|5.3800|38312377.0000|2.9897|-0.0200|5.1455|5.3345|5.5235|

### 300479.SZ price infos (rank 2 similar to 000555.SZ)
|index|date|open|high|low|close|volume|turnover_rate|pct_change|lowerband|middleband|upperband|
|----|----|----|----|----|----|----|----|----|----|----|----|
|77|2025-04-30|19.3100|19.8500|19.1000|19.6600|6985000.0000|3.5470|0.0299|17.6005|19.3575|21.1145|
|78|2025-05-06|19.9900|20.2500|19.8100|20.1400|7816845.0000|3.9694|0.0244|17.6925|19.3115|20.9305|
|79|2025-05-07|20.4000|20.5500|19.9300|20.1500|7743299.0000|3.9320|0.0005|17.7607|19.2780|20.7953|
|80|2025-05-08|19.9800|20.4800|19.9800|20.4500|6243600.0000|3.1705|0.0149|18.2142|19.4455|20.6768|
|81|2025-05-09|20.3500|20.3600|19.9200|19.9600|5139300.0000|2.6097|-0.0240|18.6053|19.5585|20.5117|
|82|2025-05-12|20.1500|20.3700|20.0400|20.3300|4720400.0000|2.3970|0.0185|18.7457|19.6455|20.5453|
|83|2025-05-13|20.6600|20.6700|19.9000|19.9500|5478600.0000|2.7820|-0.0187|18.8154|19.6880|20.5606|
|84|2025-05-14|19.9600|20.3300|19.7000|20.1600|6406300.0000|3.2531|0.0105|18.8529|19.7305|20.6081|
|85|2025-05-15|20.3300|20.3500|19.3500|19.3800|6546000.0000|3.3241|-0.0387|18.8361|19.7230|20.6099|
|86|2025-05-16|19.3800|19.5800|19.1300|19.4000|4054599.0000|2.0589|0.0010|18.8337|19.7220|20.6103|
|87|2025-05-19|19.4000|19.7000|19.1200|19.6500|4040400.0000|2.0517|0.0129|18.8463|19.7295|20.6127|
|88|2025-05-20|19.5900|19.9800|19.3500|19.9100|4695700.0000|2.3845|0.0132|18.8979|19.7605|20.6231|
|89|2025-05-21|19.8100|20.5200|19.4500|20.0100|11138149.0000|5.6559|0.0050|18.9740|19.8015|20.6290|
|90|2025-05-22|19.9500|20.4300|19.6300|20.3600|11830800.0000|6.0077|0.0175|18.9654|19.8280|20.6906|
|91|2025-05-23|20.2400|20.2600|19.3000|19.3600|11955500.0000|6.0710|-0.0491|18.9142|19.7975|20.6808|
|92|2025-05-26|19.5900|20.4800|19.4500|19.6200|9397557.0000|4.7721|0.0134|18.8996|19.7715|20.6434|
|93|2025-05-27|19.4500|19.6800|19.1300|19.2000|7293700.0000|3.7037|-0.0214|18.8587|19.7575|20.6563|
|94|2025-05-28|19.2500|19.6200|19.0400|19.1900|6843854.0000|3.4753|-0.0005|18.8058|19.7365|20.6672|
|95|2025-05-29|19.2500|20.3800|19.1100|20.3400|13791300.0000|7.0032|0.0599|18.9650|19.8155|20.6660|
|96|2025-05-30|20.2100|20.5800|19.9600|20.0900|10359145.0000|5.2604|-0.0123|19.0761|19.8655|20.6549|

### 300541.SZ price infos (rank 3 similar to 000555.SZ)
|index|date|open|high|low|close|volume|turnover_rate|pct_change|lowerband|middleband|upperband|
|----|----|----|----|----|----|----|----|----|----|----|----|
|77|2025-04-30|13.1000|13.6000|12.9700|13.4000|48074617.0000|12.4359|0.0276|10.7158|12.1235|13.5312|
|78|2025-05-06|13.6900|14.2600|13.5300|14.2400|64478106.0000|16.6792|0.0627|10.5460|12.1800|13.8140|
|79|2025-05-07|14.3000|14.3800|13.6500|13.8700|57774792.0000|14.9452|-0.0260|10.4577|12.2340|14.0103|
|80|2025-05-08|13.8100|14.0200|13.7900|13.8600|34989228.0000|9.0510|-0.0007|10.6733|12.4010|14.1287|
|81|2025-05-09|13.8000|14.2500|13.6800|13.7800|40699099.0000|10.5280|-0.0058|10.8431|12.5410|14.2389|
|82|2025-05-12|13.9500|14.1000|13.7400|14.0200|30871359.0000|7.9858|0.0174|10.9308|12.6690|14.4072|
|83|2025-05-13|14.1300|14.2800|13.8200|13.8500|27313896.0000|7.0655|-0.0121|10.9834|12.7630|14.5426|
|84|2025-05-14|14.0000|14.2800|13.8700|14.2600|40205397.0000|10.4003|0.0296|11.0265|12.8785|14.7305|
|85|2025-05-15|14.0800|14.1100|13.5800|13.6800|32079108.0000|8.2982|-0.0407|11.1103|12.9575|14.8047|
|86|2025-05-16|13.6000|13.8200|13.4900|13.6900|18074531.0000|4.6755|0.0007|11.2250|13.0425|14.8600|
|87|2025-05-19|13.6500|13.8200|13.5500|13.7400|15513988.0000|4.0132|0.0037|11.4001|13.1415|14.8829|
|88|2025-05-20|13.6600|13.7500|13.4100|13.6200|19220998.0000|4.9721|-0.0087|11.6418|13.2425|14.8432|
|89|2025-05-21|13.7000|14.1300|13.5500|13.8800|35338270.0000|9.1413|0.0191|11.8765|13.3480|14.8195|
|90|2025-05-22|13.7100|13.8700|13.4300|13.4700|25255268.0000|6.5330|-0.0295|12.0608|13.4165|14.7722|
|91|2025-05-23|13.4000|13.5200|12.9300|12.9300|26857256.0000|6.9474|-0.0401|12.1381|13.4405|14.7429|
|92|2025-05-26|12.9900|13.2500|12.9500|13.2400|15712590.0000|4.0645|0.0240|12.2439|13.4775|14.7111|
|93|2025-05-27|13.2200|13.2800|13.0600|13.1600|11762310.0000|3.0427|-0.0060|12.4804|13.5340|14.5876|
|94|2025-05-28|13.2200|13.3900|13.0600|13.1700|16441706.0000|4.2531|0.0008|12.8171|13.5960|14.3749|
|95|2025-05-29|13.1700|14.2600|13.1300|14.2600|62903024.0000|16.2717|0.0828|12.8750|13.6580|14.4410|
|96|2025-05-30|13.8100|14.0300|13.7100|13.8600|40900299.0000|10.5801|-0.0281|12.9654|13.6990|14.4326|
// endregion
```

Figure 20: Example. Part 2: Stock Price Info of Current Stock and Top-3 Similar Stocks.

//region 宏观环境:
# A股市场分析报告

## 1. 市场态势分析

### 资金流向与强度分析
近期A股市场表现出资金流入的积极态势，尤其是医疗健康和科技板块。根据龙虎榜数据，资金净流入前十的板块中，仿制药一致性评价和创新药板块分别吸引了4.33亿元和1.77亿元的资金流入，显示出市场对医药行业的强烈关注。

### 市场参与主体行为特征
从龙虎榜数据来看，游资和机构的交易行为活跃，尤其是在医疗和科技领域。游资在多个个股中表现出明显的买入倾向，尤其是华森制药（002907.SZ），其连续涨停的表现吸引了大量资金关注。

### 市场情绪指标与波动特征
市场情绪指标显示出乐观情绪，尤其是在医疗健康板块。根据市场情绪指数，当前市场情绪处于高位，反映出投资者对短期市场的信心增强。

### 近期市场主要矛盾和焦点
当前市场的主要矛盾集中在经济复苏与外部环境的不确定性之间。尽管国内政策支持力度加大，但全球经济形势和中美贸易关系仍然是市场关注的焦点。

## 2. 板块轮动分析

### 强势板块梳理与未来持续性判断
近期表现强势的板块包括仿制药一致性评价、创新药和流感板块。仿制药一致性评价板块在过去三日内涨幅达到1.16%，资金净流入4.33亿元，显示出其未来的持续性。

### 资金布局方向和规模
资金布局主要集中在医疗健康和科技板块，尤其是仿制药和创新药。资金流入的规模和强度表明市场对这些板块的信心。

### 板块轮动节奏与特征
板块轮动节奏较快，医疗健康板块在短期内表现突出，而科技板块则在中长期内有望继续吸引资金。市场情绪的变化可能导致板块间的快速切换。

### 龙头个股表现分析
华森制药（002907.SZ）作为仿制药一致性评价板块的龙头个股，近期表现强劲，连续涨停，吸引了大量资金流入。其未来表现值得关注。

## 3. 市场机会与风险分析

### 短期市场机会识别
短期内，医疗健康板块仍然是市场的主要机会，尤其是仿制药和创新药相关个股。建议投资者关注相关龙头个股的表现。

### 潜在风险因素预警
潜在风险主要来自于外部经济环境的不确定性，尤其是中美贸易关系的变化可能对市场情绪产生影响。此外，市场情绪的波动也可能导致短期内的剧烈调整。

### 交易时机与规模建议
建议投资者在医疗健康板块的回调时适度加仓，关注龙头个股的表现。交易规模应根据市场情绪和资金流向进行动态调整。

### 止盈止损参考位
建议设定止盈位在近期高点附近，止损位则可设定在个股支撑位下方，以控制风险。

## 4. 外围市场联动性分析

### 全球主要市场表现
近期全球主要市场表现分化，欧美市场在经济数据改善的背景下有所反弹，但仍面临通胀压力和利率上升的挑战。

### 外围市场风险溢出效应
美国加息预期和通胀压力可能对全球市场产生溢出效应，影响A股市场的资金流动和投资者情绪。

### 全球流动性状况
全球流动性仍然宽松，但美联储的加息政策可能导致流动性收紧，需密切关注其对市场的影响。

### 重要经济体政策动向
美国的减税政策和关税政策仍在不断调整，可能对全球经济和市场产生深远影响。投资者需关注相关政策的变化及其对市场的影响。

———

本报告基于多维度数据源的分析，旨在为短线交易决策提供参考。建议投资者密切关注市场动态，灵活调整投资策略。
// endregion

Figure 21: Example. Part 3: Macroeconomic Indicators Report.

//region 基本面数据:
### 一、公司概述

1. **所属行业与主营业务**
   – 神州信息（股票代码：000555.SZ）主要从事技术服务、农业信息化、应用软件开发、金融专用设备相关业务及集成解决方案。公司在量子科技、数字货币、网络安全等多个领域具有业务布局。

2. **行业地位**
   – 神州信息在量子通信、数字货币及网络安全等领域处于行业前列，尤其在数字金融和信息安全方面，积极参与国家信息安全战略，具备较强的市场竞争力。

3. **业务特点**
   – 公司业务涵盖多个高科技领域，尤其在量子通信和数字货币方面具有较强的技术积累和市场应用。与华为的合作进一步增强了其在金融科技和物联网领域的竞争力。

### 二、最新经营态势

1. **收入利润分析**
   – 截至2024年9月30日，神州信息的营业总收入为66.80亿元，较上年同期的66.18亿元略有增长（0.94%）。然而，净利润为–1.80亿元，显示出公司在盈利能力方面面临较大压力。

2. **盈利能力分析**
   – 销售毛利率为13.61%，销售净利率为–2.70%，表明公司在成本控制和盈利能力方面存在问题。营业总成本占营业总收入的比例高达103.83%，显示出公司在运营效率上存在较大挑战。

3. **经营效率分析**
   – 经营活动产生的现金流量净额为–7.45亿元，表明公司在经营活动中现金流出大于流入，反映出经营效率低下。应收账款周转率为2.67次，存货周转率为2.36次，显示出公司在资产管理方面的效率较低。

4. **变动趋势分析**
   – 归属母公司股东的净利润同比下降236.41%，显示出公司在盈利能力方面的显著恶化。预计2024年将出现净利润亏损，主要由于行业客户数字化转型进程放缓及市场竞争加剧。

### 三、财务状况评估

1. **资产负债分析**
   – 截至2024年9月30日，公司的总资产为126.69亿元，负债合计为65.90亿元，资产负债率为52.02%。股东权益合计为60.79亿元，表明公司在财务结构上相对稳健，但负债水平仍需关注。

2. **现金流分析**
   – 经营活动产生的现金流量净额为–7.45亿元，显示出公司在现金流管理方面的压力。尽管销售商品、提供劳务收到的现金为78.45亿元，但经营活动现金流出大于流入，反映出公司在运营中的现金流问题。

3. **资本结构分析**
   – 公司的流动比率为1.53，速动比率为1.01，表明公司在短期偿债能力上相对良好。然而，保守速动比率仅为0.65，显示出在流动性管理上存在一定风险。

4. **风险指标分析**
   – 公司的已获利息倍数为–6.94，表明公司在偿还利息方面面临较大压力，可能影响未来的融资能力和财务稳定性。

### 四、估值水平分析

1. **当前估值水平**
   – 神州信息的市盈率为–67.6942，市净率为1.7637，表明市场对公司未来盈利能力的预期较低。每股净资产为6.1858元，反映出公司在资产方面的相对价值。

2. **估值合理性分析**
   – 由于公司预计在2024年出现净利润亏损，当前的估值水平可能反映出市场对其未来盈利能力的悲观预期。市净率相对较高，可能意味着市场对公司资产的价值认可度不足。

### 五、综合结论

1. **主要经营优势**
   – 神州信息在量子科技、数字货币和网络安全等领域具有技术积累和市场应用优势，尤其与华为的合作为其提供了强大的市场支持。

2. **潜在风险提示**
   – 公司面临的主要风险包括市场竞争加剧、客户数字化转型进程放缓、现金流管理不善及高负债水平等，这些因素可能影响公司的持续盈利能力和财务稳定性。

3. **基本面趋势判断**
   – 由于公司预计在2024年将出现净利润亏损，且经营效率和现金流状况不佳，基本面趋势显示出一定的下行压力。未来需关注公司在业务结构转型和市场开拓方面的进展，以期改善经营状况。

Figure 22: Example. Part 4.1: Stock Fundamentals Report.

最新估值：神州信息(股票代码：000555.SZ)最新财务估值信息：

企业估值：
```
    当前，公司总股本为9.76亿，发行总股本为0.74亿，流通股本为9.72亿，每股净资产为5.6886，ps市销率为1.366，pcf市现率为−1.144，市净率为2.1253，市盈率为−31.7323。
```

截至报告期2025−03−31，公司财务分析指标如下：
```
{'归属母公司股东的净利润/报告期期末总股本': '−0.10元/股', '归属于普通股股东的扣除非经常性损益后的净利': '−0.10元/股', '股东权益周转率': '0.39%', '人力投入回报率rop': '−14.76%', '每股营业总收入': '2.21元/股', '每股营业收入': '2.21元/股', '每股资本公积': '2.39元/股', '每股盈余公积': '0.05元/股', '每股未分配利润': '2.32元/股', '每股留存收益': '2.37元/股', '每股现金流量净额': '−0.48/股', '每股息税前利润': '−0.13元/股', '每股企业自由现金流量': '0.15元/股', '每股股东自由现金流量': '2.39元/股', '平均净资产收益率': '−1.66%', '扣除非经常损益后的平均净资产收益率': '−1.79%', '净资产收益率roe—增发条件': '−1.79%', '总资产净利率roa': '−0.89%', '总资产报酬率roa': '−1.03%', '投入资本回报率roic': '−1.30%', '年化净资产收益率': '−6.64%', '年化总资产报酬率': '−4.12%', '年化总资产净利率': '−3.58%', '销售毛利率': '12.76%', '销售净利率': '−5.28%', '销售成本率': '87.24%', '期间费用率': '12.97%', '净利润/营业总收入': '−5.28%', '营业利润/营业总收入': '−6.48%', '息税前利润/营业总收入': '−6.08%', '营业总成本/营业总收入': '107.03%', '营业费用/营业总收入': '3.32%', '管理费用/营业总收入': '9.32%', '(管理费用+研发费用)/营业总收入': '0.33%', '资产减值损失/营业总收入': '2.16%', '经营活动净收益/利润总额': '0%', '价值变动净收益/利润总额': '0%', '营业外收支净额/利润总额': '0%', '所得税/利润总额': '0%', '扣除非经常损益后的净利润/净利润': '107.88%', '销售商品提供劳务收到的现金/营业收入': '94.41%', '经营活动产生的现金流量净额/营业收入': '−119.37%', '经营活动产生的现金流量净额/经营活动净收益': '0%', '资本支出/折旧和摊销': '0%', '资产负债率': '58.38%', '权益乘数': '2.40%', '流动资产比率(流动资产/总资产)': '81.80%', '非流动资产/总资产': '18.20%', '有形资产/总资产': '30.30%', '归属母公司股东的权益/全部投入资本': '67.82%', '带息债务/全部投入资本': '32.18%', '流动负债比率(流动负债/负债合计)': '99.07%', '非流动负债比率(非流动负债/负债合计)': '0.93%', '流动比率': '1.41', '速动比率': '0.85', '保守速动比率': '0.53', '产权比率(负债合计/归属母公司股东的权益)': '1.42', '归属母公司股东的权益/负债合计': '0.70', '归属于母公司的股东权益/带息债务': '2.11', '有形资产/负债合计': '0.52', '有形资产/带息债务': '1.55', '有形资产/净债务': '3.84', '息税折旧摊销前利润/负债合计': '−0.02', '经营活动产生的现金流量净额/负债合计': '−0.33', '经营活动产生的现金流量净额/带息债务': '−0.98', '经营活动产生的现金流量净额/流动负债': '−0.33', '经营活动产生的现金流量净额/净债务': '−2.42', '已获利息倍数(ebit/利息费用)': '−18.32', '长期债务与营运资金比率': '0.02', '应收账款及应收票据周转率': '1.04次', '无形资产周转率': '12.63次', '营业周期': '241.50天', '存货周转天数': '156.82天', '应收账款周转天数': '84.68天', '存货周转率': '0.57次', '应收账款周转率': '1.06次', '流动资产周转率': '0.21次', '固定资产周转率': '5.53次', '总资产周转率': '0.17次', '基本每股收益同比增长率': '−65.30%', '稀释每股收益同比增长率': '−65.30%', '每股经营活动产生的现金流量净额同比增长率': '−80.93%', '营业总收入同比增长率': '22.13%', '营业收入同比增长率': '22.13%', '投资收益同比增长率': '9.04%', '营业利润同比增长率': '−73.33%', '利润总额同比增长率': '−64.90%', '归属母公司股东的净利润同比增长率': '−64.63%', '扣除非经常损益后的归属母公司股东的净利润同': '−74.68%', '经营活动产生的现金流量净额同比增长率': '−80.93%', '全面摊薄净资产收益率同比增长率': '−0.76%', '每股净资产相对年初增长率': '−1.65%', '资产总计相对年初增长率': '12.75%', '归属母公司的股东权益相对年初增长率': '−1.65%', '营业成本同比增长率': '26.13%', '毛利同比增长率': '0.36%', '应付账款周转率': '0.90%', '应付账款周转天数': '100.27%', '现金循环周期': '141.24天', '营运资金周转率': '0.64次', '营运资金周转天数': '140.32天', '净资产周转率': '0.38', '净资产周转天数': '236.66', '扣除非经常性损益后的基本每股收益同比增长率': '−75.29%', '已动用资本回报率roce': '−0.02', '研发费用率': '6.28%', '管理费用率': '3.04%', '销售费用率': '3.32%'}
```

对于公司报告期2025−03−31的财务审计情况如下：
```
{'归属母公司股东的净利润/报告期期末总股本': '−0.10', '归属于普通股股东的扣除非经常性损益后的净利': '−0.10', '股东权益周转率': '0.39', '人力投入回报率rop': '−14.76', '每股营业总收入': '2.21', '每股资本公积': '2.39', '每股盈余公积': '0.05', '每股未分配利润': '2.32', '每股留存收益': '2.37', '每股现金流量净额': '−0.48', '每股息税前利润': '−0.13', '每股企业自由现金流量': '0.15', '每股股东自由现金流量': '2.39', '平均净资产收益率': '−1.66', '扣除非经常损益后的平均净资产收益率': '−1.79', '净资产收益率roe—增发条件': '−1.79', '总资产净利率roa': '−0.89', '总资产报酬率roa': '−1.03', '投入资本回报率roic': '−1.30', '年化总资产净利率': '−6.64', '年化总资产报酬率': '−4.12', '年化总资产净利率': '−3.58', '销售毛利率': '12.76', '销售净利率': '−5.28', '销售成本率': '87.24', '期间费用率': '12.97', '净利润/营业总收入': '−5.28', '营业利润/营业总收入': '−6.48', '息税前利润/营业总收入': '−6.08', '营业总成本/营业总收入': '107.03', '营业费用/营业总收入': '3.32', '管理费用/营业总收入': '9.32', '(管理费用和研发费用)/营业总收入': '0.33', '资产减值损失/营业总收入': '2.16', '扣除非经常损益后的净利润/净利润': '107.88', '销售商品提供劳务收到的现金/营业收入': '94.41', '经营活动产生的现金流量净额/营业收入': '−119.37', '资产负债率': '58.38', '权益乘数': '2.40', '流动资产比率(流动资产/总资产)': '81.80', '有形资产/总资产': '18.20', '有形资产/总资产': '30.30', '归属母公司股东的权益/全部投入资本': '67.82', '带息债务/全部投入资本': '32.18', '流动负债比率(流动负债/负债合计)': '99.07', '非流动负债比率(非流动负债/负债合计)': '0.93', '流动比率': '1.41', '速动比率': '0.85', '保守速动比率': '0.53', '产权比率(负债合计/归属母公司股东的权益)': '1.42', '归属母公司股东的权益/负债合计': '0.70', '归属于母公司的股东权益/带息债务': '2.11', '有形资产/负债合计': '0.52', '有形资产/带息债务': '1.55', '有形资产/净债务': '3.84', '息税折旧摊销前利润/负债合计': '−0.02', '经营活动产生的现金流量净额/负债合计': '−0.33', '经营活动产生的现金流量净额/带息债务': '−0.98', '经营活动产生的现金流量净额/流动负债': '−0.33', '经营活动产生的现金流量净额/净债务': '−2.42', '已获利息倍数(ebit/利息费用)': '−18.32', '长期债务与营运资金比率': '0.02', '应收账款及应收票据周转率': '1.04', '无形资产周转率': '12.63', '营业周期': '241.50', '存货周转天数': '156.82', '应收账款周转天数': '84.68', '存货周转率': '0.57', '应收账款周转率': '1.06', '流动资产周转率': '0.21', '固定资产周转率': '5.53', '总资产周转率': '0.17', '基本每股收益同比增长率': '−65.30', '稀释每股收益同比增长率': '−65.30', '每股经营活动产生的现金流量净额同比增长率': '−80.93', '营业总收入同比增长率': '22.13', '营业收入同比增长率': '22.13', '投资收益同比增长率': '9.04', '营业利润同比增长率': '−73.33', '利润总额同比增长率': '−64.90', '归属母公司股东的净利润同比增长率': '−64.63', '扣除非经常损益后的归属母公司股东的净利润同': '−74.68', '经营活动产生的现金流量净额同比增长率': '−80.93', '全面摊薄净资产收益率相对年初增长率': '−0.76', '每股净资产相对年初增长率': '−1.65', '资产总计相对年初增长率': '12.75', '归属母公司的股东权益相对年初增长率': '−1.65', '营业成本同比增长率': '26.13', '毛利同比增长率': '0.36', '应付账款周转率': '0.90', '应付账款周转天数': '100.27', '现金循环周期': '141.24', '营运资金周转率': '0.64', '营运资金周转天数': '140.32', '净资产周转率': '0.38', '净资产周转天数': '236.66', '扣除非经常性损益后的基本每股收益同比增长率': '−75.29', '已动用资本回报率roce': '−0.02', '研发费用率': '6.28', '管理费用率': '3.04', '销售费用率': '3.32'}
```

// endregion

Figure 23: Example. Part 4.2: Stock Fundamentals Report.

//region 基础信息:
神州信息(股票代码: 000555.SZ)的基本信息如下:
一、主营业务: 技术服务、农业信息化、应用软件开发、金融专用设备相关业务及集成解决方案。
二、所属行业: ['元件III', '终端设备', 'IT服务III', 'IT服务']
三、所属概念(按当前相关性排序, 从高往低): 1.数字货币: 公司领先市场发布区块链平台Sm@rtGAS和数字货币(DCEP)解决方案。数字钱包系统作为承载数字货币、发展数字货币相关业务的重要系统, 公司目前已在建设银行、广发银行、北京银行实现落地。
2.跨境支付(CIPS): 根据2025年1月15日互动易: 公司参与中国现代化支付体系建设多年, 在支付业务方面, 公司主要涉及大额实时支付系统、小额批量支付系统等支付清算业务, 有上百家银行案例。公司拥有CIPS相关技术储备, 在跨境、外币相关系统方面已有项目落地, 具备提供人民银行现代化支付体系相关系统完整的解决方案能力。
3.互联网金融: 2023年9月18日互动易: 公司深耕数字金融领域, 以未来银行整体架构规划ModelB@ank为指引, 形成了以"核心应用、云计算、数据智能、智能银行、数字金融、信贷管理、风险管理和科技监管"为矩阵的八大产品族, 并持续探索创新。公司助力金融客户建设手机银行、网上银行、视频银行、移动展业平台等数字化渠道, 利用积分、权益、商城等数字化营销手段实现获客和活客, 并通过互联网金融中台和开放平台输出金融能力下沉各场景。
4.财税数字化: 2024年9月10日互动易, 2024年上半年, 在财税数字化领域, 公司基于多年参与金税工程的业务和技术优势, 深度参与金税三期和四期项目建设, 在区块链新技术探索、国家信息工程网络安全防护响应、混合云应用研究、大数据技术应用、不同部委数据共享等多个方面, 持续推动财税数字化建设。上半年, 公司中标签约国家税务总局多边税务数据服务平台升级完善及运维项目, 参与到北京、浙江、江苏、山西等21个省级税务局的数据交换共享、管理决策支持核算与分析系统、智慧税务办公平台建设等多个项目中, 同时积极推进多个地市级税务局的信息系统建设。
5.华为鲲鹏
6.区块链: 2019年10月28日公司互动平台回复公司在2016年开始在区块链方面进行跟踪和研究, 公司还专门成立区块链研究院, 拥有专门的团队研究底层技术, 并已经掌握了区块链的核心技术, 坚持自主创新。
7.信创
8.鸿蒙概念: 2024年5月11日互动易: 公司与HUAWEI合作覆盖金融科技、智能服务以及物联网等多领域的行业应用解决方案及运维服务, 已经成为HUAWEI在中国行业解决方案领域最大的合作伙伴之一。2024年4月, 公司成功完成"fPaaS全渠道技术开发平台"的HarmonyOS NEXT原生基线版本的适配开发工作, 并正式对外发布。作为HUAWEI的战略合作伙伴, 公司不仅是HUAWEI鸿蒙首批生态伙伴, 更是率先完成面向金融机构的鸿蒙原生移动应用平台——"fPaaS全渠道技术开发平台", 帮助金融机构完成面向柜银、网银、移动信贷、移动展业等多场景的应用搭建。
9.网络安全: 公司拥有"信息安全"相关业务, 是国家信息安全战略的重要参与者和推动者。公司具有"国家安全可靠计算机信息系统集成重点企业"资质, 联合成立安全可靠产业联盟"龙安联盟", 积极推动我国各个行业的信息安全水平。
10.数据要素: 2025年4月1日公告: 公司以数据要素为驱动, 构建了全域数据管理平台, 为金融客户提供全面数据体系咨询和方案落地。形成了数据汇聚、数据治理、数据资产、数据应用、数据服务等数据解决方案。
11.AI智能体: 2025年02月18日官微: 作为国内领先的金融科技企业, 神州信息完成旗下重要集成服务解决方案"神州灵境"与DeepSeek大模型R1&V3的接入, 基于DeepSeek大模型在多任务、多模态和多语言以及知识问答领域的优势, 结合多行业场景智能化升级和应用调优, 率先推出"运维智能体、办公智能体和业务管理智能体", 通过一站式应用部署, 让客户直观感受到大模型带来的智能变化。
12.ERP概念
13.数字乡村: 主营技术服务、农业信息化相关业务及集成解决方案, 为我国金融、农业等国民经济重点行业提供技术服务、应用软件开发以及行业云建设及运营等产品和服务;
14.腾讯概念: 2024年11月7日互动易: 公司作为腾讯生态合作伙伴受邀参会, 基于腾讯云在云计算、数据库、大数据平台等领域的基础产品能力, 叠加公司在金融自主创新和金融核心系统建设领域的深厚实践积累及"科技+数据+场景"的融合创新能力, 双方围绕金融应用创新及行业数字化转型等聚焦发力。与此同时, 公司参与腾讯云发起的"腾讯云行业大模型生态计划", 共同推进大模型在产业领域的创新和落地。
15.阿里巴巴概念: 根据2023年9月18日互动易, 公司围绕金融信创与华为、阿里、中兴等生态厂商紧密合作, 落地某股份制银行PaaS云平台项目、某农信数据湖项目及某城商行千节点分布式数据库项目等多个场景类解决方案, 金融信创生态环境日益成熟。
16.量子科技: 神州信息于2012年开始在量子通信领域深入耕耘, 现阶段业务已覆盖量子通信网络建设、量子通信网络运行维护服务、量子通信网络应用开发等方面。量子计算属于量子技术的另外一个专业领域, 目前公司业务未涉及量子计算方面。
17.虚拟数字人: 根据2023年6月8日互动易, 数字人"小信"已嵌入公司最新版的个人手机银行APP产品中, 可在客户使用过程中按需提供财报、生活管家、财富助手、无障碍服务等多种服务模式, 提供全流程的智能化、场景化的客户旅程服务, 以及智能、有温度的引导服务。
18.AIGC概念: 2025年1月15日互动易回复, 公司以AIGC为核心, 融合多种"AI+"技术, 成功实现"九天揽月云原生金融PaaS平台"的智能迭代, 成功破解金融行业研发创新投入大、周期长的难点, 全面推动金融数字化转型。
19.智慧城市: 公司打造物联、数联、智联的物联网一体化开发工具产品链, 积累智慧园区通用业务场景和业务数据模型, 打造智慧城市和智慧园区物联网行业内开箱即用的物联网感知平台产品。
20.人工智能: 2019年1月份, 公司在互动平台披露, 公司拥有自主的人工智能技术及产品并应用于智能网点, 互联网平台等业务。
21.DeepSeek概念: 2025年2月19日官微: 在数字化转型浪潮中, 金融行业对软件开发的效率、安全性及合规性提出了更高要求。作为中国领先的金融科技全产业链服务商, 神州信息在极短时间内迅速完成了DeepSeek-R1满血版开源模型的本地化部署, 并接入金融代码智能辅助平台CodeMaster, 通过AI重构金融科技开发流程, 实现从代码生成到质量管控的全生命周期赋能, 助力金融机构迈向"智能编码"新纪元。
22.华为概念: 根据2025年4月21日互动易: 公司为领先的金融数字化转型合作伙伴, 多年来与华为携手, 通过持续的能力建设、技术交流和联合解决方案的发布, 覆盖金融、政企、制造、互联网、通信等重点行业领域和行业应用解决方案及运维服务等多维度服务。
23.物联网: 公司与华为共同发布的"智慧城市物联网解决方案", 成功落地北京副中心智慧城市物联网项目及中山市翠亨新区城市物联网平台项目。公司还开发了自主知识产权的物联网感知平台、物联网连接管理平台等系列产品。
24.5G: 2022年1月20日互动易回复: 公司能够提供基于5G的无线网络优化、通信大数据、物联网等产品及服务, 并自主研发5G测试终端。
// endregion

Figure 24: Example. Part 5: Stock Basic Information.

```
//region 技术面数据:
# 000555.SZ量化分析报告

## 一、核心特征研判
1. **趋势特征判定**
   – 当前价格为27.47元，较前一交易日上涨0.07元（0.25%），显示出一定的上升趋势。
   – 5日均线（MA5）为26.03元，10日均线（MA10）为25.62元，20日均线（MA20）为25.48元，均线呈现多头排列，表明
短期趋势向上。

2. **量能突破分析**
   – 最近交易日（2025年5月30日）成交量为1.03亿，显著高于5日均量（0.45亿）和10日均量（0.33亿），显示出放量特
征。
   – 5月29日的成交量为0.69亿，较前几日有明显增加，且价格上涨10.04%，表明放量与价格上涨的协同。

3. **交易活跃度评估**
   – 最近5个交易日中，上涨天数为2天，下跌天数为2天，显示出一定的波动性。
   – 平均振幅为4.77%，表明日内交易空间较为充足。

4. **适合度综合评分**
   – 综合趋势、量能和活跃度，000555.SZ具备较好的短期交易特征，适合进行日内波段操作。

## 二、多周期趋势诊断
1. **短期（5日）市场特征**
   – 涨幅：5日累计收益率为9.51%。
   – 波动率：日均波动率为0.04，显示出较低的短期波动。
   – 上涨天数占比：40%（2天上涨）。

2. **中期（15日）市场特征**
   – 涨幅：15日累计收益率为10.41%。
   – 波动率：中期波动率相对稳定，显示出趋势的持续性。
   – 上涨天数占比：约53%（8天上涨）。

3. **长期（30日）市场特征**
   – 涨幅：30日累计收益率未提供，需进一步分析。
   – 波动率：长期波动率相对较低，表明趋势稳定。
   – 上涨天数占比：约60%（18天上涨）。

4. **趋势拐点信号研判**
   – MACD指标显示出金叉信号，表明短期内可能继续上涨。
   – RSI指标为54.43，未进入超买区，仍有上升空间。

## 三、量价关系剖析
1. **量能变化特征**
   – 最近成交量显著放大，当前量能为1.03亿，较5日均量高出130%。
   – 价量相关系数为正，表明价格与成交量呈正相关。

2. **换手率诊断**
   – 日均换手率为11.82%，显示出市场活跃度较高。
   – 换手率波动特征良好，未出现异常波动。

3. **量价背离分析**
   – 当前价格上涨伴随成交量放大，未出现量价背离现象，表明趋势健康。

4. **市场活跃度评估**
   – 近期市场活跃度较高，适合短线交易。

## 四、风险收益特征
1. **收益能力评估**
   – 累计收益率：5日为9.51%，15日为10.41%。
   – 年化收益率：未提供，需进一步计算。

2. **风险控制水平**
   – 年化波动率为0.57，最大回撤为−0.41，表明风险控制良好。
   – VaR(95%)为−0.05，表明在95%的置信水平下，最大可能损失为5%。

3. **风险调整后表现**
   – 夏普比率为0.36，表明风险调整后的收益尚可。
   – 索提诺比率和卡玛比率未提供，需进一步分析。

4. **收益风险配比**
   – 当前收益与风险的配比合理，适合进行短期交易。

## 五、市场比较分析
1. **同业竞争力评估**
   – 与同行业标的（如002065.SZ、002368.SZ）相比，000555.SZ的5日累计收益率为9.51%，明显高于同行业的平均水
平。

2. **市场联动性分析**
   – 相关系数分析显示，000555.SZ与市场指数（如IT服务、ERP概念）相关性较高，表明其受市场整体趋势影响较大。

3. **超额收益分析**
   – 000555.SZ在最近5日的超额收益为5.4%，显示出较强的市场表现。

4. **Alpha贡献度**
   – Alpha贡献度未提供，需进一步计算。
// endregion
```

Figure 25: Example. Part 6: Stock Quantitative Reports.

<think>
好的，我现在要预测神州信息（000555.SZ）在下一个交易日（即6月3日）的开盘价相对于5月30日的收盘价的涨跌幅。我的分析流程如下：

1. **理解所要分析的股票的个性**：神州信息属于IT服务行业，业务涵盖金融科技、量子通信、数字货币等多个高科技领域。作为行业龙头，其股价波动受政策和技术突破双重影响。需要特别关注其在AIGC（生成式AI）领域的布局以及与华为等科技巨头的合作。

2. **理解所要预测的时间特征**：当前处于5月底至6月初的时间窗口，属于半年报披露前的敏感期。同时临近周末，需警惕周五异动对周初情绪的影响，但周一作为新的交易周通常存在惯性上涨效应。

3. **查看所提供的市场状态**：
   – **宏观环境**：A股科技板块热度持续，但整体市场面临流动性压力。美债收益率上行压制风险偏好，需警惕外资流动对成长股估值的压制。
   – **基本面数据**：公司2024年出现亏损，经营现金流持续为负，资产负债率超过58%。这些财务指标显示公司处于基本面修复初期，估值修复动能可能受制于盈利能力的不确定性。
   – **价格信息**：5月30日收盘价12.09元，较前一日上涨0.0025，成交量激增至1.03亿股创年内新高，呈现量价齐升态势。技术面显示5日均线（11.97）与10日均线（11.78）形成多头排列，MACD红柱放大。
   – **融资融券数据**：5月29日融资买入6044.86万元，占当日流入资金比例12.65%，融资余额5.15亿元触及历史60%分位。融券余额174.75万元处于低位，显示做空力量有限。
   – **新闻事件**：连续三天发布与AI金融大模型相关的重磅合作，包括与华为联合发布智能体、发起行业首个AIGC生态体系等。这些事件形成持续催化效应，推升市场关注度。

4. **初步构建分析方法逻辑**：
   – 结合技术面量价配合与资金流入情况，判断是否存在突破性上涨动能。
   – 评估政策利好（香港稳定币法案）与基本面承压（亏损、现金流）的矛盾。
   – 关注相似股票（如300079.SZ、300479.SZ）的走势对板块情绪的传导作用。

5. **按分析方法逻辑分析各维度信息**：
   – **利多因素**：
     – 技术面：5日量价齐升突破整理区间，MACD金叉形成多头趋势。
     – 政策支持：香港稳定币法案推动数字货币板块整体走强，神州信息作为概念股核心受益。
     – 事件驱动：连续三天发布AI金融大模型相关成果，与华为等巨头合作形成持续催化。
     – 融资盘介入：融资余额突破历史60%分位，显示杠杆资金积极布局。
   – **利空因素**：
     – 基本面孱弱：2024年亏损、现金流为负，经营效率低下制约估值修复空间。
     – 流动性压力：美债收益率上行可能抑制成长股估值，板块轮动压力显现。
     – 技术面阻力：12.09元附近面临12.10元整数关口压力，需观察量能能否持续放大。

6. **证据评分**：
   – **上涨支持因素**：
     1. 技术面量价配合（9分）：突破整理区间伴随放量，MACD金叉确认趋势。
     2. 政策利好催化（8分）：香港稳定币法案提升板块情绪，数字货币概念股普涨。
     3. 事件驱动持续性（8分）：三天连续发布AI金融成果，形成技术面多头氛围。
     4. 融资盘介入（7分）：融资余额突破历史分位，杠杆资金形成推力。
   – **下跌支持因素**：
     1. 基本面孱弱（7分）：亏损与现金流为负制约估值修复，需警惕技术性回调。
     2. 流动性压力（6分）：美债收益率上行可能引发板块资金撤离。
     3. 技术面阻力（6分）：12.10元关口存在压力，量能若无法持续可能引发获利了结。
     4. 融券余额低位（5分）：融券力量薄弱，空头打压动能不足。

7. **假设检验与推演**：
   – 如果政策利好持续发酵，AI金融大模型落地项目加速，股价可能突破12.10元阻力位。
   – 若市场对亏损业绩存在分歧，可能出现技术性回调至11.80元附近。
   – 考虑到相似股票（300079.SZ）在利好驱动下涨幅接近5%，板块联动效应可能放大个股涨幅。

8. **综合评分计算**：
   – 上涨平均分：（9+8+8+7）/4=8.0
   – 下跌平均分：（7+6+6+5）/4=6.0

9. **最终预测**：
   虽然基本面存在亏损隐患，但技术面量价配合与政策利好形成共振，且融资盘介入形成推力。考虑到周末消息面无重大利空，预计周一开盘将延续涨势，涨幅3%左右，方向为显著上涨。

</think>

<score>[8.0, 6.0]</score>
<pct_change>0.0300</pct_change>
<answer>up</answer>

Figure 26: Example. Part 7: Model Response.

```
[Ground Truth] Using provided label: up
[Ground Truth] Final change_pct: 0.0355665839536807, Status: up

[Format Validation]
  Format validation: PASS
  Format score: 1

[Content Validation]
  Expected: up
  Predicted: up
  Content validation: FULL MATCH
  Content validation: UP BONUS
  Content validation: change_pct GOLDEN MATCH <- |0.03-0.0355665839536807| < 0.01
  Content validation: change_pct and answer FULL MATCH

----------------------------------------------------------------------------
------------------------------- Final Score ---------------------------------
  Format: 1
  Answer: 4
  Total: 6
============================================================================
```

Figure 27: Example. Part 8: Model Response Grading.

`<change_pct>`, `<answer>`) against the ground truth, and ensures adherence to the desired output style. At this stage, 300 samples passed the validation. From these, we cherry-picked 188 samples in total, which required approximately six hours of expert annotation (around 100 USD) for final curation.

Notably, the financial data we collected spans the entire year of 2024 and June 2025. Among these, the data from January to November 2024 were used for training. From this subset of training, we extracted 300 instances (100 labeled as *up*, 100 as *hold*, and 100 as *down*) to construct the cold-start dataset for synthesis.

**RETuning** guides the model through a multistep process:

1. **Understand the Task.** The model first internalizes the task—predicting the direction of price movement from the previous close to the next open, with a $\pm 3\%$ threshold—improving alignment and reducing hallucination.

2. **Establish Analytical Principles.** It constructs a dynamic analytical framework (e.g., fundamentals, news trends, macro signals), independent of analyst commentary.

3. **Extract Evidence.** The model collects multiple context-based signals and assesses their support for each directional hypothesis.

4. **Group and Score.** Extracted evidence is grouped and scored by directional leaning, forming a soft-evidence pool.

5. **Reflect and Reconcile.** Averaging directional scores, the model enters a reflection phase to resolve conflicting evidence.

6. **Produce Structured Output.** The model generates a final decision in a consistent, interpretable format, specifying both direction and percentage change.

In detail, we employed the following workflow:

- **Step 1: Multi-Source Data Aggregation.** We systematically collected a diverse range of financial data covering the period from January 2024 to June 2025. This included quantitative data, such as historical stock prices (open, high, low, close) and trading volumes, alongside qualitative data from relevant financial news articles. For training purposes, we focused on the January–November 2024 subset, from which 300 balanced instances (up, hold, down) were selected for cold-start data synthesis. To ensure the dataset's breadth and representativeness, we selected a varied portfolio of stocks from multiple market sectors.

- **Step 2: Structured Prompt Formulation.** We engineered a series of structured prompts designed to elicit detailed, context-aware analytical reasoning from the language model.

These prompts incorporated explicit instructions, illustrative examples (few-shot learning), and comprehensive contextual information (e.g., market conditions, company background) to precisely guide the model's generation process according to our proposed thinking schema.

- **Step 3: Controlled Response Generation.** Leveraging the 671B-parameter DeepSeek-R1 model, we executed inference on the engineered prompts. To foster response diversity and prevent deterministic outputs, we employed stochastic sampling techniques, specifically temperature sampling and top-k sampling. This allowed the model to explore a wider range of analytical paths and linguistic styles.

- **Step 4: Hybrid Quality Assurance.** The generated responses underwent a rigorous two-stage filtering process. Initially, automated metrics were used to assess fundamental quality attributes such as textual coherence, logical consistency, and relevance to the prompt. Subsequently, all machine-vetted samples were subjected to manual review by a financial expert to discard any outputs that were factually incorrect, nonsensical, or failed to meet the required analytical depth.

- **Step 5: Semantic Data Augmentation.** To enrich the dataset and enhance model robustness, we applied several data augmentation techniques to the high-quality samples. Methods such as paraphrasing, synonym replacement, and back-translation were utilized to create semantically equivalent but syntactically diverse training instances, thereby reducing the risk of overfitting to specific phrasings.

- **Step 6: Final Dataset Curation.** The fully processed and augmented data was compiled into the final training set. During this stage, we ensured a balanced class distribution among the target labels (up, down, hold) and a representative allocation of stocks across different sectors and market conditions to form a well-rounded and unbiased dataset for fine-tuning.

---

你是一个专业的编辑，请按照样例将下面的文本进行润色，保持原意不变，增加细节，总体上和样例中的和"修改后"相匹配。请注意，润色后的文本应该是流畅的中文，不要有英文或其他语言的内容。请不要添加任何额外的解释或评论，只需提供润色后的文本即可。

提示一下，在润色时，你需要注意 think 包含了这些关键部分，即：对做好这个预测任务所需的关键条件的思考；整理自己的状态；明确自己的分析范式；执行范式；动态构建分析方法逻辑；应用分析方法；审慎综合分析结果，给出合理的预测。
注意，在think开始的时候，并没有急于给出分析结果，而是先从基础假设出发，建立自己的分析框架，再慢慢开始做任务。在编写think的过程中，应该有气定神闲、从容不迫的感觉。你需要在润色时保持这种语气。
注意，润色后的综合分数应该和<score>里的分数相匹配，比如<score>[6.0, 7.5]</score>，则上涨因素平均分为6，下跌因素平均分为7.5。
同时 <pct_change> 里的涨跌幅要和think里预测的涨跌幅相匹配，比如<pct_change>−0.0150</pct_change>，则涨跌幅预测为−0.0150。
同时，<answer>里预测的方向要和think里预测的方向相匹配，比如<answer>hold</answer>，则方向预测为hold。

另外，你必须增加以下细节：
1. 复述任务的时候，增加对任务目标的具体细节，比如"我现在需要预测银宝山新（002786.SZ）在下一个交易日（即2024年1月15日）的开盘价相对于2024年1月12日的收盘价的价格走势"。
2. 润色关键条件，使得表述多样化。可参考后面的思维准备样例。
3. 增加对价格数据的深度分析，将//region 价格 ... //endregion 中的信息利用起来，进行技术性推演
4. 在得到证据及评分后，额外增加假设检验，市场模拟，未来推演，反事实假设等，对这些证据进行反思，进一步优化（增加或修改）证据和评分，直到你确信在新的评分后你已经考虑了所有可能的情况。
5. 请保留口语化的规范。修改后的参考中的每一个章节都很重要，请不要省略任何章节。你需要在每个章节中都增加细节，增加对分析的深度和广度的思考。

【修改】样例：

//region 修改前
{{ example_thought_before_polished }}
//endregion
//region 修改后的think
{{ example_thought_after_polished }}
//endregion

//region 需要润色的文本
{{ todo_thought_to_be_polished }}
//endregion

//region 润色时可参考的资料
{{ stock_movement_prediction_prompt_without_task_desc }}
//endregion

你的输出应该包含在"//region 我的修改后的think"和"//endregion"之间的文本。

Figure 28: The polish prompt allows the backbone model to generate refined responses that follow the proposed thinking schema in Section 4.1.

## C.3 BIZFINBENCH

BizFinBench Lu et al. (2025) is a comprehensive financial benchmark covering 10 tasks, including financial analysis, financial news classification, financial text summarization, financial question answering, and financial named entity recognition. We evaluate the generalization ability of **RETuning** on BizFinBench. The details of each dataset type are as follows.

- **Anomalous Event Attribution (AEA):** This dataset evaluates the model's ability to trace financial anomalies based on given information such as timestamps, news articles, financial reports, and stock movements. The model must identify the cause-and-effect relationships behind sudden market fluctuations and distinguish relevant factors from noise.

- **Financial Numerical Computation (FNC):** This dataset assesses the model's ability to perform accurate numerical calculations in financial scenarios, including interest rate calculations, return on investment (ROI), and financial ratios.

- **Financial Time Reasoning (FTR):** This dataset tests the model's ability to understand and reason about time-based financial events, such as predicting interest accruals, identifying the impact of quarterly reports, and assessing financial trends over different periods.

- **Financial Tool Usage (FTU):** This dataset evaluates the model's ability to comprehend user queries and effectively use financial tools to solve real-world problems. It covers scenarios like investment analysis, market research, and information retrieval, requiring the model to select appropriate tools, input parameters accurately, and coordinate multiple tools when needed.

- **Financial Knowledge QA (FQA):** This dataset evaluates the model's understanding and response capabilities regarding core knowledge in the financial domain. It spans a wide range of financial topics, encompassing key areas such as fundamental financial concepts, financial markets, investment theory, macroeconomics, and finance.

- **Financial Data Description (FDD):** This dataset measures the model's ability to analyze and describe structured and unstructured financial data, such as balance sheets, stock reports, and financial statements.

- **Emotion Recognition (ER):** This dataset evaluates the model's capability to recognize nuanced user emotions in complex financial market environments. The input data encompasses multiple dimensions, including market conditions, news articles, research reports, user portfolio information, and queries. The dataset covers six distinct emotional categories: optimism, anxiety, negativity, excitement, calmness, and regret.

- **Stock Price Prediction (SP):** This dataset evaluates the model's ability to predict future stock prices based on historical trends, financial indicators, and market news.

- **Financial Named Entity Recognition (FNER):** This dataset focuses on evaluating the model's ability to identify and classify financial entities such as company names, stock symbols, financial instruments, regulatory agencies, and economic indicators.

Table 5 presents a detailed breakdown of the dataset, covering the evaluation dimensions, corresponding metrics, the number of instances per task, and the average token length per entry. Notably, the dataset shows considerable variability in input length, spanning from a minimum of 22 tokens to a maximum of 4,556 tokens. This wide range not only mirrors the complexity and heterogeneity of real-world financial scenarios but also poses a meaningful challenge for models—specifically, in demonstrating their capability to process both short and long financial texts effectively. Table 6 presents their maximum token length, minimum token length, and average length.

## D    EVALUATION DETAILS

We evaluate our model using several metrics, including accuracy, precision, recall, and F1-score. These metrics provide a comprehensive view of the model's performance across different aspects.

Table 5: Overview of BizFinBench Lu et al. (2025) Datasets

| Category | Data | Evaluation Dimensions | Metrics | Numbers | Avg Len. |
|---|---|---|---|---|---|
| Reasoning | Anomalous Event Attribution (AEA) | Causal consistency Information relevance Noise resistance | Accuracy | 1064 | 939 |
| | Financial Time Reasoning (FTR) | Temporal reasoning correctness | Accuracy | 514 | 1162 |
| | Financial Tool Usage (FTU) | Tool selection appropriateness Parameter input accuracy Multi-tool coordination | Judge Score | 641 | 4556 |
| Numerical calculation | Financial Numerical Computation (FNC) | Computational accuracy Unit consistency | Accuracy | 581 | 651 |
| Q&A | Financial Knowledge QA (FQA) | Question comprehension Knowledge coverage Answer accuracy | Judge Score | 990 | 22 |
| | Financial Data Description (FDD) | Trend accuracy Data consistency | Judge Score | 1461 | 311 |
| Prediction recognition | Emotion Recognition (ER) | Emotion classification accuracy Implicit information extraction | Accuracy | 600 | 2179 |
| | Stock Price Prediction (SP) | Trend judgment, Causal reasoning | Accuracy | 497 | 4498 |
| Information extraction | Financial Named Entity Recognition (FNER) | Recognition accuracy Entity classification correctness | Accuracy | 435 | 533 |

Table 6: Token Length Statistics of BizFinBench Lu et al. (2025).

| Dataset | Min | Max | Avg | Count |
|---|---|---|---|---|
| NER | 415 | 1,194 | 533.1 | 433 |
| FTU | 4,169 | 6,289 | 4,555.5 | 641 |
| AEA | 680 | 1,396 | 938.7 | 1,064 |
| ER | 1,919 | 2,569 | 2,178.5 | 600 |
| FNC | 287 | 2,698 | 650.5 | 581 |
| FDD | 26 | 645 | 310.9 | 1,461 |
| FTR | 203 | 8,265 | 1,162.0 | 514 |
| FQA | 5 | 45 | 21.7 | 990 |
| SP | 1,254 | 5,532 | 4,498.1 | 497 |

## D.1 DETAILED RESULTS ON FIN-2024[DECEMBER]

We present detailed results on **Fin-2024[December]** for different baselines in Figure 29. The results are grouped by the number of repeated sampling counts $n$ ($\log_2$ scale: 1, 2, 4, 8, 16, 32). Performance is measured using the stock movement prediction (SMP) F1 score. It can be observed that current LLMs struggle to achieve satisfactory performance on this challenging task. Most of them are not able to scale up their performance with increasing $n$. In contrast, our proposed **RETuning** method demonstrates significant improvements, especially when combined with larger models and reinforcement learning techniques. Notably, the **DeepSeek_R1_32B_SFT_GRPO** model achieves the highest F1 score of approximately 0.44 at $n = 32$, showcasing the effectiveness of our approach in enhancing model capabilities for stock movement prediction.

## D.2 RESULTS GROUPED BY OOD SPLIT ON FIN-2024[DECEMBER] AND FIN-2025[JUNE]

This section analyzes the performance of different models on the **stock movement prediction (SMP) task**, a 3-category classification problem based on price changes, under four out-of-distribution (OOD) scenarios. The evaluation focuses on the models' ability to *scale up* by increasing the repeated sampling count $n$ ($\log_2$ scale: 1, 2, 4, 8, 16, 32). Performance is measured using the SMP F1 score, and the comparison includes a baseline (Random Bound), the 14B/32B variants of DeepSeek-R1, and their optimized versions (SFT, SFT+GRPO).

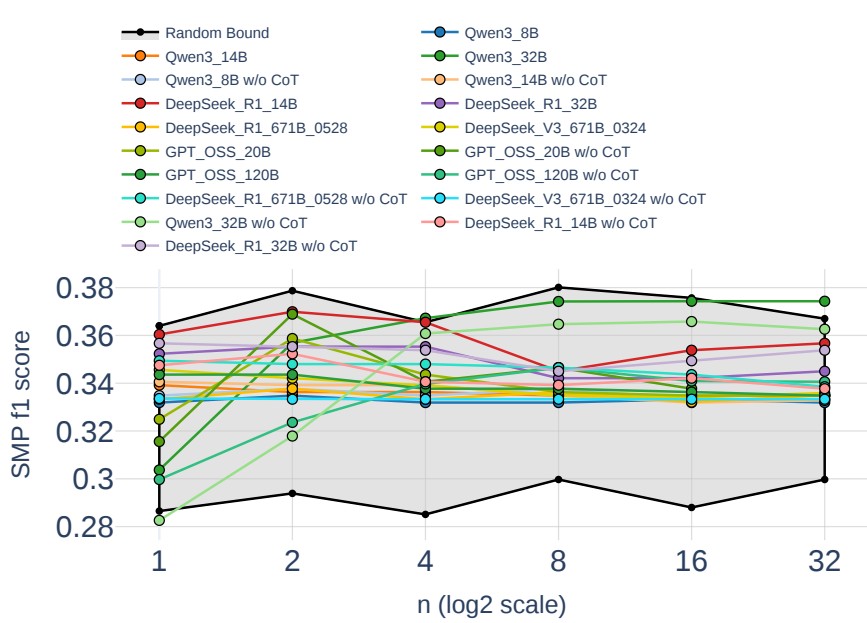

Figure 29: Detailed results on **Fin-2024[December]** for different baselines.

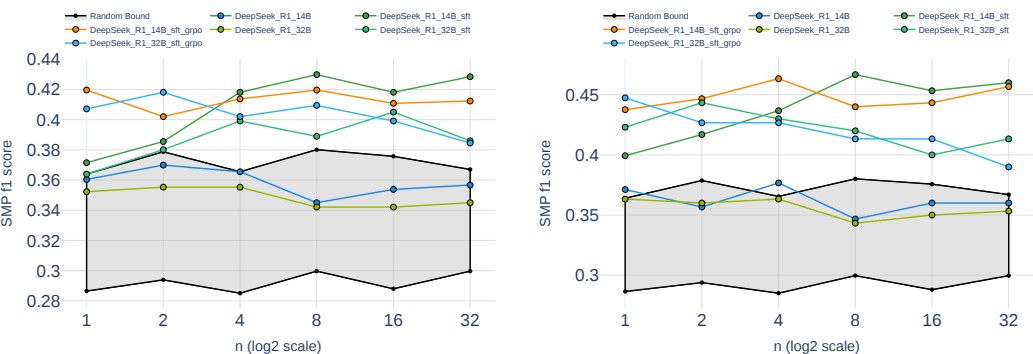

Figure 30: **Overall** f1 score results on **Fin-2024[December]**.

Figure 31: **OOD_Stock** f1 score results on **Fin-2024[December]**.

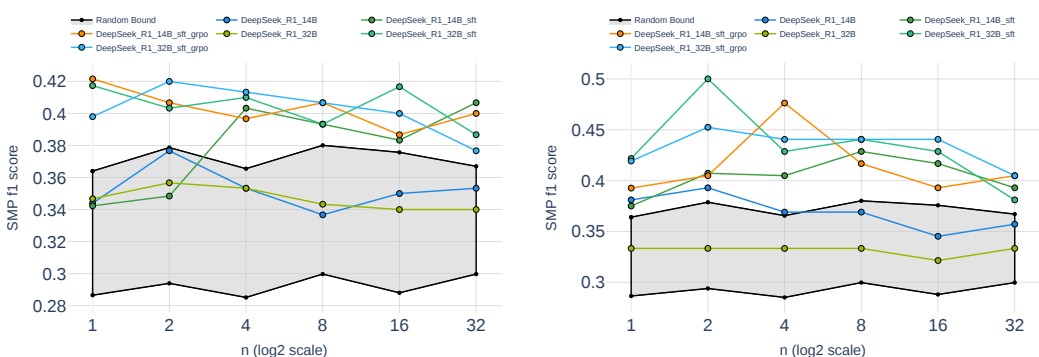

Figure 32: **OOD_Date** f1 score results on **Fin-2024[December]**.

Figure 33: **OOD_Stock&Date** f1 score results on **Fin-2024[December]**.

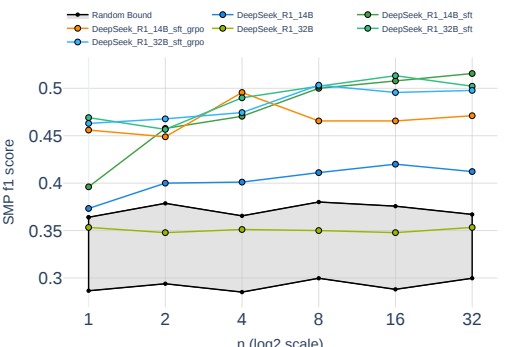 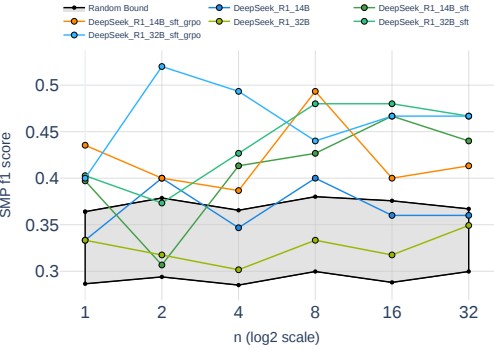

Figure 34: **OOD_Date** f1 score results on **Fin-2025[June]**.

Figure 35: **OOD_Stock&Date** f1 score results on **Fin-20254[June]**.

### D.2.1    OVERALL OOD PERFORMANCE

As shown in Figure 30, which aggregates results from the **OOD_Stock**, **OOD_Date**, and **OOD_Stock&Date** scenarios, two consistent trends emerge. First, a clear performance hierarchy is observed: 32B models consistently outperform their 14B counterparts, and optimization further improves results (SFT > Base; SFT+GRPO > SFT). Among all configurations, the **DeepSeek_R1_32B_SFT_GRPO** model achieves the highest F1 score of approximately 0.44 at $n = 32$, followed by the 32B SFT and Base models. Importantly, all variants surpass the Random Bound baseline (F1 < 0.30), demonstrating the benefits of both scaling and optimization. Second, all models exhibit *monotonic gains* as $n$ increases. For instance, DeepSeek_R1_32B_SFT_GRPO improves from about 0.36 at $n = 1$ to 0.44 at $n = 32$, corresponding to a relative gain of roughly 22%. In contrast, the 14B baseline improves only from 0.28 to 0.34 over the same range. These results confirm that repeated sampling consistently enhances performance, with larger and optimized models benefiting the most.

### D.2.2    OOD_STOCK SCENARIO (UNSEEN STOCKS)

Figure 31 evaluates performance when predicting trends for **stocks unseen during training**. This setting achieves the second-highest peak performance across all scenarios, with DeepSeek_R1_32B_SFT_GRPO reaching about 0.45 at $n = 32$. The performance gap between 32B and 14B optimized models widens to roughly 0.07 at this sampling level, underscoring the importance of parameter scale for generalizing to new stocks. Moreover, F1 scores rise sharply as $n$ increases. For example, the 32B SFT model improves from 0.37 at $n = 1$ to 0.43 at $n = 32$, a gain of around 16%. These findings suggest that repeated sampling effectively reduces noise in trend estimation for unfamiliar stocks, thereby mitigating the distribution shift.

### D.2.3    OOD_DATE SCENARIO (UNSEEN TIME PERIODS)

In the **OOD_Date** setting (Figure 32), where models are tested on unseen time periods such as December 2024, performance is notably weaker. This scenario yields the lowest peak among all four settings, with the best-performing model (32B SFT+GRPO) reaching only about 0.42 at $n = 32$. Even with maximum scaling, the gains are modest: DeepSeek_R1_32B_SFT_GRPO improves by only 0.08 (from 0.34 to 0.42) as $n$ increases from 1 to 32, which is smaller than the gains observed in stock-based shifts. These results indicate that temporal distribution shifts, likely caused by market regime changes, are more difficult to address with repeated sampling alone.

### D.2.4    OOD_STOCK&DATE SCENARIO (DUAL DISTRIBUTION SHIFTS)

The most challenging case, involving **both unseen stocks and unseen time periods**, is shown in Figure 33. Surprisingly, this dual-shift setting achieves the *highest overall peak performance*. DeepSeek_R1_32B_SFT_GRPO reaches approximately 0.50 at $n = 32$, surpassing even the OOD_Stock scenario. The scaling effect is particularly strong: the model improves by about 0.12 (from 0.38 to 0.50) when $n$ increases from 1 to 32. Moreover, the performance gap between the

32B optimized model and the 14B baseline widens substantially with larger $n$. These results highlight a synergy between model scale and repeated sampling, suggesting that larger models are especially capable of leveraging additional samples to disentangle stock-specific volatility from time-dependent shifts.

### D.2.5 SUMMARY OF SCALING-UP ABILITY ACROSS OOD SCENARIOS

Taken together, the results provide several key insights. First, repeated sampling proves to be a universally valid strategy, as all models improve with increasing $n$ across every OOD setting. Second, the magnitude of gains is scenario-dependent: the largest improvements occur in the dual OOD_Stock&Date scenario (12–22% increase), while the OOD_Date scenario yields the smallest gains (8–15%), reflecting the particular difficulty of temporal shifts. Third, the optimal configuration for scaling-up ability is the **DeepSeek_R1_32B_SFT_GRPO** model at $n = 32$, which achieves F1 scores of roughly 0.50 (OOD_Stock&Date), 0.45 (OOD_Stock), 0.44 (Overall OOD), and 0.42 (OOD_Date), consistently outperforming all other settings by 5–15%. Finally, the results underscore the **synergy between model scale, optimization, and repeated sampling**: larger models not only benefit more from optimization but also extract greater value from additional samples, with the effect most pronounced in dual-shift scenarios.

### D.3 DETAILED RESULTS ON BIZFINBENCH

We present the detailed results of various large language models on the BizFinBench benchmark Lu et al. (2025), as summarized in Table 7. The evaluation encompasses a range of tasks, including financial entity recognition, financial question answering, financial text classification, and more. The results are color-coded to highlight the top three performers for each task: golden indicates the top-performing model, silver represents the second-best result, and bronze denotes the third-best performance.

## E CASE STUDY

### E.1 CASE STUDY ON ZERO-SHOT COT PROMPTING

We present a case study comparing the responses of GPT4o, DeepSeek-R1 and Qwen3 under two different prompting strategies: zero-shot prompting without Chain-of-Thought (CoT) and zero-shot prompting with CoT. The results are illustrated in Figures 36 and 37.

We observe that without CoT prompting (Figure 36), the models struggle to provide accurate and relevant answers. Their responses often lack depth and fail to address the nuances of the questions. In contrast, with CoT prompting (Figure 37), the models demonstrate improved reasoning and accuracy. The step-by-step thought process enabled by CoT helps the models to break down complex questions and arrive at more precise answers.

### E.2 CASE STUDY ON COLD-STARTED RESPONSES

We present a case study comparing the original response and the cold-started response generated by DeepSeek_R1_14B_Instruct and DeepSeek_R1_14B_SFT. The results are illustrated in Figure 38. Before applying cold-starting techniques, the original response tends to be verbose and includes unnecessary elaboration. After cold-starting, the response becomes more concise and is much longer, which indicates the cold-started model is leveraging its reasoning capabilities to provide a more comprehensive answer. This demonstrates the effectiveness of cold-starting in enhancing the clarity and relevance of model-generated responses.

Table 7: Performance Comparison of Large Language Models on BizFinBench Lu et al. (2025). The models are evaluated across multiple tasks, with results color-coded to represent the top three performers for each task: golden indicates the top-performing model, silver represents the second-best result, and bronze denotes the third-best performance.

| Model | AEA | FNC | FTR | FTU | FQA | FDD | ER | SP | FNER | Average |
|---|---|---|---|---|---|---|---|---|---|---|
| *Close-Source LLMs* | | | | | | | | | | |
| ChatGPT-o3 | 86.23 | 61.30 | 75.36 | 89.15 | 91.25 | 98.55 | 44.48 | 53.27 | 65.13 | 73.86 |
| ChatGPT-o4-mini | 85.62 | 60.10 | 71.23 | 74.40 | 90.27 | 95.73 | 47.67 | 52.32 | 64.24 | 71.29 |
| GPT-4o | 79.42 | 56.51 | 76.20 | 82.37 | 87.79 | 98.84 | 45.33 | 54.33 | 65.37 | 71.80 |
| Gemini-2.0-Flash | 86.94 | 62.67 | 73.97 | 82.55 | 90.29 | 98.62 | 22.17 | 56.14 | 54.43 | 69.75 |
| Claude-3.5-Sonnet | 84.68 | 63.18 | 42.81 | 88.05 | 87.35 | 96.85 | 16.67 | 47.60 | 63.09 | 65.59 |
| *Open-Weight LLMs* | | | | | | | | | | |
| Qwen2.5-7B-Instruct | 73.87 | 32.88 | 39.38 | 79.03 | 83.34 | 78.93 | 37.50 | 51.91 | 30.31 | 56.35 |
| Qwen2.5-72B-Instruct | 69.27 | 54.28 | 70.72 | 85.29 | 87.79 | 97.43 | 35.33 | 55.13 | 54.02 | 67.70 |
| Qwen2.5-VL-3B | 53.85 | 15.92 | 17.29 | 8.95 | 81.60 | 59.44 | 39.50 | 52.49 | 21.57 | 38.96 |
| Qwen2.5-VL-7B | 73.87 | 32.71 | 40.24 | 77.85 | 83.94 | 77.41 | 38.83 | 51.91 | 33.40 | 56.68 |
| Qwen2.5-VL-14B | 37.12 | 41.44 | 53.08 | 82.07 | 84.23 | 7.97 | 37.33 | 54.93 | 47.47 | 49.52 |
| Qwen2.5-VL-32B | 76.79 | 50.00 | 62.16 | 83.57 | 85.30 | 95.95 | 40.50 | 54.93 | 68.36 | 68.62 |
| Qwen2.5-VL-72B | 69.55 | 54.11 | 69.86 | 85.18 | 87.37 | 97.34 | 35.00 | 54.94 | 54.41 | 67.53 |
| Qwen3-1.7B | 77.40 | 35.80 | 33.40 | 75.82 | 73.81 | 78.62 | 22.40 | 48.53 | 11.23 | 50.78 |
| Qwen3-4B | 83.60 | 47.40 | 50.00 | 78.19 | 82.24 | 80.16 | 42.20 | 50.51 | 25.19 | 59.94 |
| Qwen3-14B | 84.20 | 58.20 | 65.80 | 82.19 | 84.12 | 92.91 | 33.00 | 52.31 | 50.70 | 67.05 |
| Qwen3-32B | 83.80 | 59.60 | 64.60 | 85.12 | 85.43 | 95.37 | 39.00 | 52.26 | 49.19 | 68.26 |
| QwQ-32B | 84.02 | 52.91 | 64.90 | 84.81 | 89.60 | 94.20 | 34.50 | 56.68 | 30.27 | 65.77 |
| Xuanyuan3-70B | 12.14 | 19.69 | 15.41 | 80.89 | 86.51 | 83.90 | 29.83 | 52.62 | 37.33 | 46.48 |
| Llama-3.1-8B-Instruct | 73.12 | 22.09 | 2.91 | 77.42 | 76.18 | 69.09 | 29.00 | 54.21 | 36.56 | 48.95 |
| Llama-3.1-70B-Instruct | 16.26 | 34.25 | 56.34 | 80.64 | 79.97 | 86.90 | 33.33 | 62.16 | 45.95 | 55.09 |
| Llama 4 Scout | 73.60 | 45.80 | 44.20 | 85.02 | 85.21 | 92.32 | 25.60 | 55.76 | 43.00 | 61.17 |
| DeepSeek-V3 (671B) | 74.34 | 61.82 | 72.60 | 86.54 | 91.07 | 98.11 | 32.67 | 55.73 | 71.24 | 71.57 |
| DeepSeek-R1 (671B) | 80.36 | 64.04 | 75.00 | 81.96 | 91.44 | 98.41 | 39.67 | 55.13 | 71.46 | 73.05 |
| DeepSeek_R1_14B_Instruct | 71.33 | 44.35 | 50.45 | 81.96 | 85.52 | 92.81 | 39.50 | 50.20 | 52.76 | 59.49 |
| DeepSeek_R1_32B_Instruct | 73.68 | 51.20 | 50.86 | 83.27 | 87.54 | 97.81 | 41.50 | 53.92 | 56.80 | 66.29 |
| *Our LLMs* | | | | | | | | | | |
| DeepSeek_R1_14B_SFT | 80.63 | 51.67 | 52.61 | 83.53 | 89.05 | 96.72 | 36.68 | 50.43 | 50.85 | 65.36 |
| 14B $\Delta_{\text{Instruct}}$(SFT) | +9.25 | +7.28 | +2.19 | +1.53 | +3.42 | +3.85 | -2.93 | +0.24 | -1.92 | +5.83 |
| DeepSeek_R1_14B_SFT_GRPO | 81.46 | 52.41 | 53.47 | 83.57 | 89.02 | 95.58 | 36.83 | 54.06 | 51.24 | 66.92 |
| 14B $\Delta_{\text{Instruct}}$(SFT_GRPO) | +10.03 | +8.09 | +2.91 | +1.64 | +3.45 | +2.63 | -2.74 | +3.82 | -1.53 | +7.46 |
| 14B $\Delta_{\text{SFT}}$(SFT_GRPO) | +0.82 | +0.85 | +0.81 | +0.06 | 0.00 | -1.23 | +0.25 | +3.63 | +0.42 | +1.53 |
| DeepSeek_R1_32B_SFT | 80.45 | 66.42 | 63.28 | 86.88 | 88.43 | 93.76 | 46.05 | 55.27 | 68.41 | 70.08 |
| 32B $\Delta_{\text{Instruct}}$(SFT) | +6.75 | +15.23 | +12.37 | +3.64 | +0.83 | -4.14 | +4.52 | +1.25 | +11.63 | +3.75 |
| DeepSeek_R1_32B_SFT_GRPO | 80.67 | 66.83 | 64.45 | 86.79 | 88.52 | 91.26 | 45.68 | 54.83 | 67.75 | 70.44 |
| 32B $\Delta_{\text{Instruct}}$(SFT_GRPO) | +6.95 | +15.62 | +13.57 | +3.55 | +0.93 | -6.64 | +4.13 | +0.85 | +10.93 | +4.12 |
| 32B $\Delta_{\text{SFT}}$(SFT_GRPO) | +0.23 | +0.42 | +1.23 | -0.06 | +0.13 | -2.53 | -0.42 | -0.43 | -0.72 | +0.33 |

<think>
股票 000555.SZ 在 5 月 29 日大涨 10%，5 月 30 日继续高开高走并收在 12.09 元，成交量放大到 1 亿以上，显示出明显的资金追捧。新闻面上，数字货币板块整体强势，相关概念股集体涨停，比特币站上 11 万美元，叠加香港稳定币监管利好，市场情绪偏强。

技术面上，股价两日快速拉升，短期涨幅较大，可能引发部分获利回吐压力；但成交量和资金面支撑仍然较强。

因此，支持上涨的证据主要是：数字货币强势题材、比特币大涨、板块联动、成交量激增；支持下跌的证据是：短期累积涨幅过大，存在调整压力。综合判断，下一个交易日（6 月 3 日）开盘价相对前一交易日收盘价大概率小幅高开，涨跌幅在 1% 左右，不太可能继续大幅上冲或深度回落。属于震荡区间。
</think>

<score>[8, 5]</score>
<pct_change>0.0120</pct_change>
<answer>hold</answer>

GPT-5

<think>
基于提供的所有信息，我对神州信息（000555.SZ）下一个交易日（2025-06-03）的开盘价相对于当前交易日（2025-05-30）收盘价的涨跌进行推理：

1. **支持上涨的证据**：
   – 新闻面利好：数字货币概念股近期集体大涨，香港稳定币政策进展，比特币突破11万美元，神州信息作为A股稳定币概念股直接受益。公司近期举办论坛发布"乾坤"数智底座白皮书，与华为、腾讯云等合作成立生态体系，这些积极消息可能提升市场信心。
   – 技术面偏向：价格趋势向上，均线呈多头排列，5月29日和30日成交量显著放大（5月30日成交量1.03亿股），价量配合良好，MACD金叉，RSI为54.43未超买，显示短期动能强劲。
   – 资金面支持：融资买入增加（5月29日融资买入6044.86万元），融资余额占流通市值4.39%超过历史60%分位，表明投资者情绪乐观。
   – 板块效应：宏观环境显示资金流入科技板块，市场情绪乐观，数字货币和金融科技热点可能延续。

2. **支持下跌的证据**：
   – 基本面较弱：公司2024年预计净利润亏损，盈利能力差（销售净利率-5.28%），现金流为负，估值偏高（市净率2.1253），基本面支撑不足。
   – 短期涨幅较大：5月29日股价大涨10.04%，5月30日继续上涨0.25%，可能面临获利回吐压力。
   – 类似股票回调：参考相似股票（如300079.SZ、300479.SZ）在5月30日普遍下跌，可能带来联动回调风险。

3. **综合权衡**：
   – 正面因素占主导，尤其是新闻和技术面利好显著，但基本面疲软和短期涨幅可能限制上涨空间。预计下一个交易日开盘可能小幅高开，但涨幅不会过大，大概率在震荡范围内。
   – 预测涨跌幅为0.80%（0.0080），属于-3%到3%的震荡区间。

支持上涨证据评分8分，支持下跌证据评分3分。
</think>
<score>[8, 3]</score>
<pct_change>0.0080</pct_change>
<answer>hold</answer>

DeepSeek-R1

<think>
首先，我需要综合所有信息对神州信息（000555.SZ）下一个交易日（2025-06-03）的开盘价相对于2025-05-30收盘价的涨跌幅进行预测。

1. **当前价格与技术面分析**：
   – 2025-05-30收盘价为12.09元。
   – 5月29日股价涨停（+10.04%），5月30日高开至12.79元（+5.8%），但收盘仅微涨0.25%，收于12.09元，说明高位有明显抛压。
   – 成交量在5月29日（6911万）和5月30日（1.03亿）显著放大，显示短期情绪高涨但分歧加大。
   – 技术指标方面，MACD金叉、RSI=54.43（未超买），短期趋势偏多，但放量滞涨可能预示短期回调。

2. **新闻与事件驱动**：
   – 5月29-30日密集发布利好：数字货币概念集体大涨（公司为概念股）、"乾坤"数智底座白皮书发布、与华为/腾讯/阿里共建AIGC金融生态、成立金融科技大模型实验室等。
   – 香港稳定币条例通过、比特币突破11万美元，利好数字货币板块。
   – 融资余额上升至5.15亿元（历史60%分位），显示杠杆资金看多。

3. **基本面矛盾**：
   – 公司2025Q1净利润为负（-0.10元/股），经营现金流为负，ROE为负，基本面承压。
   – 但市场更关注题材和事件驱动，尤其在AI+金融、数字货币等热点加持下，短期情绪可能忽略基本面。

4. **同类表现**：
   – 同为数字货币/金融科技股的300079.SZ、300479.SZ、300541.SZ在5月29日均大涨（+4.97%~+8.28%），但5月30日普遍回调（-0.02%~-2.81%），显示板块短期获利回吐压力。
   – 神州信息5月30日仅微涨0.25%，相对抗跌，可能保留部分上行动能。

5. **假期与时间因素**：
   – 5月30日（周五）收盘后至6月3日（周二）开盘有3天间隔（含周末），期间无重大宏观数据发布，但需考虑周末情绪发酵。
   – 市场整体情绪偏乐观（A股医疗/科技板块资金流入），数字货币板块热度仍在。

6. **综合判断**：
   – 利好密集且属当前热点（AI+金融+数字货币），短期情绪支撑较强。
   – 但技术面出现放量滞涨，叠加同类股回调，预计6月3日开盘可能高开但幅度有限，或小幅震荡。
   – 考虑到5月30日已消化部分利好，且基本面不佳，大幅高开（>3%）可能性低。
   – 预计开盘涨幅在0.5%~1.5%之间，取中值约0.95%。

证据评分：
– 支持上涨（a）：多重利好（数字货币政策、AI合作、论坛成果）、融资买入增加、板块热度高 → 8.5分。
– 支持下跌（b）：基本面亏损、技术面滞涨、同类股回调、高换手后抛压 → 6.0分。

涨跌幅预测：0.0095（0.95%），在-3%~3%之间，故方向为hold。
</think>
<score>[8.5, 6.0]</score>
<pct_change>0.0095</pct_change>
<answer>hold</answer>

Qwen3

Figure 36: Zero-shot responses. The model struggles to provide accurate and relevant answers without the step-by-step reasoning process enabled by CoT prompting.

2322
2323
2324
2325
2326
2327
2328
2329
2330
2331
2332
2333
2334
2335
2336
2337
2338
2339
2340
2341
2342
2343
2344
2345
2346
2347
2348
2349
2350
2351
2352
2353
2354
2355
2356
2357
2358
2359
2360
2361
2362
2363
2364
2365
2366
2367
2368
2369
2370
2371
2372
2373
2374
2375

<think>
我们要预测 000555.SZ（神州信息）在 2025-06-03（周二）的开盘价相对 2025-05-30 收盘价的涨跌。

**第一步：理解个性**
000555.SZ 属于金融科技与数字货币概念股，叠加 AI 大模型、国产云金融等热点。它是政策驱动型、消息敏感型成长股，而非稳定的蓝筹股，因此更容易受短期利好消息带动。

**第二步：时间特征**
预测日期是周五收盘后的下一个交易日，即 2025-06-03 周二开盘，中间隔了一个周末和端午节前市场氛围。A 股在周末/节日前容易出现资金调仓效应，但热点题材往往在节后延续。

**第三步：市场状态信息**
新闻重点：

* 数字货币概念大涨（海联金汇、翠微股份涨停，比特币突破 11 万美元）。
* 香港稳定币立法，稳定币概念被市场看好。
* 神州信息本身在 5 月底主办金融论坛，发布"乾坤"数智底座，与华为、腾讯云、阿里云合作，成立大模型生态联盟。
  这些消息直接关联神州信息，属于实质性利好。

**第四步：构建分析逻辑**

* 消息面：强利好，板块共振，叠加公司自身利好。
* 技术面：股价近期在 11 元上下震荡，带有企稳迹象，5 月 30 日收盘在 11 元附近，成交量较低但题材可能带来放量高开。
* 资金面：数字货币与大模型双重热点，容易吸引短线资金。
* 风险面：利好消息已集中释放，部分涨幅可能提前兑现，但鉴于比特币创新高与政策支撑，预期仍偏正向。

**第五步：证据分类**
支持上涨：

* 数字货币板块集体大涨，正面传导（9/10）。
* 比特币突破 11 万美元，全球市场风险偏好提升（8/10）。
* 香港稳定币立法，长期利好，短期预期提升（7/10）。
* 神州信息自身金融论坛、大模型生态消息，直接公司利好（8.5/10）。
* 技术面企稳，存在反弹动力（6.5/10）。

支持下跌：

* 前期震荡，利好或被提前炒作，存在高开低走风险（5/10）。
* 假期效应资金可能偏谨慎（4/10）。
* 行业题材易受政策与市场情绪波动影响（3.5/10）。

**第六步：综合**
支持上涨平均：(9 + 8 + 7 + 8.5 + 6.5) / 5 ≈ 7.8
支持下跌平均：(5 + 4 + 3.5) / 3 ≈ 4.2

**第七步：未来推演**
由于比特币创新高、板块共振，以及公司自身热点叠加，短期资金会倾向拉升，开盘预期高开 3%~4%。

所以，最终判断：上涨，高开幅度约 +3.4%。方向归类为 up（显著上涨）。
</think>

<score>[7.8, 4.2]</score>
<pct_change>0.0340</pct_change>
<answer>up</answer>

GPT4o

<think>
作为金融分析师，我将严格按照指定的思维范式进行推理。

1. **理解股票个性**：000555.SZ（神州信息）属于IT服务、数字货币、信创、AI、华为概念等多重热门题材的成长型科技股。公司并非传统蓝筹，而是具有高波动性、事件驱动型特征的科技成长股。其股价对政策、技术发布、行业合作等消息高度敏感。

2. **理解时间特征**：当前交易日为2025-05-30（周五），下一个交易日为2025-06-03（周二），中间间隔一个周末（无重大节假日）。通常，周末可能积累情绪，但无长假效应，因此开盘主要受节前最后一个交易日及周末新闻影响。

3. **信息维度分析**：
   – **新闻面**：近期密集发布重大利好。5月29–30日，"数云原力2025"论坛召开，神州信息发布"乾坤"数智底座、成立AIGC金融生态、与华为/腾讯/阿里等共建联盟、与青岛银行设实验室、牵头金融安全国际标准。同时，数字货币板块整体大涨（5月30日多只个股涨停），直接受益于香港《稳定币条例草案》通过及比特币突破11万美元。
   – **技术面**：5月29日放量涨停（+10.04%），5月30日高开后震荡收涨0.25%，成交量继续放大至10.6%换手率，量价配合良好。均线呈多头排列，MACD金叉，RSI 54.43未超买，技术形态强势。
   – **资金面**：5月29日融资买入6044万元，占当日流入12.65%，融资余额5.15亿元（历史60%分位），显示杠杆资金积极介入；融券余额低，空头力量弱。
   – **基本面**：公司2025Q1仍亏损（每股–0.10元），经营现金流为负，基本面承压。但市场当前更关注其在AI、数字货币、信创等新质生产力领域的布局，而非短期盈利。
   – **板块联动**：同属数字货币/IT服务板块的可比个股（如300079、300479、300541）在5月29日均大幅上涨（+4.97%、+5.99%、+8.28%），5月30日虽有回调，但整体板块热度仍在。

4. **动态构建分析逻辑**：
   – 对于成长型科技股，在重大利好事件密集发布+板块集体异动+技术突破+资金流入的背景下，短期情绪驱动往往压倒基本面。
   – 需权衡：利好是否已被5月29–30日涨幅充分定价？从5月30日走势看，高开低走但收盘仍站稳12元（仅微跌0.25% from high），说明承接力强，未出现利好兑现式大跌。
   – 周末无负面新闻，反而因6月2日仍有媒体重申论坛成果，情绪持续发酵。

5. **证据分类与评分**：
   **支持上涨证据**：
   – 数字货币政策利好（香港稳定币立法+比特币新高）：直接关联神州信息数字货币业务，评9分。
   – 公司重大产品发布与生态合作（乾坤底座、AIGC联盟、华为联合智能体等）：提升技术壁垒与市场地位，评9.5分。
   – 技术面强势（放量突破、均线多头、MACD金叉）：短期动能充足，评9分。
   – 融资资金大幅流入、融券压力小：杠杆资金看多，评8.5分。
   – 板块联动效应强（可比个股亦大涨）：形成板块合力，评8.5分。
   → 综合支持上涨评分：(9+9.5+9+8.5+8.5)/5 ≈ 8.9 → **9.0**

   **支持下跌证据**：
   – 基本面持续亏损、现金流为负：长期隐忧，但短期影响弱，评4分。
   – 5月29–30日已累计大涨约12%，存在技术性回调压力：但量能健康，评5分。
   – 市场整体聚焦医药板块（宏观报告），科技股可能轮动放缓：但数字货币属独立热点，评3分。
   → 综合支持下跌评分：(4+5+3)/3 ≈ 4.0 → **4.0**

6. **假设检验与反事实推演**：
   – 若无数字货币利好，仅靠AI发布，涨幅可能有限 → 但数字货币是当前最强主线，两者叠加形成共振。
   – 若5月30日收长上影线，则可能回调 → 但实际收盘接近当日均价，多头控盘。
   – 若周末出现监管利空 → 但实际无负面，反而有正面重申。
   → 所有路径均指向短期继续上行。

7. **涨跌幅预测**：
   – 参考历史：5月29日涨停后，5月30日开盘溢价6.5%（12.79/12.06–1），但全天震荡收平。
   – 下一交易日（6月3日）开盘溢价应小于5月30日，因部分利好已兑现，但板块热度仍在。
   – 合理开盘涨幅：2.5%–3.5%。取中值3.0%，即0.0300。
   – 方向判断：3.0% → **up**

最终结论：情绪与技术面主导，短期触发"显著上涨"。
</think>

<score>[9.0, 4.0]</score>
<pct_change>0.0300</pct_change>
<answer>up</answer>

Qwen3

Figure 37: CoT Prompting responses. The model demonstrates improved reasoning and accuracy when guided through a step-by-step thought process.

2376
2377
2378
2379
2380
2381
2382
2383
2384
2385
2386
2387
2388
2389
2390
2391
2392
2393
2394
2395
2396
2397
2398
2399
2400
2401
2402
2403
2404
2405
2406
2407
2408
2409
2410
2411
2412
2413
2414
2415
2416
2417
2418
2419
2420
2421
2422

<think>
### 推理过程

1. **市场环境分析**:
   – **宏观环境**: A股市场近期资金流入积极,尤其是医疗健康和科技板块。市场情绪乐观,但外部经济环境存在不确定性。
   – **行业动态**: 数字货币概念股表现强劲,受政策利好推动。稳定币在香港的进展和比特币价格上涨对市场有正向刺激作用。
   – **公司新闻**: 神州信息发布了"乾坤"企业级数智底座白皮书,与华为等合作推出金融知识问答联合智能体,并成立金融科技大模型联合实验室。这些进展显示公司在金融科技领域的领先地位和创新能力。

2. **技术面分析**:
   – **趋势特征**: 均线多头排列,显示短期趋势向上。
   – **量能分析**: 成交量显著放大,价量关系正相关,显示资金流入积极。
   – **指标信号**: MACD金叉,RSI未超买,趋势健康。

3. **基本面分析**:
   – **财务状况**: 尽管净利润亏损,但公司在技术研发和市场拓展方面有积极进展,未来盈利能力有望提升。
   – **估值水平**: 当前估值较低,市盈率为负,可能反映市场对未来的悲观预期,但随着业务改善,估值有上升空间。

4. **综合评估**:
   – **利好因素**: 政策支持、行业领先、技术突破、资金流入。
   – **风险因素**: 历史亏损、高负债、现金流压力。

### 证据评分

– **支持上涨的证据**:
   1. **行业利好政策**: 数字货币和AI大模型的应用推动行业发展,评分: 9/10。
   2. **技术突破和新品发布**: 发布"乾坤"平台和AIGC生态体系,提升竞争力,评分: 8/10。
   3. **资金流入和市场情绪**: 融资买入增加,市场情绪乐观,评分: 8/10。
   4. **量价配合**: 成交量放大,价量正相关,评分: 7/10。
   5. **均线多头排列**: 显示短期趋势向上,评分: 7/10。

– **支持下跌的证据**:
   1. **历史亏损**: 公司过去净利润亏损,评分: 6/10。
   2. **高负债和现金流问题**: 负债率较高,现金流净额为负,评分: 7/10。
   3. **行业竞争加剧**: 科技和金融板块竞争激烈,评分: 6/10。
   4. **政策不确定性**: 全球经济和贸易政策的不确定性,评分: 6/10。

### 涨跌幅预测

综合考虑利好因素和技术面的积极信号,预计下一个交易日的开盘价将上涨约1.50%。

### 方向预测

预计涨幅在3%以内,属于震荡区间。
</think>
<score>[10.0, 6.0]</score>
<pct_change>0.0150</pct_change>
<answer>hold</answer>

DeepSeek-14B

<think>
好的,我现在要预测神州信息(000555.SZ)在下一个交易日(即6月3日)的开盘价相对于5月30日的收盘价的涨跌幅。我的分析流程如下:

1. **理解所要分析的股票的个性**: 神州信息属于IT服务行业,业务涵盖金融科技、量子通信、数字货币等多个高科技领域。作为行业龙头,其股价波动受政策和技术突破双重影响。需要特别关注其在AIGC(生成式AI)领域的布局以及与华为等科技巨头的合作。

2. **理解所要预测的时间特征**: 当前处于5月底至6月初的时间窗口,属于半年报披露前的敏感期。同时临近周末,需警惕周五异动对周初情绪的影响,但周一作为新的交易周通常存在惯性上涨效应。

3. **查看所提供的市场状态**:
   – **宏观环境**: A股科技板块热度持续,但整体市场面临流动性压力。美债收益率上行压制风险偏好,需警惕外资流动对成长股估值的压制。
   – **基本面数据**: 公司2024年出现亏损,经营现金流持续为负,资产负债率超过58%。这些财务指标显示公司处于基本面修复初期,估值修复动能可能受制于盈利能力的不确定性。
   – **价格信息**: 5月30日收盘价12.09元,较前一日上涨0.0025,成交量激增至1.03亿股创年内新高,呈现量价齐升态势。技术面显示5日均线(11.97)与10日均线(11.78)形成多头排列,MACD红柱放大。
   – **融资融券数据**: 5月29日融资买入6044.86万元,占当日流入资金比例12.65%,融资余额5.15亿元触及历史60%分位。融券余额174.75万元处于低位,显示做空力量有限。
   – **新闻事件**: 连续三天发布与AI金融大模型相关的重磅合作,包括与华为联合发布智能体、发起行业首个AIGC生态体系等。这些事件形成持续催化效应,推升市场关注度。

4. **初步构建分析方法逻辑**:
   – 结合技术面量价配合与资金流入情况,判断是否存在突破性上涨动能。
   – 评估政策利好(香港稳定币法案)与基本面承压(亏损、现金流)的矛盾。
   – 关注相似股票(如300079.SZ、300479.SZ)的走势对板块情绪的传导作用。

5. **按分析方法逻辑分析各维度信息**:
   – **利多因素**:
     – 技术面: 5日量价齐升突破整理区间,MACD金叉形成多头趋势。
     – 政策支持: 香港稳定币法案推动数字货币板块整体走强,神州信息作为概念股核心受益。
     – 事件驱动: 连续三天发布AI金融大模型相关成果,与华为等巨头合作形成持续催化。
     – 融资盘介入: 融资余额突破历史60%分位,显示杠杆资金积极布局。
   – **利空因素**:
     – 基本面屏弱: 2024年亏损、现金流为负,经营效率低下制约估值修复空间。
     – 流动性压力: 美债收益率上行可能抑制成长股估值,板块轮动压力显现。
     – 技术面阻力: 12.09元附近面临12.10元整数关口压力,需观察量能能否持续放大。

6. **证据评分**:
   – **上涨支持因素**:
     1. 技术面量价配合(9分): 突破整理区间伴随放量,MACD金叉确认趋势。
     2. 政策利好催化(8分): 香港稳定币法案提升板块情绪,数字货币概念股普涨。
     3. 事件驱动持续性(8分): 三天连续发布AI金融成果,形成技术面多头氛围。
     4. 融资盘介入(7分): 融资余额突破历史分位,杠杆资金形成推力。
   – **下跌支持因素**:
     1. 基本面屏弱(7分): 亏损与现金流为负制约估值修复,需警惕技术性回调。
     2. 流动性压力(6分): 美债收益率上行可能引发板块资金撤离。
     3. 技术面阻力(6分): 12.10元关口存在压力,量能若无法持续可能引发获利了结。
     4. 融券余额低位(5分): 融券力量薄弱,空头打压动能不足。

7. **假设检验与推演**:
   – 如果政策利好持续发酵,AI金融大模型落地项目加速,股价可能突破12.10元阻力位。
   – 若市场对亏损业绩存在分歧,可能出现技术性回调至11.80元附近。
   – 考虑到相似股票(300079.SZ)在利好驱动下涨幅接近5%,板块联动效应可能放大个股涨幅。

8. **综合评分计算**:
   – 上涨平均分: (9+8+8+7)/4=8.0
   – 下跌平均分: (7+6+6+5)/4=6.0

9. **最终预测**:
   虽然基本面存在亏损隐患,但技术面量价配合与政策利好形成共振,且融资盘介入形成推力。考虑到周末消息面无重大利空,预计周一开盘将延续涨势,涨幅3%左右,方向为显著上涨。

</think>

<score>[8.0, 6.0]</score>
<pct_change>0.0300</pct_change>
<answer>up</answer>

DeepSeek-14B-SFT

2423
2424
2425

Figure 38: Case study on cold-started responses. Left is the original response, and right is the cold-started response. The cold-started response is more concise and to the point, avoiding unnecessary elaboration.
