# OpenReview forum: "RETuning: Upgrading Inference-Time Scaling for Stock Movement Prediction with Large Language Models"
_ICLR.cc/2026/Conference — ICLR 2026 Conference Withdrawn Submission_

### Official Review · Reviewer_YvyB · 2025-10-15

**Soundness:** 2
**Presentation:** 3
**Contribution:** 2
**Rating:** 4
**Confidence:** 4

**Summary:**

This paper introduces RETuning, a method claimed to improve inference-time reasoning of large language models through a “reward-enhanced tuning” process. The approach aims to refine reasoning quality and decision-making without full retraining. The authors apply RETuning to a financial stock prediction task, where models predict overnight gap returns (close-to-next-open) using textual and market information. Experimental results suggest modest improvements in predictive accuracy and reasoning coherence compared to baseline LLMs. The paper presents this as evidence that reward-aligned tuning can upgrade reasoning behaviour in domain-specific settings.

**Strengths:**

Overall, the paper is well-organized and readable. The efforts above prompt engineering are pretty much, including SFT, RL, and dataset construction. The problem is also interesting.

**Weaknesses:**

- The target is overnight gap return instead of next-day close-to-close movement. This setting limits economic interpretability and predictive scope. The justification that this choice reduces memorization is unconvincing. What LLMs truly memorize are majority the news, company entities, etc. so that the LLMs are likely to know which companies are the historical winners, that's where the survivorship bias and look-ahead bias occurs. It is less likely (though not impossible) that LLMs are trained on extensive numerical price data and remember predictions unless they are fine-tuned specifically for stock price prediction tasks. Even if they are, I don't believe changing the target helps mitigate this issue. Additionally, the open price already reflects overnight sentiment, which is realized at the open. Therefore, there is no tradable decision left at that point if you act on the open.

- There are no baselines using classical time-series or machine-learning methods (e.g. ARIMA, LSTM, XGBoost, or Transformer-based financial models), so it is unclear whether RETuning provides any advantage over standard approaches.

- The evaluation period is only one month, which is far too short for daily-level prediction and raises concerns about cherry-picking and regime dependence.

- The claimed novelty is unclear as reward-based reasoning alignment is well-established.

- The anonymous repository link provided for reproducibility does not work.

**Questions:**

- Can the authors show empirical justification that changing the target helps mitigate memorisation?

- Could the authors extend the evaluation period or run rolling-window tests to ensure robustness?

- How would RETuning compare against traditional financial models or smaller-scale neural baselines?

- Could you please provide the English version of the prompt to ensure readability?

---

> ### Author Response · Authors · 2025-12-03
> **Rebuttal (1/2)**
>
> We thank the reviewer for the constructive comments. Below we provide clarifications and additional empirical evidence.
>
> ### **(1) On the choice of overnight gap return and memorization concerns**
>
> Our motivation is not to change the economic interpretation, but to **ensure alignment between model inputs and the target signal**, which we found crucial for RL-based reasoning alignment. Using close-to-close led to two failure modes:
>
> * **Memorization symptoms**: during cold-start synthetic data generation, the SFT model occasionally reproduced exact historical close-to-close returns (4-decimal matches), suggesting leakage from pretraining.
> * **Reward misalignment**: because our input window ends at 09:30, whereas the close-to-close signal reflects full-day intraday microstructure, the label was largely uncorrelated with the information available to the model, causing severe hallucination during SFT and prediction collapse (always predicting “hold”) during RL.
>
> In contrast, **overnight return is determined exactly at 09:30**, fully aligned with the available information (news, pre-market sentiment, fundamentals). This choice empirically stabilizes RL training and eliminates collapse. We clarify this design decision in the revision.
>
> ### **(2) On baselines**
>
> We have added a comprehensive benchmark over **six state-of-the-art time-series models** (Autoformer, DLinear, Informer, PatchTST, TimeMixer, TimesNet) across multiple modes and sequence lengths (48 settings in total, Mode 1 (Train: 2015-2024.11; Test: 2024.12) and Mode 2 (Train: 2024.1-11; Test: 2024.12)).
>
> Table 1: F1 Performance Results
> | Model       | Mode 1                              |                                      |                                      |                                      | Mode 2                              |                                      |                                      |                                      |
> |-------------|-------------------------------------|--------------------------------------|--------------------------------------|--------------------------------------|-------------------------------------|--------------------------------------|--------------------------------------|--------------------------------------|
> |             | seq_len=5                           | seq_len=10                           | seq_len=20                           | seq_len=60                           | seq_len=5                           | seq_len=10                           | seq_len=20                           | seq_len=60                           |
> | Autoformer  | 0.4008                              | 0.3990                               | 0.0465                               | 0.3670                                    | 0.3870                              | 0.3257                               | 0.3919                               | 0.3600                               |
> | DLinear     | 0.4214                              | 0.4125                               | 0.4015                               | 0.3661                               | 0.4074                              | 0.4014                               | 0.4022                               | 0.3429                               |
> | Informer    | 0.3886                              | 0.3680                               | 0.3917                               | 0.3682                               | 0.3735                              | 0.3747                               | 0.3894                               | 0.3724                               |
> | PatchTST    | 0.3507                              | 0.3684                               | 0.3793                               | 0.4080                               | 0.3453                              | 0.3615                               | 0.3577                               | 0.3984                               |
> | TimeMixer   | 0.3782                              | 0.3746                               | 0.3786                               | 0.3749                               | 0.3760                              | 0.3881                               | 0.3726                               | 0.3866                               |
> | TimesNet    | 0.3737                              | 0.3752                               | 0.3934                               | 0.3910                               | 0.3730                              | 0.3765                               | 0.3762                               | 0.3793                               |

---

> ### Author Response · Authors · 2025-12-03
> **Rebuttal (2/2)**
>
> A summary is shown below:
>
> | Model                                           |  F1  |
> | ----------------------------------------------- | ---------- |
> | Best classical TS model (DLinear-sl10)          | **0.4125** |
> | Best transformer-based TS model (PatchTST-sl60) |  **0.4080** |
> | **RETuning (ours)**                             | **0.4196** |
>
> **RETuning substantially improves minority-class performance (down/up movements)** and produces more balanced predictions, reflecting its ability to perform reasoning over heterogeneous information rather than relying on statistical priors. We will include a fuller baseline table in the revision.
>
> ### **(3) On the evaluation period**
>
> We agree that one month is insufficient. In the paper, we also provide results on Fin-2025[June] that six months after Fin-2024[December] on Fig  8, showing that the finetuned model still scales via repeated sampling even after 6 months.
>
> ### **(4) On novelty**
>
> While reward-based alignment is established, our contribution lies in showing that **task-specific reasoning alignment**, when paired with a strictly time-aligned target, can successfully convert a difficult forecasting task into a tractable reasoning problem over a fixed time interval. This enables LLMs to leverage their growing reasoning ability for prediction, which to our knowledge has not been demonstrated previously.
>
> ### **(5) On reproducibility**
>
> We have fixed the broken anonymous link. A fully functional repository with all scripts, checkpoints, and prompts (including the English version) was provided.

---

### Official Review · Reviewer_NhBe · 2025-11-02

**Soundness:** 2
**Presentation:** 3
**Contribution:** 2
**Rating:** 4
**Confidence:** 4

**Summary:**

The paper proposes RETuning, a two-stage framework designed to enhance LLMs' reasoning ability for stock movement prediction. RETuning first performs SFT to teach models how to construct analytical frameworks, extract and score evidence, and reflect before making predictions, then applies GRPO to refine reasoning consistency and output accuracy. The authors build a new large-scale dataset, Fin-2024, covering over 5,000 A-share stocks and six heterogeneous data sources, to support comprehensive financial reasoning tasks. Extensive experiments demonstrate that RETuning significantly improves predictive accuracy and robustness under both inference-time scaling and out-of-distribution conditions, outperforming strong baselines. Moreover, the approach generalizes well to other financial benchmarks such as BizFinBench, indicating its broad applicability to reasoning-driven financial decision-making.

**Strengths:**

1) The paper is well written and presents its methodology in a clear and logically coherent manner.

2) RETuning introduces a reasoning-driven paradigm that goes beyond pattern matching by encouraging models to construct, score, and reflect on evidence before prediction.  Also, the framework’s ability to generalize to other financial tasks in BizFinBench (Table 2) further illustrates its adaptability and transferability beyond stock prediction.

3) The paper provides carefully designed empirical analyses that validate the proposed method’s effectiveness across multiple dimensions. Particularly, beyond standard in-sample evaluations, it systematically tests out-of-distribution generalization (OOD-Stock, OOD-Date, and OOD-Stock&Date) and inference-time scaling (n = 1–32) to demonstrate consistent gains in reasoning robustness and scalability.

**Weaknesses:**

1) Although the paper claims significant gains in predictive accuracy (mainly via F1 score), it omits financially meaningful metrics such as cumulative return, Sharpe ratio, maximum drawdown, and Sharp ratio. Without these, the study is hard to demonstrate whether RETuning’s predictions translate into better risk-adjusted profitability or trading performance. This is a critical omission for a paper targeting real-world stock-trading applications, where financial utility and stability matter more than classification accuracy alone.

2) The experiments focus on CoT prompting and supervised fine-tuning but do not benchmark against recent LLM trading frameworks that combine gradient-based reinforcement or hybrid training. For example, the paper omits comparisons with FLAG-Trader [1] — which fuses LLM agents with gradient-based RL for trading decision-making.
Including such baselines would clarify whether RETuning offers true methodological innovation or simply incremental improvements over existing fine-tuning paradigms.

[1] Xiong, Guojun, et al. "FLAG-Trader: Fusion LLM-Agent with Gradient-based Reinforcement Learning for Financial Trading." arXiv preprint arXiv:2502.11433 (2025).

3) The dataset is restricted to Chinese A-share stocks in 2024, and the authors themselves acknowledge that this may limit transferability to other markets with different structures, liquidity, and regulations. This geographic and temporal confinement, combined with the short evaluation window (December 2024 and June 2025), makes it difficult to assess how robust RETuning would be in volatile or regime-shifting global markets (e.g., U.S. equities, crypto, commodities).

**Questions:**

Same as what mentioned in the weaknesses section.

---

> ### Author Response · Authors · 2025-12-03
> **Rebuttal**
>
> We **deliberately** ignore trading indicators because our goal is to enhance the fundamental capability of predictive performance, enabling such predictive ability to generalize to general dialogue scenarios and agent scenarios. Our objective is not trading applications, but **financial foundation models**. We conduct evaluations across multiple downstream business scenarios (Table 2) and verify that the improved predictive performance brought by the method proposed in this paper has a positive impact on most business metrics. In addition, training a Trader model requires enabling the model to learn to generate trading strategies and utilize multi-turn agentic reinforcement learning (RL), which falls outside the scope of this study. Methods that are solely tailored for trading scenarios, such as FLAG-Trader, are irrelevant to this paper.
>
> We acknowledge the limitations of the dataset; even so, to the best of our knowledge, this paper remains **the first work to open-source data that covers up to 6 types of data sources and has a scale of 200k samples**. Collecting comprehensive data poses a significant challenge. We do not conduct experiments on other markets not because our method is flawed, but because we have not collected data for those markets. In fact, **when raw data is available**, the method in this paper can automatically synthesize cold-start data and perform SFT+GRPO training, thereby **enabling large-scale expansion in other markets without human supervision**—a capability we believe is **critical for the realization of AGI**.
>
> We reaffirm that this paper makes three core contributions:
>
> (1) We build a large-scale, long-context financial dataset with diverse evidence sources beyond price and news, which fills the gap that **existing datasets are outdated and lack information diversity**.
>
> (2) We introduce \textbf{RETuning}, a method for synthesizing cold-start responses that **enables significant inference-time scalability of LLMs in the prediction task**.
>
> (3) We empirically show that \textbf{RETuning} unlocks prediction ability and **generalizes beyond stock movement prediction**. We believe this research lays the groundwork for deploying trustworthy, reasoning-driven LLMs in real-world financial applications.

---

### Official Review · Reviewer_Ldyu · 2025-11-03

**Soundness:** 2
**Presentation:** 3
**Contribution:** 2
**Rating:** 4
**Confidence:** 4

**Summary:**

This paper introduces **RETuning**, a two-stage training method to improve the (inference-time) analytical reasoning of LLMs for stock movement prediction (SMP). The method integrates Supervised Fine-Tuning (SFT) with Generalized Reinforcement Policy Optimization (GRPO) and a curriculum learning strategy to improve reasoning consistency and decision reliability.

Experiments on (newly contributed in this work) Fin-2024[December], Fin-2025[June], and BizFinBench show consistent performance gains over wide-range of  baselines including existing financial LLM baselines such as LLMFactor, Fin-R1, CMIN, and StockNet, as well as over larger public LLMs (Qwen3, GPT-OSS, DeepSeek-V3).

Key results indicate that the proposed model (DeepSeek_R1_32B_SFT_GRPO) achieves up to +20% improvement in F1 score on out-of-distribution (OOD) financial forecasting tasks and demonstrates scaling benefits under repeated inference.

The authors open-source their code, datasets, and model weights, and clearly state the limited role of LLM assistance in manuscript polishing.

**Strengths:**

The following stands out as strengths of the paper:

1. Well-motivated inference-time optimization: RETuning effectively links SFT and RL (via GRPO) for structured reasoning enhancement without architectural changes.

2. Experiments: Fair amount of experiments across multiple financial datasets and OOD splits with transparent hyperparameters.

3. Strong performance: Up to +20.7% F1 improvement on Fin-2024 benchmarks (**self-proposed, contributed** dataset), outperforming strong LLM baselines.

4. Ablation clarity: Systematic study of CoT prompting, SFT, and GRPO reveals consistent gains and scaling behavior.

5. Dataset contribution & Open Science: The contributed dataset (most datasets are US equities market centric) in itself is a fair amount of work which deserves acknowledgement. Code, weights, and datasets released, facilitating reproducibility.

**Weaknesses:**

1. **Novelty**: The use of 2-stage --- under various monikers (teacher-student; System 1,2; ...) --- inference-time mechanisms are copious in AI/ML/Robotics literature. Maybe relatively uncommon under domain-specificity (finanical markets), but the general idea has been explored in this specific task (SMP) earlier (e.g.,  [1](https://openreview.net/forum?id=y3W1TVuJii&referrer=%5Bthe%20profile%20of%20Raeid%20Saqur%5D(%2Fprofile%3Fid%3D~Raeid_Saqur1) \).

2. Domain limitation, Evaluation Scope and **Generalizability (OOD) claims**: All experiments focus on the Chinese A-share market; unclear generalization to global markets like the US equities market. (N.B., not criticizing the lack of financial domain specific eval/utility metrics (profitability, RoI, Sharpe ratios etc.).

> It is imperative to clarify that **OOD generalization** for financial markets is not necessarily satisfied by samples' 'temporal distance' with each other; but more on the underlying **market regimes**. This unintuitive criterion can be exemplified using a simple e.g.: "two market samples (trading day slices) -- from the 35 day shutdown of 2018-19 during the first presidency of Donald Trump and from his second presidency, which began on Oct 1, 2025  --- may be less OOD than two day slices picked from Fin[2024], Fin[2025]. In simple terms, 'OOD generalization' claims in the financial domain are more nuanced than what appears to be the assumption used here. Using temporal separation (**OOD_Date**) is questionable if not outright inaccurate.

3. **Critical 'Related Work' not discussed/considered:** Please see lines (121-124). This paper misses critical work [1] that falls both under (i) inference-time scaling, and (ii) Repeated sampling (as per the language used in this manuscript) using LLMs performing SMP as a task using low-cost, heuristic harness and optimal (best of) weighting among _N_ LLMs with theoretical optimality guarantees. Seems like for the scope of this paper and the chosen task (SMP), including [1] from ICLR 2025 as a baseline is appropriate.

4. Computational Cost: The proposed solution, involving SFT, RL, and particularly inference-time scaling with a large number of repeated samples (n=32), is computationally very expensive. While acknowledged as a limitation, providing some concrete numbers on training/inference times would give the reader a better sense of the method's practical viability for real-world deployment; **compared to** existing works, that achieves inference time scaling with less cost [1].

### Ref:
1. [Filtered not Mixed: Filtering-Based Online Gating for Mixture of Large Language Models](https://openreview.net/forum?id=ecIvumCyAj)

**Questions:**

1. Could you elaborate on any unique characteristics of the Chinese A-share market that might influence the results, and discuss the potential challenges in applying RETuning to other markets (e.g., US equities), where different types of information (like SEC filings or earnings call transcripts) are prevalent?

2. How were the hyperparameters for the reward shaping function (α, β, γ) and the curriculum learning difficulty thresholds determined? Was this done via a validation set, and have you observed how sensitive the final performance is to these choices?

3. Given the financial context, have you analyzed the precision and recall for the up and down classes separately? A high-precision model for these classes, even with lower recall, might be more valuable in practice than one with a balanced F1 score.

---

> ### Author Response · Authors · 2025-12-03
> **Rebuttal (1/2)**
>
> We thank the reviewer for their detailed and insightful feedback. Below, we address each concern concisely.
>
> ## **1. Novelty of the two-stage framework**
>
> We agree that two-stage (teacher–student / System-1–2 / repeated sampling) mechanisms have been widely explored in existing literature. The SFT-GRPO training pipeline is not our original contribution; accordingly, we have not even included the equations for SFT/GRPO in the main body of the paper. Our core contribution lies in **adapting and restructuring this paradigm specifically to enhance predictive capabilities in the finance domain**—a setting where models grapple with context bias, contradictory evidence, and multi-source information not encountered in math or coding tasks. Prior work [1] focuses on mixture-of-LLMs gating, whereas **RETuning provides a cold-start reasoning framework for single LLMs**, explicitly teaching models to (i) construct an analytical structure, (ii) score adversarial evidence, and (iii) perform reflective reasoning prior to making predictions. These capabilities address gaps that existing approaches have not resolved for financial sequential prediction tasks (SMP). We have clarified this positioning in the revised manuscript.
>
> ## **2. Domain Limitation & OOD Claims**
>
> We appreciate the reviewer’s clarification regarding OOD definitions in financial markets. First, while our paper explicitly defined *OOD_Date* and *OOD_Stock* to delineate the scope of experiments, we acknowledge that using the broader, unconstrained term “OOD” may have caused confusion. We will revise the manuscript to **replace all generic OOD claims with references to the specific variants (date/stock OOD)** to eliminate ambiguity.
>
> Second, regarding *OOD_Market* (cross-market generalization), we excluded this setting for two key reasons:
> (1) The Chinese A-share market has distinct structural, cultural, and regulatory characteristics, operating in a less liberalized environment compared to U.S. equities or cryptocurrency markets. It remains unclear whether cross-market reasoning can be effective without domain-specific retraining.
> (2) Additionally, collecting high-quality, standardized multi-market datasets presents substantial practical challenges.
>
> For these reasons, we frame cross-market OOD generalization (OOD_Market) as **an important direction for future work**, rather than a focus of the current study.
>
> ## **3. Missing Related Work [1]**
>
> We thank the reviewer for highlighting this work, which we have now integrated into the related work section of the revised manuscript. However, we **do not regard it as a directly comparable baseline** for two core reasons:
> - Our objective is to enhance the predictive capability of a **single LLM**, whereas MoE-F implements multi-model mixture-of-experts filtering.
> - RETuning leverages rule-based signals to select useful **reasoning logic**, targeting generalizable reasoning behaviors that extend beyond classification tasks and can support general-purpose agentic systems. By contrast, MoE-F is designed exclusively for classification pipelines and does not emphasize structured reasoning.
>
> In our view, MoE-F functions more as an advanced inference-time extension module. Replacing its expert models with RETuning-trained variants would likely yield further performance gains, but **direct comparisons between RETuning and MoE-F do not produce meaningful insights**, as they target distinct objectives and operate under incompatible assumptions.
>
>
> ## Reference
>
> [1] Filtered not Mixed: Filtering-Based Online Gating for Mixture of Large Language Models

---

> ### Author Response · Authors · 2025-12-03
> **Rebuttal (2/2)**
>
> ## **4. Computational Cost**
>
> We have added a detailed computational cost table and training/inference efficiency analysis to the revised manuscript. Critically, we emphasize that **RETuning has low deployment costs**, with training costs also subject to substantial optimization:
> - SFT uses only **188 reject-sampling examples + 10k general reasoning examples**.
> - RL with a sampling factor of **n = 8** already delivers significant performance gains.
> - Curriculum learning reduces RL training data volume by **25×** (from ~200k full-sample data to ~8k medium-difficulty samples).
>
> For deployment, Fig. 7 demonstrates that **an RL-trained model with n = 1 matches or outperforms an SFT model with n = 32**. Furthermore, applying inference-time scaling (n = 32) to the RL model yields only marginal additional benefits. Thus, there is no need to concern oneself with test-time computational burden: **deploying the RL model with n = 1 sampling is fully sufficient**. Inference-time scaling can be extended theoretically for maximum performance, but this is optional and not a requirement of the RETuning framework.
>
> Below is the full computational cost table (training hyperparameters are provided in Appendix Tables 3–4; evaluation setup: 684 samples, average input of 28k tokens, average output of 1k tokens, 8×H100 GPUs with tensor parallelism (tp) = 4):
>
> | Base Model       | SFT Stage | RL Stage | Eval n=1 | Eval n=2 | Eval n=4 | Eval n=8 | Eval n=16 | Eval n=32 |
> | :--------------- | :-------- | :------- | :------- | :------- | :------- | :------- | :-------- | :-------- |
> | DeepSeek-R1-14B  | 5.2 h     | 28.5 h   | 13.4 min | 21.0 min | 33.2 min | 58.1 min | 1 h 50 min | 3 h 35 min |
> | DeepSeek-R1-32B  | 11.3 h    | 42.4 h   | 15.0 min | 19.4 min | 31.1 min | 52.5 min | 1 h 33 min | 3 h 02 min |
>
> ## Responses to Reviewer Questions
>
> **Q1. Chinese A-share characteristics and challenges when transferring to US markets**
>
> The Chinese A-share market is dominated by domestic investors and has unique cultural and regulatory attributes, with high sensitivity to political events. For example, amid recent China-Japan political frictions, **Furi Co., Ltd.** experienced seven consecutive daily limit-ups due to the phonetic similarity between its name and a Chinese phrase meaning “capturing Japanese troops.” To logically interpret such trends, LLMs must be familiar with Chinese historical context and adopt the perspective of domestic retail investors. This type of trend—driven by inferential links between news events and stock names—is rarely observed in U.S. equity markets, making cross-market reasoning generalizability difficult to guarantee.
>
> Additionally, as the reviewer noted, cross-market scenarios involve distinct information types (e.g., SEC filings, earnings call transcripts). Models must learn to process such data during the cold-start phase, which requires additional data synthesis.
>
> Finally, data quality remains a core challenge. Post-training (including SFT and GRPO) is highly sensitive to data “alignment quality.” Forcing predictions when prompts lack key information will only induce hallucinations, offering no meaningful improvement to predictive performance.
>
> **Q2. Hyperparameters (α, β, γ) and curriculum thresholds**
>
> First, we conducted a grid search over the reward function hyperparameters (α, β, γ) using the validation set and found they do not exert a statistically significant impact on model performance; we therefore assigned them equal weights in the final framework.
>
> Second, rather than performing a formal search for curriculum learning difficulty thresholds, we selected the medium-difficulty range based on the sample difficulty distribution in Fig. 5(b). This range ensures approximate balance across the three label categories, preserving label equilibrium during training.
>
> Notably, we observed that without curriculum learning, adjusting the reward function yields no meaningful improvements. This is because the price fluctuations underpinning the reward function follow an approximately normal distribution, rendering RL reward signals purely noisy. Only by using curriculum learning to filter a targeted subset of samples and conducting RL exclusively on this subset (while retaining price-fluctuation-based rewards) can the model’s predictive capabilities be effectively stabilized.
>
> **Q3. Precision/recall for up/down classes**
>
> Yes, we analyzed the precision and recall of predictions for upward and downward price movement classes. Notably, in contrast to conclusions from traditional financial applications, we found that **maintaining high recall is critical during RL training**. A low recall rate triggers prediction collapse during RL: the model converges rapidly to the class with the highest precision, leading to training failure.

---

### Official Review · Reviewer_mNU5 · 2025-11-04

**Soundness:** 2
**Presentation:** 2
**Contribution:** 2
**Rating:** 2
**Confidence:** 4

**Summary:**

This paper focuses on improving stock price movement prediction by applying reinforcement learning to a reasoning-oriented large language model (LLM) trained on a customized dataset. The dataset combines numerical features with textual information, such as news, and the authors incorporate additional contextual elements, such as analytical frameworks, evidence grouping, and reflection, into CoT. The reinforcement learning setup largely follows standard procedures.

While the proposed dataset is interesting, its limited temporal coverage reduces its potential impact and applicability in real-world financial scenarios. Moreover, the paper’s technical contribution to reinforcement learning for LLMs appears to be also limited.

**Strengths:**

1. This paper develops a multimodal dataset including numerical and text data of stock markets for enhancing the stock price classification with reasoning LLMs.

2. The experiment is conducted on multiple LLMs and financial tasks in addition to stock price classification.

**Weaknesses:**

1. Arbitrary labeling threshold
   The target setup appears arbitrary, as labels are defined using a fixed 3% threshold. Given that stock markets are highly dynamic with varying spreads, a more convincing rationale is needed to justify this static labeling choice.

2. Clarity of results in Figure 3
   Figure 3 is unclear. It is not specified which results correspond to experiments conducted without using chain-of-thought (CoT).

3. Language coverage in the dataset
   In the appendix, most of the analysis frameworks appear to be in Chinese. It is unclear whether the dataset includes other languages.

4. Limited testing period
   Due to the short temporal coverage of the data, testing is performed on only one month of data. For stock markets, such a short testing period is not convincing. In particular, the year-end market may exhibit low liquidity, short-term reversions, or profit-taking behaviors, making the results from this month less representative.

**Questions:**

Refer to the weaknesses section.

---

> ### Author Response · Authors · 2025-12-03
> **Rebuttal**
>
> We thank the reviewer for the detailed and insightful feedback. Below we address each concern concisely.
>
> 1. **Labeling threshold**: Thank you for the suggestion. We agree that market volatility varies over time. Our choice of ±3% just follow regulatory circuit-breaker conventions and common practitioner definitions of meaningful short-term movements in the A-share market. Considering dynamic labeling is an optional direction for future work. However, to the best of our knowledge, there is currently no widely adopted dynamic labeling method that can claim to accurately capture the market's highly dynamic nature with varying spreads. Furthermore, the so-called dynamic labeling methods actually introduce biases from human researchers, which are prone to causing **reward hacking** by human. This is not conducive to models truly relying on their own reasoning capabilities to learn predictive skills.
> 2. **Figure 3 clarity**: We appreciate this comment. We have revised Figure 3 and its caption to clearly distinguish results with and without CoT prompting, and we explicitly highlight the corresponding symbols and lines in the legend.
> 3. **Language coverage**: Thank you for raising this point. As  stated in Appendix C Line 1188-1190, the dataset focuses on Chinese A-share stocks, allowing the LLM to fully leverage its pre-trained knowledge in Chinese to analyze the Chinese stock market, so all data and analytical frameworks are in Chinese (no other languages). This alignment ensures consistency for Chinese-focused LLMs.
> 4. **Testing period**: We extended testing with Fin-2025[June] (6-month horizon) and OOD_Date splits to mitigate year-end bias. We will highlight these results in the revised manuscript to demonstrate temporal generalizability.

---

### Note · Authors · 2026-01-05

I have read and agree with the venue's withdrawal policy on behalf of myself and my co-authors.